# Exploring Structural Degradation in Dense Representations for Self-supervised Learning

**Siran Dai**[1,2]     **Qianqian Xu**[3,4] *     **Peisong Wen**[5]
**Yang Liu**[5]     **Qingming Huang**[5,3*]

[1] Institute of Information Engineering, Chinese Academy of Sciences
[2] School of Cyber Security, University of Chinese Academy of Sciences
[3] State Key Laboratory of AI Safety, Institute of Computing Technology, CAS
[4] Peng Cheng Laboratory
[5] School of Computer Science and Tech., University of Chinese Academy of Sciences

daisiran@iie.ac.cn     xuqianqian@ict.ac.cn
{wenpeisong, qmhuang}@ucas.ac.cn     liuyang232@mails.ucas.ac.cn

## Abstract

In this work, we observe a counterintuitive phenomenon in self-supervised learning (SSL): longer training may impair the performance of dense prediction tasks (e.g., semantic segmentation). We refer to this phenomenon as Self-supervised Dense Degradation (SDD) and demonstrate its consistent presence across sixteen state-of-the-art SSL methods with various losses, architectures, and datasets. When the model performs suboptimally on dense tasks at the end of training, measuring the performance during training becomes essential. However, evaluating dense performance effectively without annotations remains an open challenge. To tackle this issue, we introduce a Dense representation Structure Estimator (DSE), composed of a class-relevance measure and an effective dimensionality measure. The proposed DSE is both theoretically grounded and empirically validated to be closely correlated with the downstream performance. Based on this metric, we introduce a straightforward yet effective model selection strategy and a DSE-based regularization method. Experiments on sixteen SSL methods across four benchmarks confirm that model selection improves mIoU by 3.0% on average with negligible computational cost. Additionally, DSE regularization consistently mitigates the effects of dense degradation. Code is available at https://github.com/EldercatSAM/SSL-Degradation.

## 1 Introduction

Self-Supervised Learning (SSL) has greatly benefited from advancements in training algorithms, larger datasets and models, and extended training periods, leading to significant success in image-level representation learning [10, 16, 3, 29]. However, dense (patch or pixel-level) representation learning remains challenging with only slight improvements [70, 5].

While training models for extended periods to extract high-quality representations has been a common practice in SSL, we identify and study a counterintuitive phenomenon, named Self-supervised Dense Degradation (SDD), which helps explain the challenges in self-supervised dense representation learning. As illustrated in Fig. 1, although the training loss converges and classification performance steadily improves, the dense performance declines at the later stages of training. Consequently, the final checkpoint exhibits a significant performance gap compared to the best performance observed during training.

---

*Corresponding authors.

39th Conference on Neural Information Processing Systems (NeurIPS 2025).

Extensive experiments on sixteen state-of-the-art methods across four benchmarks confirm that the SDD phenomenon consistently appears across diverse training approaches and evaluation protocols. More importantly, it persists even when training and evaluation are conducted on the same dataset. This demonstrates that SDD highlights the performance inconsistency between different tasks rather than overfitting to the data distribution, introducing a new challenge to the SSL community.

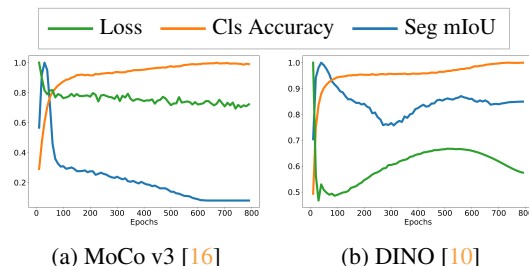

(a) MoCo v3 [16]      (b) DINO [10]

Figure 1: The Self-supervised Dense Degradation (SDD) phenomenon. While the training loss converges and classification performance gradually increases, the segmentation performance declines and the final checkpoint is not optimal. The metrics are normalized to 1 for a better illustration.

Because of SDD, the common practice of training until convergence results in suboptimal dense performance. Finding a metric to predict downstream dense performance thus becomes essential for both understanding the cause of SDD and reducing its negative impact. Typically, models are evaluated on a labeled validation set; however, the cost of evaluating even a single checkpoint can exceed that of one full epoch of pre-training, making this impractical. Moreover, SSL typically lacks access to downstream data or labels. Although previous studies have tried estimating SSL downstream performance in an unsupervised manner [2, 24, 69, 68, 36, 60], these metrics mainly focus on image-level tasks and are negatively correlated with dense-level performance in our experiments.

To guide the development of a better metric, we first provide a theoretical analysis based on error rate decomposition. We show that the downstream error rate is small if **1)** intra-class representation radius is smaller than inter-class distance, and **2)** effective dimensionality of representations is large.

Based on this analysis, we propose a Dense Representation Structure Estimator (DSE), which consists of measures for class-separability and effective dimensionality, directly corresponding to our theoretical findings. Using the DSE metric, we propose two strategies to counteract SDD. In the off-the-shelf setting, we suggest a simple yet effective method of selecting the checkpoint with the highest local DSE. In the online training setting, we integrate the DSE metric as a regularizer.

Empirical evaluations demonstrate that the proposed DSE accurately predicts downstream performance, significantly outperforming existing metrics. By visualizing components of the DSE, we explain that the cause of the SDD phenomenon is either a loss of class separability or dimensional collapse. For example, in Fig. 1, MoCo v3's performance drop is due to dense dimensional collapse, while DINO's degradation arises from decreased class separability. Moreover, our proposed checkpoint selection approach improves the mIoU by 3.0% on average, and incorporating DSE into training further improves both DSE scores and downstream dense performance, effectively eliminating SDD's negative effects.

We summarize our contributions as follows:

- We identify the SDD phenomenon in SSL, revealing an inconsistency between image-level and dense-level performance, which is prevalent in state-of-the-art methods and negatively impacts dense tasks.

- We introduce the Dense Representation Structure Estimator (DSE), which accurately and efficiently predicts downstream dense performance without relying on downstream data.

- Using the DSE metric, we propose model selection and regularization strategies that effectively reduce the negative impact of the SDD phenomenon.

- Extensive experiments on sixteen leading SSL methods across four benchmarks demonstrate the precision of DSE and the effectiveness of the proposed approaches in addressing the SDD phenomenon.

## 2   Related Work

### 2.1   Self-supervised Learning

**Contrastive Learning.** Contrastive self-supervised methods have shown significant progress in recent years [11, 73, 28, 61, 62]. These methods typically construct positive examples (different augmented views of an image) and negative examples (samples from other images), then use contrastive loss [49] to train models to differentiate between them [12, 15, 78, 4, 32, 27, 30].

**Non-contrastive Learning.** Recent advances in self-supervised learning avoid explicit negative examples. These methods often employ siamese networks and aim to achieve cross-view consistency by aligning representations between teacher and student networks [26, 10, 83, 50, 14, 9].

**Masked Image Modeling.** Masked image modeling can be framed as a generative task. Models are trained to reconstruct original images from masked inputs [29, 74, 3, 13]. Recent works [83, 50] achieved great success by combining latent-space reconstruction with self-distillation.

**Dense Representation Learning.** Another line of work focuses on learning dense representations. Using techniques such as dense alignment [86, 70, 31, 71], clustering [84, 58, 40, 59, 5], and reconstruction [83, 50], these methods achieve strong results on dense prediction tasks. However, despite optimizing dense representations directly, performance degradation is still observedduring training.

### 2.2   Unsupervised Transferability Estimation

Recently, $\alpha$-REQ [2] uses the parameter of power-law distributions of covariance matrix singular values as a metric. RankMe [24] employs the effective rank [53] to estimate the transferability performance, and Lidar [60] further improves it by introducing linear discriminative analysis. Other approaches analyze feature activation statistics [36], coding rate reduction [77] or model memorization effects [69, 68]. Despite progress in image-level tasks, evaluating dense representations remains an open challenge. Due to space limitation, more related works are discussed in Appendix B.

## 3   The Self-supervised Dense Degradation Phenomenon

In self-supervised learning, models are typically trained for long periods. It is widely recognized that downstream classification performance generally improves as training loss converges [10, 14, 16]. However, we observe that dense performance actually degrades during pretraining, with the performance at the final checkpoint being significantly worse than that of the best model. This observation contradicts previous intuitions. We term this phenomenon **Self-supervised Dense Degradation (SDD)**, and empirical and theoretical analyze this phenomenon in this section.

### 3.1   Empirical Observations of the SDD Phenomenon

**SDD is a General and Harmful Phenomenon.** To investigate whether SDD occurs broadly, we conduct extensive experiments. The main findings are summarized in Fig. 2. These experiments confirm that SDD occurs consistently across **1) Various Pre-training Approaches**: SDD exists in sixteen state-of-the-art methods across different types of training loss, model architecture and optimization strategies, and **2) Various Evaluation Protocols**: SDD is evident in both linear probing and transfer learning scenarios (where the backbone is not frozen), spanning diverse downstream datasets, evaluation hyperparameters, and tasks. Detailed results can be found in Appendix E. **SDD is Not Caused by Overfitting the Training Data.** We further investigate whether SDD stems from memorizing the training data. To test this, we train and evaluate DINO [10] on the same COCO dataset. The trend of dense performance mirrors that observed for DINO trained on ImageNet, with the final checkpoint experiencing a significant degradation of $4.0\%$ in mIoU. This result indicates that SDD is not due to overfitting. Further details are presented in Appendix E.3.

Since SDD occurs broadly and is not related to dataset overfitting, identifying a suitable metric to predict downstream performance would be valuable for understanding and mitigating this degradation. However, defining such a metric is challenging. In the following subsection, we seek theoretical insights to help develop such a performance measure.

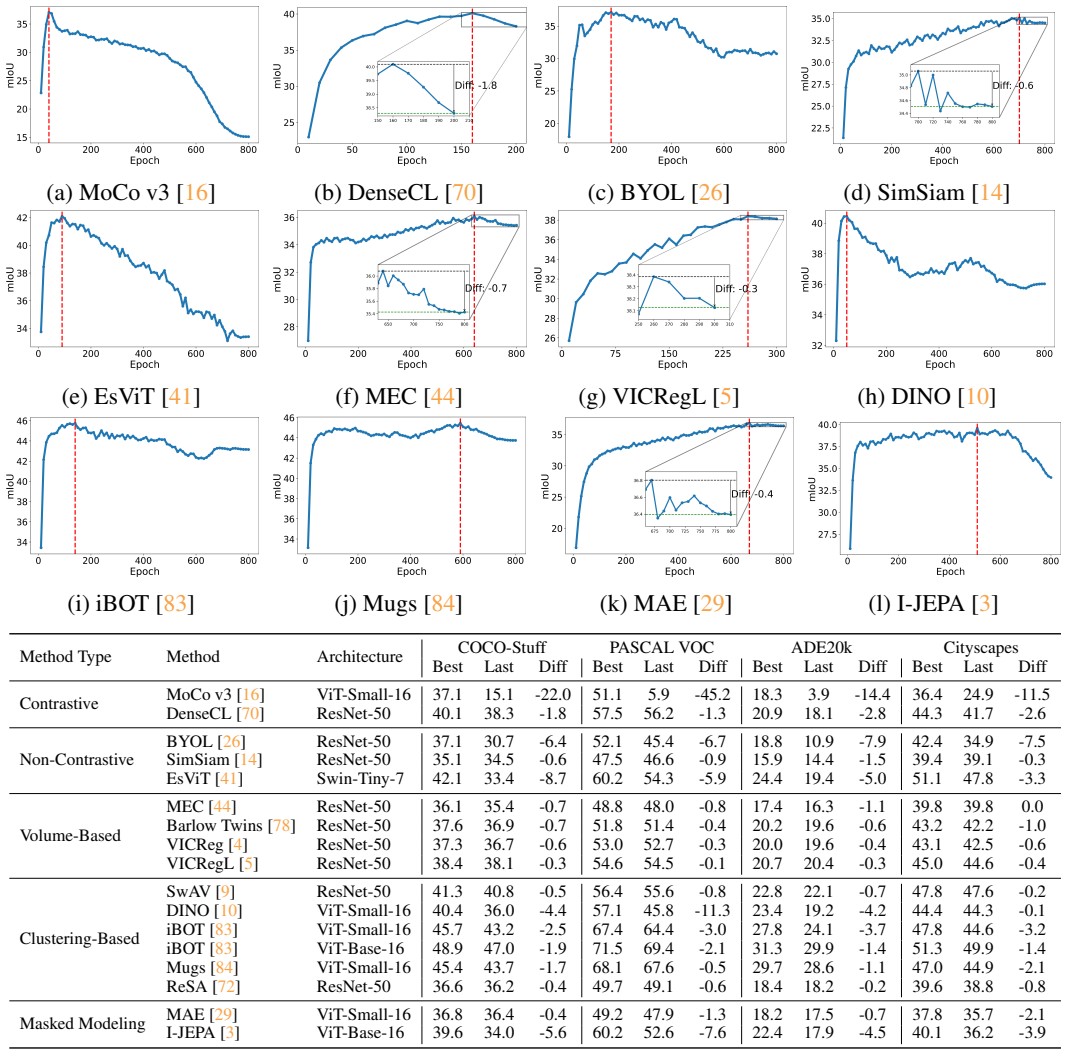

Figure 2: **Top:** The change in dense performance throughout the pretraining process, assessed via linear segmentation on the COCO-Stuff dataset. Performance degradation is consistently observed across all methods. **Bottom:** A performance gap between the best and the last models is present across all datasets and methods.

## 3.2 Theoretical Analysis of the SDD Phenomenon

Our goal is to establish a theoretically grounded metric for downstream performance. To achieve this, we analyze the downstream performance and identify two crucial factors influencing it: **1)** class separability, quantified by the difference between inter-class distance and intra-class radius (formally presented in Thm. **2**), and **2)** the dimensionality of the representations (Cor. **5**). With these insights, readers interested primarily in methodology and experiments may skip the remainder of this section without any loss of continuity.

### 3.2.1 Problem Formulation

Given that SDD appears across various methods, analyzing their training processes within a single unified framework is challenging. Therefore, we mainly focus on the linear probing approach, which aims to train a classifier using fixed (dense) representations. Specifically, given a downstream dataset $\mathcal{D} = \{X_i\}_{i=1}^{\bar{N}}$, consisting of $\bar{N}$ images, a fixed encoder $f_{\boldsymbol{\theta}} : \mathcal{X} \rightarrow \mathbb{R}^{N \times d}$ produces $N$ dense representations $\{\boldsymbol{z}_i\}_{i=1}^{N}$ for each image, where $d$ is the dimension of the representations. To simplify, we formulate dense linear probing as a classification problem, where each representation

$\boldsymbol{z}_i$ is assigned to one of $K$ latent classes, represented as $y(\boldsymbol{z}_i)$. The aim of linear probing is to train a classifier $G(\boldsymbol{z})$ (for example, a linear head) that accurately maps each $\boldsymbol{z}$ to its correct latent class. Following the analysis in [34], we choose a simple Nearest Neighbor (NN) classifier:

$$G(\boldsymbol{z}) = \arg\min_{k\in[K]} ||\boldsymbol{z} - \boldsymbol{\mu}_k||,$$

where $\boldsymbol{\mu}_k = \mathbb{E}_{\boldsymbol{z}:y(\boldsymbol{z})=k}[\boldsymbol{z}]$ denotes the center of the representations for the $k$-th class. The error rate of the fixed encoder $f_{\boldsymbol{\theta}}$ on the downstream dataset is given by:

$$\mathrm{Err}_{\mathcal{D}}(f_{\boldsymbol{\theta}}) = \mathbb{E}_{\boldsymbol{x}\in\mathcal{D}}\left[\mathbb{P}_{\boldsymbol{z}\in f_{\boldsymbol{\theta}}(\boldsymbol{x})}\left[y(\boldsymbol{z}) \neq G(\boldsymbol{z})\right]\right].$$

Since the NN classifier can be viewed as a special case of any linear classifier, its error rate naturally serves as an upper bound for all classifiers.

### 3.2.2 Decomposing the Downstream Error Rate

Next, we decompose downstream performance and identify the factors that influence downstream accuracy. For the NN classifier $G$, a representation can be correctly classified if the following condition holds:

$$\underbrace{||\boldsymbol{z} - \boldsymbol{\mu}_{y(\boldsymbol{z})}||}_{\text{Intra-class distance}} - \underbrace{\min_{k\in[K]\setminus y(\boldsymbol{z})} ||\boldsymbol{z} - \boldsymbol{\mu}_k||}_{\text{Inter-class distance}} \leq 0. \tag{1}$$

Inspired by this relationship, downstream performance can be expressed in terms of intra and inter-class distances. However, directly measuring these distances is not feasible. The main challenge arises from the fact that, in a self-supervised setting, class labels $y(\boldsymbol{z})$ are unavailable. While techniques such as $k$-means clustering can be utilized to generate pseudo labels, we find that instance-wise distance measures still face a critical issue, resulting in meaningless predictions:

**Proposition 1.** *The instance-wise intra-class distance is always smaller than the inter-class distance when using $k$-means pseudo-labels. Thus, the estimated accuracy is always 1, regardless of the actual situation.*

Proof can be found in the Appendix A. The fundamental reason for this issue lies in the fact that the instance-wise distance measure tends to underestimate the intra-class distance for those examples near the decision boundary. We present an illustration of this issue in Fig. 3. To address this issue, we replace the instance-wise measure with a class-wise radius. As a result, we reformulate the condition that $\boldsymbol{z}$ could be correctly classified as:

$$\underbrace{\mathcal{R}_{y(\boldsymbol{z})}}_{\text{Intra-class radius}} - \underbrace{\min_{k\in[K]\setminus y(\boldsymbol{z})} ||\boldsymbol{z} - \boldsymbol{\mu}_k||}_{\text{Inter-class distance}} \leq 0. \tag{2}$$

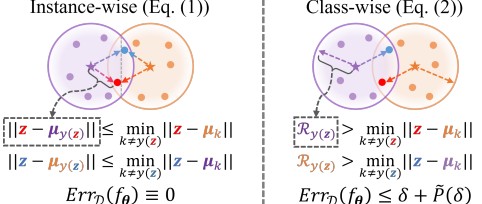

Figure 3: **Left:** Instance-wise condition (Eq. 1) predicts all examples in the intersection area as **correctly** classified, leading to an inaccurate error rate estimation. **Right:** Our class-wise condition (Eq. 2). Examples in the intersection area are accurately predicted as **misclassified**.

Building on this idea, we further demonstrate that under the assumption that the representations within each class are concentrated (e.g., following a sub-Gaussian distribution), the downstream performance can be guaranteed.

**Theorem 2** (Class-relevant Measure for Downstream Performance). *Let $Z^j = \{Z : y(Z) = j\}$ be the set of examples in the $j$-th class with $|Z^j| = N_j$. Assume that for all $j \in [K]$, the examples $\{\boldsymbol{z}_i^j\}_{i=1}^{N_j}$ in $Z^j$ are i.i.d. $R$-sub-Gaussian random vectors in $\mathbb{R}^d$. Denote $\bar{\boldsymbol{z}}^j = \frac{1}{N_j}\sum_{i=1}^{N_j} \boldsymbol{z}_i^j$ and $Z_c^j = \left[\boldsymbol{z}_1^j - \bar{\boldsymbol{z}}^j, \cdots, \boldsymbol{z}_N^j - \bar{\boldsymbol{z}}^j\right]$ as the centered embedding matrix for $Z^j$. Then, for any $\delta > 0$:*

$$\underbrace{Err_{\mathcal{D}}(f_{\boldsymbol{\theta}})}_{\text{Downstream error rate}} \leq \delta + \mathbb{P}_{\boldsymbol{z}}\left(\underbrace{D_{\min}^{\boldsymbol{z}}}_{\text{Inter-class distance}} - \underbrace{\frac{\sum_{i=1}^d \sigma_i(Z_c^{y(\boldsymbol{z})})}{\sqrt{N_{y(\boldsymbol{z})}-1}}}_{\text{Estimated intra-class radius}} < C_\delta\right). \tag{3}$$

*Here, $\sigma_i(\cdot)$ represents the $i$-th singular value, $C_\delta$ is a margin term that jointly determined by $R, \delta$, and $N_j$ (please refer to Appendix A.2.3 for exact formulation), and $D_{\min}^{\boldsymbol{z}} = \min_{k\in[K]\setminus y(\boldsymbol{z})} ||\boldsymbol{z} - \boldsymbol{\mu}_k||$ denotes the minimal inter-class distance.*

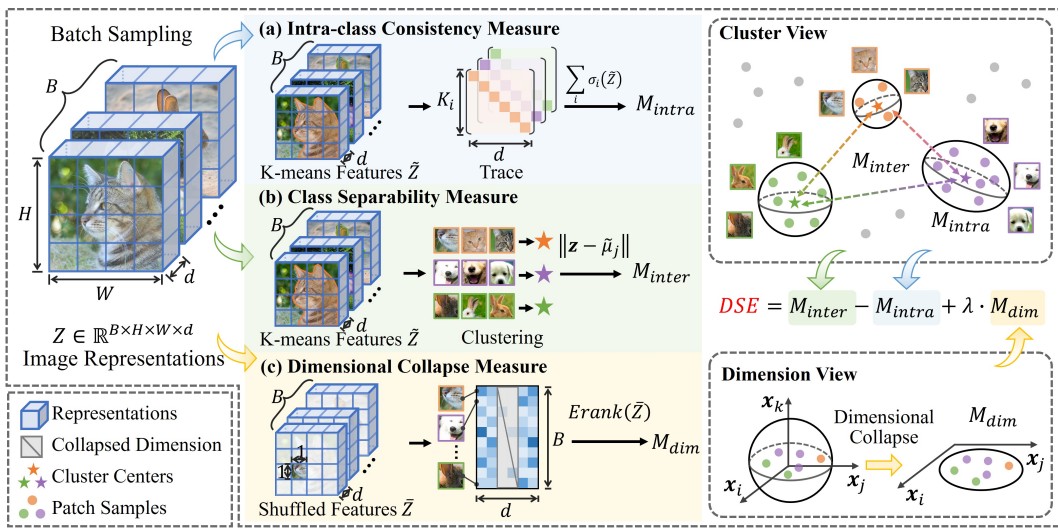

Figure 4: The proposed Dense representation Structure Estimator (DSE) consists of three components. The intra-class consistency measure and inter-class distance measure jointly conduct the class-separability measure, which is motivated by the results in Thm 2. The dimensional collapse measure estimates the effective dimensionality, which corresponds to the analysis in Cor. 5.

**Remark 3.** In this theorem, we estimate the intra-class radius $\mathcal{R}_{y(z)}$ with the normalized trace of the representation matrix. When this radius (plus a margin term $\tilde{C}_\delta$) is smaller than the inter-class distance, a simple NN classifier would be enough to separate these representations.

**Remark 4.** For simplicity, we ignore the label estimation error in this theorem. A complete version, including the effect of $k$, is provided in Appendix A.4. Briefly, the bound is tightest when $k$ equals the true number of classes, but it generally remains valid when $k$ exceeds the actual number of classes.

Next, we reveal another key factor affecting the downstream error rate: the dimensionality of representations.

**Corollary 5** (Error Rate Decay with Dimensionality). *Under Thm. 2's assumptions with* $\min_{k \neq y(z)} \|\boldsymbol{\mu}_{y(z)} - \boldsymbol{\mu}_k\| > \sqrt{d}R \left( 2 + \sqrt{\frac{\log(8/\delta)}{N_j}} + \sqrt{3} \right)$, *for any* $\delta > 0$:

$$Err_{\mathcal{D}}(f_{\boldsymbol{\theta}}) \leq \delta + 2K \exp\left( -\tilde{C}_\delta \cdot d \right), \tag{4}$$

*where* $\tilde{C}_\delta > 0$ *is a constant.*

**Remark 6.** This corollary establishes a connection between our analysis and the broadly studied dimensional collapse phenomenon in SSL [24, 85]. When $d$ is small, the downstream performance experiences significant degradation. These findings highlight the critical need to jointly assess the dimensionality of representations.

## 4 Addressing SDD via Dense Representation Structure Estimator

### 4.1 Dense Representation Structure Estimator

Inspired by the analysis in the previous section, we propose a metric called Dense representation Structure Estimator (DSE) with the following formulation:

$$\text{DSE} = \mathbb{E}_{\boldsymbol{z}} \underbrace{\left[ D_{\min}^{\boldsymbol{z}} - \frac{\sum_{i=1}^d \sigma_i(Z_c^{y(z)})}{\sqrt{(N_{y(z)} - 1)}} \right]}_{\text{Class separability measure}} + \underbrace{\lambda \cdot M_{dim.}}_{\text{Effective dimensionality}}$$

The first term is derived from the result of Thm. 2. Since the cumulative density function in Eq. 3 decreases monotonically with the deviation, the deviation between two terms can be treated as a

---

**Algorithm 1** DSE-based Model Selection

---

**Input:** Training dataset $X$, checkpoints $\{f_{\boldsymbol{\theta}}^i\}_{i=1}^N$, maximum number of candidates $T$.
**for** $i = 1$ **to** $N$ **do**
    Sample a batch of data $\bar{X}$ from $X$ and calculate the dense representations $Z = f_{\boldsymbol{\theta}}^i(X)$
    Calculate the metric $P_i$ based on Eq. 5
**end for**
Select the local maximum points by $\bar{C} = \{i : i = \arg\max_{j \in [i-2, i+2]} P_j\}$
Keep the indices in $\bar{C}$ with the top-$T$ metric and obtain $C = \{f_{\boldsymbol{\theta}}^i : i \in \bar{C}\}$.
**Output:** Model candidates $C$.

---

measure of downstream performance. The second term corresponds to the analysis in Cor. 5. For an intuitive understanding, readers can refer to Fig. 4.

**Measuring Class Separability.** Given a batch of dense representations $Z = \{\boldsymbol{z}_i\}_{i=1}^{B \times N}$, we first calculate the $k$-means on all $B \times N$ representations to obtain a pseudo-label $\tilde{y}(\boldsymbol{z}) \in [k]$ for all representations $\boldsymbol{z}$. Let $\tilde{Z}^j = \{\boldsymbol{z} \in Z : y(\boldsymbol{z}) = j\}$ denotes the representations in the $j$-th cluster and $\tilde{N}_j = |\tilde{Z}^j|$ represents the number of representations, the intra-class radius and inter-class distance are calculated as:

$$M_{intra} = \frac{1}{k} \sum_{j=1}^{k} \frac{\sum_{i=1}^{\min\{\tilde{N}_j, d\}} \sigma_i(\tilde{Z}_c^j)}{\sqrt{(\tilde{N}_j - 1)}}, \quad M_{inter} = \frac{1}{k} \sum_{j=1}^{k} \frac{1}{N_j} \sum_{\boldsymbol{z} \in \tilde{Z}^j} \min_{i \neq j} \|\boldsymbol{z} - \tilde{\boldsymbol{\mu}}_i\|_2.$$

where $\tilde{Z}_c^j = \tilde{Z}^j - \mathbf{1}\frac{1}{\tilde{N}_j} \sum_{i=1}^{\tilde{N}_j} \boldsymbol{z}_i^T$ is the centered representation matrix, and $\tilde{\boldsymbol{\mu}}_j = \frac{1}{|\tilde{Z}^j|} \sum_{\boldsymbol{z} \in \tilde{Z}^j} \boldsymbol{z}$ represents the center of the $j$-th cluster.

**Measuring Dimensional Collapse.** As discussed in Cor. 5, the dimensionality of representations affects their separability, and thus it should also be considered. Building on previous work [24], we first randomly sample $B'$ dense representations from different images (ensuring their independence) and concatenate them to a $B' \times d$ matrix $\bar{Z}$. By setting $B' \gg d$, the rank of $\bar{Z}$ reflects the number of non-collapsed dimensions of representations. Thus, we compute its effective rank [53]:

$$M_{dim} = \text{Erank}(\bar{Z}) = \exp\left(-\sum_{i=1}^{d} p_i \log p_i\right),$$

where $p_i = \frac{\sigma_i(\bar{Z})}{||\sigma_i(\bar{Z})||_1}$ is the $i$-th normalized singular value of $\bar{Z}$.

**Final Formulation of DSE.** DSE is calculated by:

$$\text{DSE} = M_{inter} - M_{intra} + \lambda \cdot M_{dim}, \tag{5}$$

where $\lambda$ is a parameter that rescales the measure of effective dimensionality to the same amplitude of class-separability statistics. In practice, it is taken as:

$$\lambda = \frac{\text{Std}(M_{inter} - M_{intra})}{\text{Std}(M_{dim})},$$

where $\text{Std}(\cdot)$ denotes the standard deviation calculated across all checkpoints.

## 4.2 Mitigating SDD Phenomenon with the DSE Metric

**DSE-Guided Off-the-shelf Model Selection.** When modifying the training process is not feasible, we propose selecting the best model from the saved checkpoints using the DSE metric to reduce the negative impact of the SDD phenomenon. When comparing two models, the one with a higher DSE indicates better class separability and effective dimensionality, and is therefore expected to perform better according to our theory. Based on this idea, we first compute the DSE metric and select checkpoints corresponding to local maxima as potential candidates. To reduce computational costs, we then choose the top $T$ ($T = 3$) checkpoints with the highest DSE values from this candidate set as our final models. The complete procedure is shown in Alg. 1.

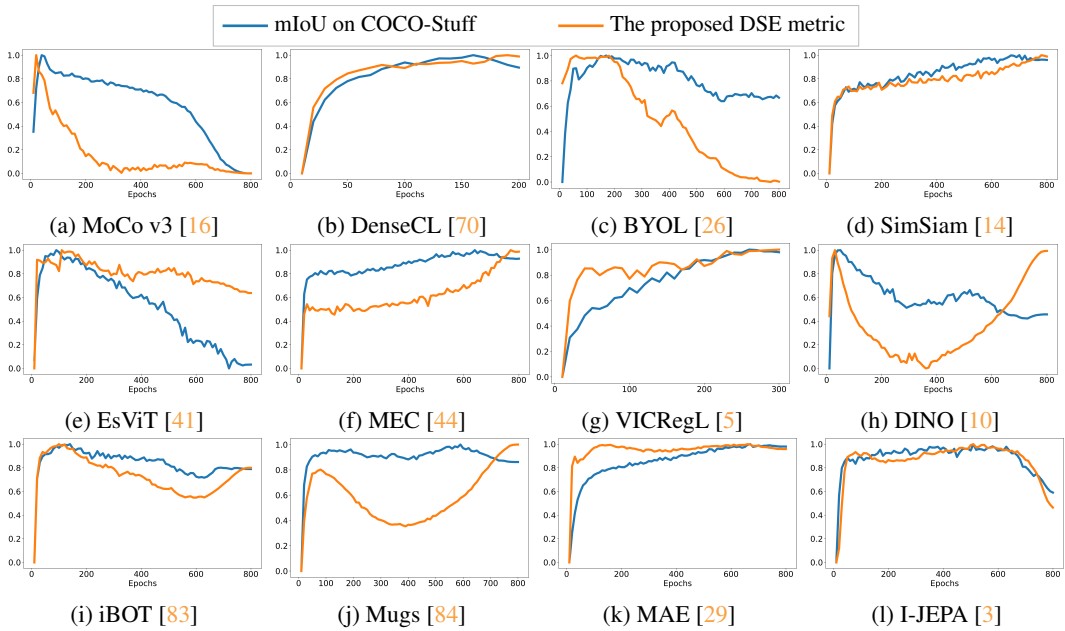

Figure 5: The proposed DSE metric precisely predicts the downstream performance.

**DSE-regularized Online Optimization.** Since the DSE metric represents a lower bound on performance and all operations involved in computing DSE are differentiable, we also consider directly optimizing the DSE as an explicit regularizer. Specifically, we add the negative DSE metric to the original loss function for each learning framework:

$$\mathcal{L} = \mathcal{L}_{original} - \beta \cdot \text{DSE}.$$

In our experiments, we set $\lambda$ in DSE to 1 and $\beta$ to 0.001. Empirically, we find that training for 10 epochs starting from the checkpoint with the best initial performance effectively mitigates the SDD phenomenon and enhances downstream performance. More details are presented in Appendix C.

## 5 Empirical Studies

### 5.1 DSE Metric is a Precise Estimator for Dense Performance

**Experiment Setup.** To assess the effectiveness of the proposed DSE metric, we examine the correlation between DSE and downstream segmentation performance across sixteen SSL methods and four datasets. We leverage linear transfer learning as the downstream task, with settings remain the same as introduced in the previous section. To save time, only 2048 images ($\sim 0.16\%$) of the **training dataset** (ImageNet-1k) are used to compute the metric, and both the metric and dense performance are evaluated every 10 epochs.

To validate the correlation between the proposed metric and downstream performance, the results are evaluated using Kendall's $\tau$ coefficient [37]. The coefficient ranges from $[-1, 1]$. When $\tau = 1$, the metric is perfectly aligned with the downstream performance, and $\tau = -1$ represents that they are inversely correlated. We provide a detailed explanation in the App. D.

**Analysis of the Empirical Results.** We draw two main conclusions from Fig. 5 and Tab. 1. **1) The DSE metric accurately reflects downstream performance.** The metric curve consistently aligns with downstream performance across different datasets and methods. Hypothesis tests yield an average Kendall's $\tau$ coefficient of 0.57, confirming the reliability of DSE. **2) Compared to existing estimators, DSE is better suited for dense tasks.** Current estimators are ineffective for dense performance evaluation due to the SDD phenomenon. Even if these estimators are adapted for dense representations (specifically, the adapted RankMe [24] is equivalent to our $M_{dim}$), DSE still significantly outperforms them. The advantage of DSE stems from its ability to more comprehensively characterize dense performance, as analyzed in detail in the following subsection. Additional empirical analyses, omitted here due to space constraints, are provided in Appendix F.

Table 1: **Left: Kendall's $\tau$ coefficient of the DSE metric.** We denote the insignificant results with $*$ ($p > 0.05$), otherwise, the results are significant with $p < 0.005$. **Right: Comparison of Kendall's $\tau$ coefficients for different estimators.** Methods with $\dagger$ are adapted to dense representations.

| Method | COCO | PVOC | ADE20k | Cityscapes |
|---|---|---|---|---|
| MoCo v3 [16] | 0.55 | 0.58 | 0.60 | 0.45 |
| DenseCL [70] | 0.70 | 0.81 | 0.52 | 0.63 |
| BYOL [26] | 0.61 | 0.48 | 0.79 | 0.78 |
| SimSiam [14] | 0.82 | 0.84 | 0.51 | 0.76 |
| EsViT [41] | 0.70 | 0.58 | 0.65 | 0.58 |
| MEC [44] | 0.68 | 0.65 | 0.15* | 0.63 |
| Barlow Twins [78] | 0.55 | 0.57 | 0.53 | 0.61 |
| VICReg [4] | 0.73 | 0.75 | 0.74 | 0.76 |
| VICRegL [5] | 0.72 | 0.74 | 0.72 | 0.72 |
| SwAV [9] | 0.90 | 0.91 | 0.89 | 0.88 |
| DINO [10] | 0.00* | 0.07* | 0.15* | 0.42 |
| iBOT [83] | 0.68 | 0.64 | 0.49 | 0.07* |
| Mugs [84] | 0.01* | 0.47 | 0.46 | 0.20 |
| ReSA [72] | 0.74 | 0.72 | 0.71 | 0.64 |
| MAE [29] | 0.38 | 0.44 | 0.46 | 0.21 |
| I-JEPA [3] | 0.49 | 0.38 | 0.59 | 0.28 |

| Estimator | Images ↓ | COCO | VOC | ADE | City | Avg |
|---|---|---|---|---|---|---|
| $\alpha$-ReQ [2] | 25600 | -0.07 | -0.05 | -0.05 | 0.09 | -0.02 |
| RankMe [24] | 25600 | -0.10 | -0.09 | -0.14 | 0.00 | -0.08 |
| Lidar [60] | 10000 | -0.37 | -0.36 | -0.26 | -0.21 | -0.30 |
| $\alpha$-ReQ$^\dagger$ [2] | 2048 | 0.17 | 0.19 | 0.11 | 0.10 | 0.14 |
| RankMe$^\dagger$ [24] | 2048 | 0.25 | 0.26 | 0.22 | 0.23 | 0.24 |
| Lidar$^\dagger$ [60] | 2048 | 0.38 | 0.37 | 0.33 | 0.23 | 0.33 |
| **DSE (Ours)** | **2048** | **0.58** | **0.60** | **0.56** | **0.49** | **0.57** |

It is noteworthy that the DSE metric is theoretically derived from the class-relevance downstream tasks like semantic segmentation. To see if DSE can accurately predict the dense performance beyound segmentation, we present more results on depth estimation task in the Appendix F.1.

## 5.2 DSE Metric Demystifies the SDD Phenomenon

Based on these observations, we provide insights into the causes of the SDD phenomenon from two perspectives: 1) SDD arises from insufficient class separability, reduced effective dimensionality, or both. While SSL aims to balance semantic alignment and the effective dimensionality of representations, these two objectives often involve a trade-off. For instance, fully uniform representations have high effective dimensionality but may lack separability; conversely, focusing too much on semantic alignment can lead to dimension collapse. We argue that the degradation seen in different methods is due to the failure to maintain this trade-off during training, which causes a bias toward one objective. 2) The reasons for degradation depend specifically on the method. As illustrated in Fig. 6, MoCo v3's performance degradation is due to dimensional collapse in dense features, which reduces representation separability. Additionally, the unusual performance drop observed in DINO at around 300 epochs corresponds to a slower reduction in intra-class distances relative to inter-class distances, decreasing overall class separability.

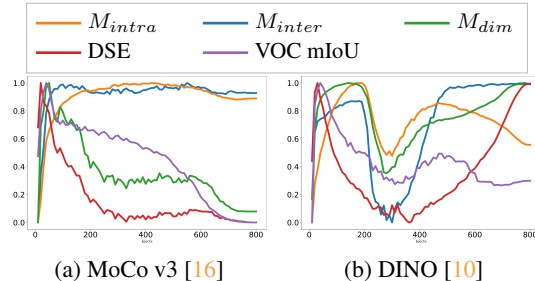

(a) MoCo v3 [16]    (b) DINO [10]

Figure 6: Different components of DSE metric.

These insights also explain why DSE improves over other estimators. Although other estimators accurately measure effective dimensionality, they fail to predict collapses caused by reduced class separability. DSE metric addresses this by providing a more complete measure of dense performance.

## 5.3 DSE-based Approaches Effectively Mitigate the Negative Impact of SDD

**DSE-based Model Selection is Accurate and Efficient.** We validate the effectiveness of our proposed model selection method on four benchmark datasets. As shown in Tab. 2, our model selection consistently improves mIoU and accuracy across all methods and datasets, achieving an average improvement of $3.0\%$ in mIoU. Compared to the previous state-of-the-art method, iBOT, our approach further improves the best mIoU by an average of $2.5\%$.

Table 3: Our approach achieves a performance gain comparable to supervised oracle, without requiring testing data or labels, and with negligible computational cost.

| Estimator | Data | Label | $\Delta$ mIoU ↑ | GPU hours ↓ |
|---|---|---|---|---|
| Loss | | | -1.0 | 0.0 |
| Supervised | ✓ | ✓ | +3.6 | 2.43 |
| DSE | | | +3.0 | 0.025 ($\sim 0.01\times$) |

Table 2: The overall performance of DSE-based model selection.

| Method | Architecture | COCO-Stuff | | PASCAL VOC | | ADE20k | | Cityscapes | |
|---|---|---|---|---|---|---|---|---|---|
| | | mIoU | Acc | mIoU | Acc | mIoU | Acc | mIoU | Acc |
| MoCo v3 [16] | ViT-Small-16 | 15.1 | 53.9 | 5.9 | 75.5 | 3.9 | 49.7 | 24.9 | 83.5 |
| +MS | ViT-Small-16 | 30.9 (+15.8) | 69.4 (+15.5) | 42.0 (+36.1) | 85.5 (+10.0) | 16.0 (+12.1) | 63.2 (+13.5) | 34.8 (+9.9) | 86.8 (+3.3) |
| DenseCL [70] | ResNet-50 | 38.3 | 72.0 | 56.2 | 89.2 | 18.1 | 64.5 | 41.7 | 87.1 |
| +MS | ResNet-50 | 39.7 (+1.4) | 72.8 (+0.8) | 56.9 (+0.7) | 89.5 (+0.3) | 20.5 (+2.4) | 66.0 (+1.5) | 43.4 (+1.7) | 87.8 (+0.7) |
| BYOL [26] | ResNet-50 | 30.7 | 65.2 | 45.4 | 85.8 | 10.9 | 57.8 | 34.9 | 84.5 |
| +MS | ResNet-50 | 37.1 (+6.4) | 70.3 (+5.1) | 51.1 (+5.7) | 87.9 (+2.1) | 18.7 (+7.8) | 63.5 (+5.7) | 42.2 (+7.3) | 87.3 (+2.8) |
| SimSiam [14] | ResNet-50 | 34.5 | 67.8 | 46.6 | 86.7 | 14.4 | 59.7 | 39.1 | 85.4 |
| +MS | ResNet-50 | 35.0 (+0.5) | 68.1 (+0.3) | 47.0 (+0.4) | 86.9 (+0.2) | 15.6 (+1.2) | 60.5 (+0.8) | 39.3 (+0.2) | 85.6 (+0.2) |
| EsViT [41] | Swin-Tiny-7 | 33.4 | 66.3 | 54.3 | 87.6 | 19.4 | 61.3 | 47.8 | 88.9 |
| +MS | Swin-Tiny-7 | 41.6 (+8.2) | 73.6 (+7.3) | 59.8 (+5.5) | 89.7 (+2.1) | 24.4 (+5.0) | 67.5 (+6.2) | 50.8 (+3.0) | 89.5 (+0.6) |
| MEC [44] | ResNet-50 | 35.4 | 67.8 | 48.0 | 87.1 | 16.3 | 60.3 | 39.8 | 85.4 |
| +MS | ResNet-50 | 35.6 (+0.2) | 68.0 (+0.2) | 48.5 (+0.5) | 87.2 (+0.1) | 17.3 (+1.0) | 60.7 (+0.4) | 39.7 (-0.1) | 85.4 (+0.0) |
| Barlow Twins [78] | ResNet-50 | 36.9 | 69.0 | 51.4 | 88.1 | 19.6 | 62.4 | 42.3 | 86.8 |
| +MS | ResNet-50 | 37.4 (+0.5) | 69.3 (+0.3) | 51.6 (+0.2) | 88.2 (+0.1) | 19.9 (+0.3) | 62.6 (+0.2) | 42.8 (+0.5) | 87.0 (+0.2) |
| VICReg [4] | ResNet-50 | 36.7 | 69.2 | 52.7 | 88.1 | 19.6 | 62.5 | 42.5 | 86.7 |
| +MS | ResNet-50 | 37.2 (+0.5) | 69.5 (+0.3) | 53.0 (+0.3) | 88.3 (+0.2) | 19.8 (+0.2) | 62.6 (+0.1) | 42.8 (+0.3) | 86.9 (+0.2) |
| VICRegL [5] | ResNet-50 | 38.1 | 71.5 | 54.5 | 88.7 | 20.4 | 64.6 | 44.6 | 87.9 |
| +MS | ResNet-50 | 38.3 (+0.2) | 71.6 (+0.1) | 54.6 (+0.1) | 88.8 (+0.1) | 20.7 (+0.3) | 64.8 (+0.2) | 44.8 (+0.2) | 88.0 (+0.1) |
| SwAV [9] | ResNet-50 | 40.8 | 72.5 | 55.6 | 89.0 | 22.1 | 65.6 | 47.6 | 88.7 |
| +MS | ResNet-50 | 41.0 (+0.2) | 72.5 (+0.0) | 56.0 (+0.4) | 89.0 (+0.0) | 22.8 (+0.7) | 65.7 (+0.1) | 47.6 (+0.0) | 88.7 (+0.0) |
| DINO [10] | ViT-Small-16 | 36.0 | 69.7 | 45.8 | 87.3 | 19.2 | 64.8 | 44.3 | 89.1 |
| +MS | ViT-Small-16 | 40.1 (+4.1) | 74.5 (+4.8) | 56.3 (+10.5) | 89.8 (+2.5) | 22.7 (+3.5) | 68.3 (+3.5) | 44.3 (+0.0) | 89.1 (+0.0) |
| iBOT [83] | ViT-Small-16 | 43.2 | 73.6 | 64.4 | 91.7 | 24.1 | 69.6 | 44.6 | 89.2 |
| +MS | ViT-Small-16 | 45.4 (+2.2) | 76.5 (+2.9) | 66.8 (+2.4) | 92.5 (+0.8) | 27.5 (+3.4) | 71.5 (+1.9) | 46.5 (+1.9) | 89.7 (+0.5) |
| MAE [29] | ViT-Small-16 | 36.4 | 71.7 | 47.9 | 88.2 | 17.5 | 65.1 | 35.7 | 87.0 |
| +MS | ViT-Small-16 | 36.7 (+0.3) | 71.9 (+0.2) | 48.9 (+1.0) | 88.2 (+0.0) | 18.2 (+0.7) | 65.5 (+0.4) | 37.2 (+1.5) | 87.6 (+0.6) |
| Mugs [84] | ViT-Small-16 | 43.7 | 74.9 | 67.6 | 92.3 | 28.6 | 70.7 | 44.9 | 89.2 |
| +MS | ViT-Small-16 | 44.6 (+0.9) | 75.9 (+1.0) | 67.6 (+0.0) | 92.3 (+0.0) | 28.6 (+0.0) | 70.7 (+0.0) | 45.5 (+0.6) | 89.0 (-0.2) |
| ReSA [72] | ResNet-50 | 36.2 | 68.2 | 49.1 | 87.0 | 18.2 | 61.2 | 38.8 | 84.7 |
| +MS | ResNet-50 | 36.6 (+0.4) | 68.5 (+0.3) | 49.5 (+0.4) | 87.4 (+0.4) | 18.3 (+0.1) | 61.2 (+0.0) | 39.6 (+0.8) | 85.0 (+0.3) |
| I-JEPA [3] | ViT-Base-16 | 34.0 | 68.6 | 52.6 | 88.3 | 17.9 | 63.4 | 36.2 | 86.3 |
| +MS | ViT-Base-16 | 39.6 (+5.6) | 72.6 (+4.0) | 59.3 (+6.7) | 89.7 (+1.4) | 22.4 (+4.5) | 66.9 (+3.5) | 38.6 (+2.4) | 87.4 (+1.1) |

Figure 7: Effect of DSE Regularization on iBOT [83] and I-JEPA [3].

In Tab. 3, we compare our model selection method with two baseline estimators: training loss and supervised downstream performance. Loss-based selection fails to track dense performance under the SDD phenomenon, and supervised selection is impractical due to high computational cost. In contrast, our DSE-based selection achieves competitive results with approximately $97.2\times$ speed-up, highlighting both efficiency and effectiveness.

**DSE Regularization Improves Dense Performance.** As shown in Fig. 7, DSE regularization consistently enhances model performance and the DSE metric. More importantly, it reverses the trend of dense degradation, demonstrating its fundamental capability to mitigate the SDD phenomenon. Results for additional methods and component ablation studies are provided in the Appendix F.9.

## 6 Conclusion

This paper identifies a widespread and detrimental phenomenon in self-supervised learning, termed Self-supervised Dense Degradation (SDD). Specifically, SDD occurs when dense task performance degrades over the course of training, causing the final checkpoint to exhibit a significant performance gap compared to the best intermediate checkpoint. To address SDD, we propose a Dense Representation Structure Estimator (DSE) that evaluates representation quality for downstream tasks by quantifying class separability and effective dimensionality. Our DSE is theoretically justified and empirically shown to correlate strongly with downstream task performance. To eliminate the harm of SDD, based on DSE, we introduce a checkpoint selection method for the off-the-shelf setting, and also optimize DSE directly as a regularizer. Experiments of sixteen SSL methods across four benchmark datasets confirm that these approaches effectively mitigate the negative effects of SDD.

## Acknowledgments

This work was supported in part by National Natural Science Foundation of China: 62525212, 62236008, 62441232, U21B2038, and U23B2051, in part by Youth Innovation Promotion Association CAS, in part by the Strategic Priority Research Program of the Chinese Academy of Sciences, Grant No. XDB0680201, in part by the China National Postdoctoral Program for Innovative Talents, Grant No. BX20250377.

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

## Appendix Contents

# A Proofs

## A.1 Proof of Proposition 1

**Proposition 7** (Restate of Prop. 1). *Let $\mathcal{Z} = \{z_1, z_2, \ldots, z_n\}$ be a set of points in $\mathbb{R}^d$, and let $\mathcal{C} = \{c_1, c_2, \ldots, c_k\}$ be the set of cluster centers obtained by the K-means algorithm. For each point $z \in \mathcal{Z}$, let $c(z) \in \mathcal{C}$ denote the cluster center to which $z$ is assigned by k-means. Then, for every $z \in \mathcal{Z}$ and for all $c_i \in \mathcal{C}$ with $c_i \neq c(z)$, the Euclidean distance satisfies*

$$\|z - c(z)\|_2 \leq \|z - c_i\|_2.$$

*As a result, the estimated error rate of the NN classifier is always 0:*

$$\widetilde{Err}_{\mathcal{D}}(f_{\boldsymbol{\theta}}) = \mathbb{E}_{\boldsymbol{x} \in \mathcal{D}}\left[\mathbb{P}_{\boldsymbol{z} \in f_{\boldsymbol{\theta}}(\boldsymbol{x})}\left[\|z - c(z)\| > \min_{c_i \neq c(z)} \|z - c_i\|\right]\right] \equiv 0.$$

*Proof.* By the definition of the K-means clustering algorithm, each point $z \in \mathcal{Z}$ is assigned to the cluster with the nearest center. Specifically, $c(z)$ is chosen to minimize the squared Euclidean distance to $z$ among all cluster centers in $\mathcal{C}$. Formally,

$$c(z) = \arg\min_{c_j \in \mathcal{C}} \|z - c_j\|_2^2.$$

Assume, for contradiction, that there exists a point $z \in \mathcal{Z}$ and a cluster center $c_i \in \mathcal{C}$ with $c_i \neq c(z)$ such that

$$\|z - c(z)\|_2 > \|z - c_i\|_2.$$

Squaring both sides (since the Euclidean distance is non-negative), we obtain

$$\|z - c(z)\|_2^2 > \|z - c_i\|_2^2.$$

However, this contradicts the definition of $c(z)$ as the cluster center that minimizes the squared distance to $z$. Therefore, our assumption must be false, and it must hold that

$$\|z - c(z)\|_2 \leq \|z - c_i\|_2$$

for all $c_i \in \mathcal{C}$ with $c_i \neq c(z)$. It yields that:

$$\mathbb{P}_{\boldsymbol{z} \in f_{\boldsymbol{\theta}}(\boldsymbol{x})}\left[\|z - c(z)\| > \min_{c_i \neq c(z)} \|z - c_i\|\right] \equiv 0.$$

This completes the proof. $\qquad\square$

## A.2 Proof of Theorem 2

**Proof Scratch.** We first present a proof scratch for better understanding. The idea is straightforward: by assuming representation in each class is $R$-sub-Gaussian, by the property of concentration, the representations lie in the main part of the distribution with probability of at least $1 - \delta$ for any $\delta > 0$. If the radius is smaller than the minimal inter-class distance, all the representations in the main part would be correctly classified, leading to a final error rate smaller than $1 - \delta$.

### A.2.1 Preliminaries

We first list some important properties used in the derivation.

**Lemma 8** (Sub-Gaussian Property). *A random variable $X$ is $R$-sub-Gaussian if its moment generating function satisfies:*

$$\mathbb{E}[\exp(\lambda X)] \leq \exp\left(\frac{\lambda^2 R^2}{2}\right), \quad \forall \lambda \in \mathbb{R}.$$

**Lemma 9** (Sub-Exponential Norm Bound). *If $X \in \mathbb{R}^d$ is an $R$-sub-Gaussian vector with independent coordinates, then $\|X\|^2 = \sum_{i=1}^d X_i^2$ is sub-exponential. Specifically, the sub-exponential norm $\|X^2\|_{\varphi_1}$ satisfies:*

$$\|X^2\|_{\varphi_1} \leq CR^2 d,$$

*where $C > 0$ is an absolute constant.*

*Proof.* Since each $X_i$ is $R$-sub-Gaussian, $X_i^2$ is sub-exponential with $\|X_i^2\|_{\varphi_1} \le CR^2$ (by [65], Prop 2.7.1). The sum of $d$ independent sub-exponential variables has a norm of at most $CR^2d$. □

**Lemma 10** (Chernoff Bound for Sub-Exponential Tails). *Let $Y = \|X\|^2$ where $X \in \mathbb{R}^d$ is an $R$-sub-Gaussian vector with independent coordinates. Then $Y$ is sub-exponential with $\|Y\|_{\psi_1} \le CR^2d$. For any $t > 0$:*

$$P\left(Y \ge \mathbb{E}[Y] + C_1 R^2 d(t + \sqrt{t})\right) \le e^{-t}.$$

*Proof.* From Lemma 9, $\|Y\|_{\psi_1} \le CR^2 d =: \alpha$. Using the sub-exponential tail bound:

$$P(Y - \mathbb{E}[Y] \ge t) \le \exp\left(-c\min\left(\frac{t^2}{\alpha^2}, \frac{t}{\alpha}\right)\right).$$

Set $t = \alpha(s + s^{1/2})$ for $s > 0$. Then:

$$\min\left(\frac{t^2}{\alpha^2}, \frac{t}{K}\right) = \min\left(s^2 + 2s^{3/2} + s, \; s + s^{1/2}\right) \ge s.$$

Thus $P(Y \ge \mathbb{E}[Y] + CR^2 d(s + \sqrt{s})) \le e^{-cs}$. Rename $s \to t/c$ for absolute constant $C_1 = C\max\{\frac{1}{c}, \frac{1}{\sqrt{c}}\}$ yields the proof. □

**Lemma 11** (Bernstein's Inequality [65]). *Let $Y_1, \ldots, Y_n$ be be independent sub-exponential random variables with $\|Y_i\|_{\varphi_1} \le K$. For any $t \ge 0$,*

$$P\left(\left|\sum_{i=1}^n (Y_i - \mathbb{E}[Y])\right| \ge t\right) \le 2\exp\left(-C_2 n \min\left(\frac{t^2}{K^2}, \frac{t}{K}\right)\right),$$

*where $C_2 > 0$ is an absolute constant.*

**Lemma 12** (Norm Concentration for Sub-Gaussian Variables [65]). *If $X \in \mathbb{R}^d$ is $R$-sub-Gaussian, then for all $t > 0$,*

$$P\left(\|X\| \ge C_3 R(\sqrt{d} + t)\right) \le \exp(-t^2).$$

*where $C_3 > 0$ is an absolute constant.*

### A.2.2 Important Lemmas for the Proof

Next, we proof some crucial lemmas for the proof.

**Lemma 13** (Trace Concentration for Sub-Gaussian Random Vectors). *Let $X \in \mathbb{R}^d$ be a mean-zero $R$-sub-Gaussian random variable; i.e., for all $\alpha \in \mathbb{R}^d$,*

$$\mathbb{E}\left[e^{\alpha^\top X}\right] \le \exp\left(\frac{R^2 \|\alpha\|^2}{2}\right).$$

*Let $\mu = \mathbb{E}[X]$ and $\Sigma = \text{Cov}(X)$. Suppose $\{X_i\}_{i=1}^N$ are i.i.d. copies of $X$, and define*

$$\widehat{\Sigma} = \frac{1}{N-1}\sum_{i=1}^N (X_i - \overline{X})(X_i - \overline{X})^\top, \quad \overline{X} = \frac{1}{N}\sum_{i=1}^N X_i.$$

*Then with probability at least $1 - \frac{\delta}{2}$,*

$$\left|\text{tr}(\widehat{\Sigma}) - \text{tr}(\Sigma)\right| \le \tilde{C} R^2 \left(d\sqrt{\frac{\log(8/\delta)}{N}} + \frac{d + \log(\frac{8}{\delta})}{N}\right).$$

*where $\tilde{C} > 0$ is a constant.*

*Proof.* We know

$$\text{tr}(\Sigma) = \mathbb{E}\left[\|X\|^2\right] - \|\mu\|^2.$$

$$\text{tr}(\widehat{\Sigma}) \;=\; \frac{1}{N-1} \sum_{i=1}^{N} \Big[ \|X_i\|^2 \Big] - \frac{N}{N-1} \|\overline{X}\|^2.$$

Rearrange the terms, we have

$$\text{tr}(\widehat{\Sigma}) - \text{tr}(\Sigma) \;=\; \frac{N}{N-1} \Big[ \underbrace{\Big( \tfrac{1}{N} \sum_{i=1}^{N} \|X_i\|^2 - \mathbb{E}[\|X\|^2] \Big)}_{\text{Term A}} - \underbrace{\big( \|\overline{X}\|^2 - \|\mu\|^2 \big)}_{\text{Term B}} \Big].$$

where the factor $\frac{N}{N-1} \approx 1$ only introduces a constant factor. Next, we bound *Term A* and *Term B* separately.

**1. Bounding Term A.** Define

$$\text{Term A} \;=\; \frac{1}{N} \sum_{i=1}^{N} \|X_i\|^2 \;-\; \mathbb{E}[\|X\|^2].$$

Since $X$ is $R$-sub-Gaussian, Lem. 9 imply that $\|X\|^2$ is sub-exponential with $\|X^2\|_{\varphi_1} \le CR^2$. Let $Y_i = X_i^2$, $K = CR^2$, applying Bernstein's inequality (Lem. 11) gives:

$$\mathbb{P}\Big( \big| \|X\|^2 - \mathbb{E}[\|X\|^2] \big| \ge t \Big) \;\le\; 2 \exp\Big( -C_2 N \min \Big( \frac{t^2}{(CR^2 d)^2}, \frac{t}{CR^2 d} \Big) \Big).$$

Next, we solve for $t$ to achieve the desired confidence $1 - \frac{\delta}{4}$. Set the right-hand side equal to $\frac{\delta}{4}$:

$$2 \exp\Big( -C_2 N \min \Big( \frac{t^2}{(CR^2 d)^2}, \frac{t}{CR^2 d} \Big) \Big) = \frac{\delta}{4}.$$

This equation has two regimes:

- **Small** $t$: $\frac{t^2}{(CR^2 d)^2} = \frac{1}{C_2 N} \log\big( \frac{8}{\delta} \big) \Rightarrow t = CR^2 d \sqrt{\frac{\log(8/\delta)}{C_2 N}}$.

- **Large** $t$: $\frac{t}{CR^2 d} = \frac{1}{C_2 N} \log\big( \frac{8}{\delta} \big) \Rightarrow t = \frac{CR^2 d \log(8/\delta)}{C_2 N}$.

To unify both regimes, we define

$$t = CR^2 d \sqrt{\frac{\log(8/\delta)}{C_2 N}} + \frac{CR^2 d \log(8/\delta)}{C_2 N}.$$

Thus, with probability at least $1 - \frac{\delta}{4}$, we conclude

$$\big| \text{Term A} \big| \;\le\; CR^2 d \sqrt{\frac{\log(8/\delta)}{C_2 N}} + \frac{CR^2 d \log(8/\delta)}{C_2 N}.$$

**2. Bounding Term B.** Since $X$ is zero-mean, we write

$$\text{Term B} \;=\; \|\overline{X}\|^2 - \|\mu\|^2 \;=\; \|\overline{X}\|^2.$$

Applying Lem. 12 and substituting $Y = \bar{X}$ and $R \to R/\sqrt{N}$:

$$P\Big( \|\bar{X}\| \ge C_3 \frac{R}{\sqrt{N}} (\sqrt{d} + t) \Big) \le \exp(-t^2).$$

Set $e^{-t^2} = \delta/4 \Rightarrow t = \sqrt{\log(4/\delta)}$, substituting $t$ into above inequality:

$$\|\bar{X}\| \le C_3 \frac{R}{\sqrt{N}} (\sqrt{d} + \sqrt{\log(4/\delta)}). \quad \text{with probability of at least } 1 - \frac{\delta}{4}.$$

Using $\sqrt{d} + \sqrt{\log(4/\delta)} \le \sqrt{2(d + \log(4/\delta))}$ (by Cauchy-Schwarz), we simplify:

$$\|\bar{X}\| \le C_3 R \sqrt{\frac{2(d + \log(4/\delta))}{N}}. \quad \text{with probability of at least } 1 - \frac{\delta}{4}.$$

Then immediately with probability at least $1 - \frac{\delta}{4}$.

$$\left|\text{Term B}\right| \le \|\bar{X}\|^2 \le 2C_3^2 R^2 \frac{d + \log(\frac{4}{\delta})}{N}.$$

**3. Combining the bounds.** With probability at least $1 - \frac{\delta}{2}$ (by a union bound), both Term A and Term B satisfy their respective bounds. Hence

$$\left|\text{tr}(\widehat{\Sigma}) - \text{tr}(\Sigma)\right| \le \frac{N}{N-1}\left(\left|\text{Term A}\right| + \left|\text{Term B}\right|\right)$$

$$\le 2\left[CR^2 d\sqrt{\frac{\log(8/\delta)}{C_2 N}} + \frac{CR^2 d\log(8/\delta)}{C_2 N} + 2C_3^2 R^2 \frac{d + \log(\frac{4}{\delta})}{N}.\right].$$

By selecting $\tilde{C} = 2\max\{\frac{C}{\sqrt{C_2}}, \frac{C}{C_2}, 2C_3^2\}$, we have

$$\left|\text{tr}(\widehat{\Sigma}) - \text{tr}(\Sigma)\right| \le \tilde{C} R^2 \left(d\sqrt{\frac{\log(8/\delta)}{N}} + \frac{d + \log(\frac{8}{\delta})}{N}\right).$$

Choosing $\delta$ small enough or letting $N$ grow large makes the $\frac{1}{N}$ term less significant, so the main deviation is typically on the order of

$$O\left(d\sqrt{\frac{\log(8/\delta)}{N}}\right).$$

This completes the proof. $\qquad\square$

**Lemma 14** (Sub-Gaussian Radius Bound). *Let $X \in \mathbb{R}^d$ be a mean-zero $R$-sub-Gaussian random vector with covariance matrix $\Sigma \in \mathbb{R}^{d \times d}$. For any $t > 0$, with probability at least $1 - \frac{\delta}{2}$,*

$$\|X\| \le \sqrt{\text{trace}(\Sigma) + C_1 R^2 d\left(\log(2/\delta) + \sqrt{\log(2/\delta)}\right)}.$$

*Proof.* Since $X$ is $R$-sub-Gaussian, by Lem. 10, denote $Y = X^2$ and $\mathbb{E}[Y] = \mu$, for any $t > 0$:

$$P\left(Y \ge \mu + C_1 R^2 d(t + \sqrt{t})\right) \le e^{-t}.$$

By identifying $\mu = \text{trace}(\Sigma)$, we arrive at

$$p\left(\|X\|^2 \ge \text{trace}(\Sigma) + C_1 R^2 d(t + \sqrt{t})\right) \le e^{-t}.$$

Putting $t = \log\left(\frac{2}{\delta}\right)$ yields

$$p\left(\|X\| \le \sqrt{\text{trace}(\Sigma) + C_1 R^2 d\left(\log(2/\delta) + \sqrt{\log(2/\delta)}\right)}\right) \ge 1 - \frac{\delta}{2}.$$

This completes the proof. $\qquad\square$

**Lemma 15** (Bounded Radius of Sub-Gaussian Variables). *Let $Z \in \mathbb{R}^d$ be an $R$-sub-Gaussian random variable, and $\{z_i\}_{i=1}^N$ are i.i.d copies of $Z$. Denote $\bar{z} = \frac{1}{N}\sum_{i=1}^N z_i$ and*

$$Z_c = [z_1 - \bar{z}, \quad z_2 - \bar{z}, \quad \cdots, \quad z_N - \bar{z}]$$

*as the centered embedding matrix. Its singular values are $\sigma_1, \sigma_2, \ldots, \sigma_d \ge 0$. For any $0 < \delta < 1$, with probability at least $1 - \delta$:*

$$\|Z - \mathbb{E}[Z]\| \le \frac{\sum_{i=1}^d \sigma_i(Z_c)}{\sqrt{(N-1)}}$$

$$+ \sqrt{C_1 R^2 d\left(\log(2/\delta) + \sqrt{\log(2/\delta)}\right) + \tilde{C} R^2 \left(d\sqrt{\frac{\log(8/\delta)}{N}} + \frac{d + \log(\frac{8}{\delta})}{N}\right)}.$$

*Proof.* By Lem. 14, we have:

$$p\left(\|Z - \mathbb{E}[Z]\| \sqrt{\text{trace}(\Sigma) + C_1 R^2 d \left(\log(2/\delta) + \sqrt{\log(2/\delta)}\right)}\right) \geq 1 - \frac{\delta}{2}.$$

Denote $\widehat{\Sigma} = \frac{1}{N-1} Z_c^T Z_c$ as the centered representation matrix. By Lem. 13, with probability of at least $1 - \delta$ (with the union bound), we have:

$$\|Z - \mathbb{E}[Z]\| \leq \sqrt{\text{trace}(\widehat{\Sigma}) + C_1 R^2 d \left(\log(2/\delta) + \sqrt{\log(2/\delta)}\right) + \tilde{C} R^2 \left(d\sqrt{\frac{\log(8/\delta)}{N}} + \frac{d + \log(\frac{8}{\delta})}{N}\right)}.$$

With $\sqrt{a + b} \leq \sqrt{a} + \sqrt{b}$, rearrange the terms:

$$\|Z - \mathbb{E}[Z]\| \leq \sqrt{\text{trace}(\widehat{\Sigma})}$$
$$+ \sqrt{C_1 R^2 d \left(\log(2/\delta) + \sqrt{\log(2/\delta)}\right) + \tilde{C} R^2 \left(d\sqrt{\frac{\log(8/\delta)}{N}} + \frac{d + \log(\frac{8}{\delta})}{N}\right)}.$$

Since

$$\text{trace}(\widehat{\Sigma}) = \frac{1}{N-1} \sum_{i=1}^{d} \sigma_i(Z_c)^2 \leq \frac{1}{N-1} \left(\sum_{i=1}^{d} \sigma_i(Z_c)\right)^2.$$

Thus, with probability at least $1 - \delta$, we have:

$$\|Z - \mathbb{E}[Z]\| \leq \frac{\sum_{i=1}^{d} \sigma_i(Z_c)}{\sqrt{(N-1)}}$$
$$+ \sqrt{C_1 R^2 d \left(\log(2/\delta) + \sqrt{\log(2/\delta)}\right) + \tilde{C} R^2 \left(d\sqrt{\frac{\log(8/\delta)}{N}} + \frac{d + \log(\frac{8}{\delta})}{N}\right)}.$$

This completes the proof.

$\square$

### A.2.3   Proof of the Main Theorem

**Theorem 16** (Formal version of Thm 2). *Let $Z^j = \{Z : y(Z) = j\}$ be the examples in $j$-th class with $|Z^j| = N_j$. Assume that for all $j \in [K]$, $Z^j \in \mathbb{R}^d$ is an $R$-sub-Gaussian random variable, and $\{z_i^j\}_{i=1}^{N_j}$ are i.i.d copies of $Z^j$. Denote $\bar{z}^j = \frac{1}{N_j} \sum_{i=1}^{N_j} z_i^j$ and $Z_c^j = \left[z_1^j - \bar{z}^j, \ z_2^j - \bar{z}^j, \ \cdots, \ z_N^j - \bar{z}^j\right]$ as the centered embedding matrix for $Z^j$. Then, for any $\delta > 0$:*

$$Err_{\mathcal{D}}(f_{\boldsymbol{\theta}}) \leq \delta + \tilde{P}(\delta).$$

*With $\sigma_i(\cdot)$ represents the $i$-th singular value*

$$\tilde{P}(\delta) = \mathbb{P}_{\boldsymbol{z}}\left(\frac{\sum_{i=1}^{d} \sigma_i(Z_c^{y(\boldsymbol{z})})}{\sqrt{(N_{y(\boldsymbol{z})} - 1)}} + C_\delta^{y(\boldsymbol{z})} > \min_{k \in [K] \setminus y(\boldsymbol{z})} \|\boldsymbol{z} - \boldsymbol{\mu}_k\|\right).$$

*Here,*

$$C_\delta^{y(\boldsymbol{z})} = \sqrt{C_1 R^2 d \left(\log(2/\delta) + \sqrt{\log(2/\delta)}\right) + \tilde{C} R^2 \left(d\sqrt{\frac{\log(8/\delta)}{N_{y(\boldsymbol{z})}}} + \frac{d + \log(\frac{8}{\delta})}{N_{y(\boldsymbol{z})}}\right)}$$

*is a positive class-irrelevant bias term and $C_1, \tilde{C}$ are positive constants.*

*Proof.* Using a NN classifier, a representation $\boldsymbol{z}$ could be correctly classified if:

$$||\boldsymbol{z} - \boldsymbol{\mu}_{y(\boldsymbol{z})}|| \leq ||\boldsymbol{z} - \boldsymbol{\mu}_k||$$

holds for all $k \in [K]\backslash y(\boldsymbol{z})$.

Using the result of Lem. 14, for any class $j$ and $\delta > 0$, the expected distance of the intra-class distance is bounded with probability at least $1 - \delta$:

$$\mathbb{E}_{\boldsymbol{z}:y(\boldsymbol{z})=j}[||\boldsymbol{z} - \boldsymbol{\mu}_{y(\boldsymbol{z})}||] \leq \frac{\sum_{i=1}^d \sigma_i^j(Z_c)}{\sqrt{N_j - 1}}$$
$$+ \sqrt{C_1 R^2 d\left(\log(2/\delta) + \sqrt{\log(2/\delta)}\right) + \tilde{C}R^2\left(d\sqrt{\frac{\log(8/\delta)}{N_j}} + \frac{d + \log(\frac{8}{\delta})}{N_j}\right)}.$$

For the sake of simplicity, we define

$$C_\delta^j = \sqrt{C_1 R^2 d\left(\log(2/\delta) + \sqrt{\log(2/\delta)}\right) + \tilde{C}\,R^2\left(d\sqrt{\frac{\log(8/\delta)}{N_j}} + \frac{d + \log(\frac{8}{\delta})}{N_j}\right)}.$$

For the representations in the $j$-th class, we separate it into two parts: the main part $Z_m^j$ in which all examples lie in the radius, and the outside part $Z_o^j = Z^j\backslash Z_m^j$. From the concentration property, we have $Z^j = Z_m^j \cup Z_o^j$ and $P_{\boldsymbol{z}\in Z^j}(\boldsymbol{z} \in Z_m^j) \geq 1 - \delta$.

For the main part, the accuracy of the NN classifier on the $j$-th class could be calculated by:

$$P_{\boldsymbol{z}\in Z_m^j}\left(\frac{\sum_{i=1}^d \sigma_i(Z_c^j)}{\sqrt{(N_j - 1)}} + C_\delta^j \leq \min_{k\in[K]\backslash j} ||\boldsymbol{z} - \boldsymbol{\mu}_k||\right).$$

Thus, the error rate of the $j$-th class should be at least (assuming all the representations in the outside part are misclassified):

$$\text{Err}_{\mathcal{D}}^j(f_{\boldsymbol{\theta}}) \leq \delta + P_{\boldsymbol{z}\in Z^j}\left(\frac{\sum_{i=1}^d \sigma_i(Z_c^j)}{\sqrt{(N_j - 1)}} + C_\delta^j > \min_{k\in[K]\backslash j} ||\boldsymbol{z} - \boldsymbol{\mu}_k||\right).$$

Rearrange the terms and take an expectation on $j$ yields the result. $\qquad\square$

## A.3 Proof of Corollary 5

We first introduce some useful lemmas for the proof.

**Lemma 17** (Sub-Gaussian Norm Concentration [65]). *For $R$-sub-Gaussian vectors $X \in \mathbb{R}^d$:*

$$\mathbb{P}\left(||X|| \geq R\sqrt{d} + t\right) \leq 2\exp\left(-\frac{t^2}{2R^2}\right).$$

**Lemma 18** (Intra-class Concentration (Adapted from [66])). *For any $R$-sub-Gaussian class $j$ with $N_j$ samples:*

$$\frac{1}{\sqrt{N_j - 1}}\sum_{i=1}^d \sigma_i(Z_c^j) \leq R\sqrt{d}\left(1 + \sqrt{\frac{\log(8/\delta)}{N_j}} + \frac{\log(8/\delta)}{N_j}\right)$$

*holds with probability $\geq 1 - \delta/4$.*

**Lemma 19** (Dimensional Scaling of the Concentration Term). *The concentration term $C_\delta^j$ admits the dimensional scaling:*

$$C_\delta^j \leq R\sqrt{\underbrace{\sqrt{Cd\log(2/\delta)}}_{\text{Sub-Gaussian term}} + \underbrace{C'd}_{\text{Covariance term}}} + \mathcal{O}\left(R\sqrt{\frac{d}{N_j}}\right).$$

*For $d \geq \log(8/\delta)$, this simplifies to $C_\delta^j \leq \sqrt{3}R\sqrt{d}$ with probability $\geq 1 - \delta/4$.*

**Corollary 20** (Formal Version of Corollary 5). *Under Thm 2's assumptions with the condition* $\min_{k \neq y(z)} \|\boldsymbol{\mu}_{y(z)} - \boldsymbol{\mu}_k\| > \sqrt{d}R\left(2 + \sqrt{\frac{\log(8/\delta)}{N_j}} + \sqrt{3}\right)$, *for any* $\delta > 0$:

$$Err_{\mathcal{D}}(f_{\boldsymbol{\theta}}) \leq \delta + 2K \exp\left(-\tilde{C}_\delta \cdot d\right),$$

*where* $\tilde{C}_\delta = \frac{\sqrt{\frac{\log(8/\delta)}{N_j}} + \sqrt{3}}{2} > 0$ *is a constant.*

*Proof.* By the triangle inequality

$$\|\boldsymbol{z} - \boldsymbol{\mu}_k\| \geq \|\boldsymbol{\mu}_{y(z)} - \boldsymbol{\mu}_k\| - \|\boldsymbol{z} - \boldsymbol{\mu}_{y(z)}\|.$$

$\boldsymbol{z}$ would be correctly classified when:

$$\|\boldsymbol{z} - \boldsymbol{\mu}_{y(z)}\| \leq \min_{k \neq j} \|\boldsymbol{\mu}_{y(z)} - \boldsymbol{\mu}_k\| - \frac{\sum_{i=1}^d \sigma_i(Z_c^{y(z)})}{\sqrt{(N_{y(z)} - 1)}} - C_\delta^{y(z)}.$$

Using Lemma 18 and Lemma 19, and denote $\Delta = \min_{k \neq y(z)} \|\boldsymbol{\mu}_{y(z)} - \boldsymbol{\mu}_k\|/\sqrt{d}$, we have:

$$\|\boldsymbol{z} - \boldsymbol{\mu}_{y(z)}\| \leq \sqrt{d}\left(\Delta - R\left(1 + \sqrt{\frac{\log(8/\delta)}{N_j}} + \sqrt{3}\right)\right).$$

For the sake of simplicity, denote $\tilde{R} = R\left(1 + \sqrt{\frac{\log(8/\delta)}{N_j}} + \sqrt{3}\right)$. By the concentration of sub-Gaussian norm (17), we know:

$$\mathbb{P}\left(\|\boldsymbol{z} - \boldsymbol{\mu}_{y(z)}\| \geq R\sqrt{d} + t\right) \leq 2\exp\left(-\frac{t^2}{2R^2}\right).$$

Selecting $t = (\Delta - \tilde{R} - R)\sqrt{d}$ gives:

$$\mathbb{P}\left(\|\boldsymbol{z} - \boldsymbol{\mu}_{y(z)}\| \geq \sqrt{d}(\Delta - \tilde{R})\right) \leq 2\exp\left(-d\frac{\sqrt{\frac{\log(8/\delta)}{N_j}} + \sqrt{3}}{2}\right).$$

Denote $\tilde{C}_\delta = \frac{\sqrt{\frac{\log(8/\delta)}{N_j}} + \sqrt{3}}{2}$ and apply the union bound across all classes yields:

$$\tilde{P}(\delta) \leq 2K \exp\left(-\tilde{C}_\delta \cdot d\right).$$

Immediately,

$$Err_{\mathcal{D}}(f_{\boldsymbol{\theta}}) \leq \delta + 2K \exp\left(-\tilde{C}_\delta \cdot d\right),$$

which completes the proof. $\qquad\square$

### A.4 Analysis of the Effect of k

In the previous analysis, we assumed the pseudo-label to be accurate for simplicity. In this subsection, we analyze the error introduced by $k \neq C$. From the original error decomposition:

$$\text{Err}(k) \leq \delta + \tilde{P}_C(\delta) = \delta + \tilde{P}_k(\delta) + \underbrace{(\tilde{P}_C(\delta) - \tilde{P}_k(\delta))}_{\Delta(k)}.$$

It is easy to tell that $\text{Err}(k) \leq \delta + \tilde{P}_k(\delta) \implies \text{Err}(k) \leq \delta + \tilde{P}_C(\delta)$ when $\Delta(k) \geq 0$. In the next theorem, we model the relationship between $\Delta(k)$ and $k$.

**Theorem 21** (Error Bound with Clustering Deviation). *For any* $\delta > 0$, *the error bound satisfies:*

$$Err \leq \delta + \tilde{P}_k(\delta) + \Delta(k)$$

*where* $\Delta(k) := \tilde{P}_C(\delta) - \tilde{P}_k(\delta)$ *exhibits the following properties:*

- $\Delta(k) \geq 0$ *when $k > C$.*

- *As $k$ decreases from $C$ to 1, $\Delta(k)$ first increases to a positive peak, then decreases to negative values.*

*Proof.* Intuitively, the proof starts by $k = C$ and discusses the change of $\Delta(k)$ with two cases: 1) Increasing k leads to the splitting of original clusters, and 2) Decreasing k merges two adjacent clusters.

**Case 1:** $k > C$ (**Over-clustering**) When $k > C$, since over clustering converts some false positives into true positives without affecting negative predictions, $\Delta(k) \geq 0$ naturally holds.

**Case 2:** $k < C$ (**Under-clustering**) Denote the class center and radius as $\boldsymbol{\mu}_j := \mathbb{E}[\boldsymbol{z} | \boldsymbol{z} \in \mathcal{C}_j]$, $\mathcal{R}_j := \sup_{\boldsymbol{z} \in \mathcal{C}_j} \|\boldsymbol{z} - \boldsymbol{\mu}_j\|$, respectively. The distance between class centers is then defined as $d_{ij} := \|\boldsymbol{\mu}_i - \boldsymbol{\mu}_j\|$. When merging $\mathcal{C}_1$ and $\mathcal{C}_2$ into $\mathcal{C}^*$, define the merged center and radius as $\boldsymbol{\mu}_*$ and $\mathcal{R}_*$, respectively.

The impact of merging depends on the separation between classes. Let $\mathcal{C}_1$ and $\mathcal{C}_2$ be two classes with separation ratio $\rho = \frac{d_{12}}{\mathcal{R}_1 + \mathcal{R}_2}$. We distinguish two regimes:

- Overlapping Merging ($\rho \ll 1$): Classes are poorly separated, with $d_{12}$ small relative to their radius.

- Separated Merging ($\rho \geq 1$): Classes are distinct, with $d_{12}$ comparable to or larger than radius.

When merging two overlapping classes ($\rho \ll 1$), define the center and radius of the merged cluster $\mathcal{C}^* = \mathcal{C}_1 \cup \mathcal{C}_2$ as: $\boldsymbol{\mu}_* = \frac{N_1 \boldsymbol{\mu}_1 + N_2 \boldsymbol{\mu}_2}{N_1 + N_2}$ and $\mathcal{R}_* \leq \max(\mathcal{R}_1, \mathcal{R}_2) + \frac{\min(N_1, N_2)}{N_1 + N_2} d_{12}$, respectively. For $\boldsymbol{z} \notin \mathcal{C}^*$, the minimal distance to other classes improves due to the merged center's shift:

$$D_{\min}^{\boldsymbol{z}}(k) \geq D_{\min}^{\boldsymbol{z}}(C) + \underbrace{\|\boldsymbol{\mu}_* - \boldsymbol{\mu}_{\text{proj}}\|}_{\text{Gain from center shift}},$$

where $\boldsymbol{\mu}_{\text{proj}}$ is the nearest original center. This increases the margin $D_{\min}^{\boldsymbol{z}}(k) - \mathcal{R}_*$ for non-merged classes, reducing $\tilde{P}_k(\delta)$.

For $\boldsymbol{z} \in \mathcal{C}^*$, the radius $\mathcal{R}_*$ remains comparable to original radius since $d_{12}$ is small. The dominant effect is the elimination of misclassification between $\mathcal{C}_1$ and $\mathcal{C}_2$. Thus, $\tilde{P}_k(\delta) < \tilde{P}_C(\delta)$, resulting in $\Delta(k) > 0$.

When merging well-separated classes ($\rho \geq 1$), the merged radius $\mathcal{R}_* = \max(\mathcal{R}_1, \mathcal{R}_2) + d_{12}$ becomes significantly larger. For $\boldsymbol{z} \in \mathcal{C}^*$:

$$D_{\min}^{\boldsymbol{z}}(k) - \mathcal{R}_* \leq \|\boldsymbol{z} - \boldsymbol{\mu}_*\| - \mathcal{R}_* \leq \mathcal{R}_1 + \frac{N_2}{N_1 + N_2} d_{12} - d_{12} \ll 0,$$

increasing $\tilde{P}_k(\delta)$. For $\boldsymbol{z} \notin \mathcal{C}^*$, the minimal distance may decrease slightly, but the dominant effect is the inflated $\mathcal{R}_*$, leading to $\tilde{P}_k(\delta) > \tilde{P}_C(\delta)$ and $\Delta(k) < 0$.

From above analysis, we see that as $k$ decreases from $C$ to 1, the trajectory of $\Delta(k)$ follows:

- **Initial Mergers:** Overlapping classes are merged first (since $k$-means prioritizes reducing within-cluster variance). This reduces $\tilde{P}_k(\delta)$, causing $\Delta(k) > 0$.

- **Late Merges:** Remaining classes are better separated, and merging them inflates $\mathcal{R}_*$ significantly. $\tilde{P}_k(\delta)$ increases to 1, making $\Delta(k) < 0$.

$\square$

Since $\Delta(k)$ could possibly be smaller than 0 when $k < C$, in practice, we suggest taking $k$ equal to or slightly larger than the real number of clusters to ensure the tightest bound.

# B  Additional Related Works

**Self-supervised Learning Approaches Beyond Images.** In recent years, self-supervised learning has been applied to multiple modalities, including video [46, 45, 63], point clouds [81, 55], time-series data [79, 80], and cell images [18, 54]. While SSL has achieved strong results in these areas, examining whether similar fine-grained performance degradation occurs during training is a valuable direction for future work.

**Studies on Class Separability.** In this work, our theoretical analysis uncovers the connection between class separability and downstream performance. We clarify that this connection is not unique to our study; it has been widely explored in the machine learning community [22, 6, 23]. In the context of self-supervised learning, several studies have examined this relationship through the lens of alignment and uniformity in contrastive learning [67], or through coding rate reduction [77].

Although we do not introduce the concept of class separability, our work makes non-trivial contributions by bridging the downstream performance with measurable factors. Furthermore, we propose practical methods for evaluating the quality of dense representations.

**Supervised Transferability Estimation.** Due to the high computational cost of transfer learning, a large number of studies have emerged to estimate downstream performance without fine-tuning [33, 35, 76, 20, 19, 7, 1, 56, 42]. Early approaches focus on approximating the posterior distribution of target datasets [64, 48, 43]. Many works also leverage the energy score [25] or the class separability [75, 51] as the metric. While effective in supervised settings, these methods require labeled data, limiting their applicability to self-supervised scenarios.

**Concurrent Studies on SSL Degradation.** We also acknowledge a concurrent study, DINO v3 [57], which investigates degradation in self-supervised learning. In this paper, we show that the SDD phenomenon is widespread across sixteen state-of-the-art methods, datasets, and tasks, and we propose a theoretically grounded metric for performance estimation, model selection, and regularization. In contrast, DINO v3 focuses on degradation within the iBOT/DINO v2 family and introduces gram-matrix distillation to address it. The findings in DINO v3 strongly support the scalability of the SDD phenomenon. Their gram-matrix loss selects an early model with hand-crafted iteration steps as the teacher for correlation distillation, which could be improved by integrating our DSE-based model selection. Thus, the techniques in DINO v3 and those in this paper are likely complementary, and DINO v3 provides valuable scaling evidence for the SDD phenomenon that we do not include here due to resource limitations.

We also note that Wen et al. [71] study performance degradation during training from the perspective of insufficient semantic concentration. Their semantic concentration framework effectively mitigates the degradation by improving intra-class compactness.

# C Detailed Methodology for DSE-regularized Online Optimization

As mentioned earlier, the DSE metric provides a lower bound on dense performance. Since all operations involved in computing DSE are differentiable, directly optimizing DSE can potentially address the SDD problem effectively. In practice, we include DSE explicitly as a regularizer in the training process:

$$\mathcal{L} = \mathcal{L}_{original} - \beta \cdot \text{DSE}.$$

Although calculating DSE itself is model-agnostic, integrating it into the training procedure requires certain model-specific adjustments. Specifically, all baseline methods in this study use a Joint-Embedding Self-Supervised Learning (JE-SSL) framework, which employs a siamese network with two global views of size $224 \times 224$ during pretraining. Thus, we use the student model (the encoder) to extract dense representations from these global views, and then compute the DSE based on these representations. For approaches involving masked modeling, we additionally feed the unmasked views through the student model to obtain dense representations.

To illustrate how DSE can be integrated into the training process, we take the training procedure of DINO [10] as an example. We provide the code for the DSE regularizer class in the Supplementary Material. This allows DSE regularization to be easily integrated into any JE-SSL framework by extracting dense representations from the student model and including the DSE regularizer in the loss function.

---

**Algorithm 2** An example PyTorch pseudocode of DINO with DSE-regularized training.

```
# fs, ft: student and teacher encoder
# gs, gt: student and teacher heads
# C: center (K)
# tps, tpt: student and teacher temperatures
# l, m: network and center momentum rates
# DSE: DSE Estimator
# a: weight of DSE regularization loss

ft.params = fs.params
gt.params = gs.params
for x in loader: # load a minibatch x with n samples
    x1, x2 = augment(x), augment(x) # random views
    z1, z2 = fs(x1), fs(x2) # student output n-by-(p+1)-by-d
    z1_cls, z1_patch = z1[:,0], z1[:,1:]
    z2_cls, z2_patch = z2[:,0], z2[:,1:] # extract cls and patch tokens

    z_patch = Concat(z1_patch, z2_patch)

    s1, s2 = gs(z1_cls), gs(z2_cls) # student output n-by-K
    t1, t2 = gt(ft(x1)), gt(ft(x2)) # teacher output n-by-K

    dse = - DSE(z_patch)
    loss = H(t1, s2)/2 + H(t2, s1)/2 + a * dse
    loss.backward() # back-propagate

    # student, teacher and center updates
    update(gs) # SGD
    gt.params = l*gt.params + (1-l)*gs.params
    C = m*C + (1-m)*cat([t1, t2]).mean(dim=0)

def H(t, s):
    t = t.detach() # stop gradient
    s = softmax(s / tps, dim=1)
    t = softmax((t - C) / tpt, dim=1) # center + sharpen
    return - (t * log(s)).sum(dim=1).mean()
```

---

# D Detailed Settings

## D.1 Pretraining

For pretraining, we reimplement all methods based on their original settings, but we disable automatic mixed precision (AMP) in I-JEPA [3] to prevent training instability. All models are trained for 800 epochs on ImageNet-1k [39], except for SwAV [9], VICReg [4], VICRegL [5], and DenseCL [70]. We observe model collapse when training SwAV for more epochs, and take the default settings for DenseCL, VICReg and VICRegL. The full pretraining hyperparameters are listed in Tab. 4.

Table 4: The hyperparameters used for pretraining.

| Method | Architecture | Learning Rate | Optimizer | Warm-up Epochs | Epochs | Batch size | Image size |
|---|---|---|---|---|---|---|---|
| MoCo v3 [16] | ViT-Small-16 | 1.5e-4 | AdamW | 40 | 800 | 4096 | $2 \times 224^2$ |
| DenseCL [70] | ResNet-50 | 0.03 | SGD | 0 | 200 | 256 | $2 \times 224^2$ |
| BYOL [26] | ResNet-50 | 0.2 | LARS | 10 | 800 | 1024 | $2 \times 224^2$ |
| SimSiam [14] | ResNet-50 | 0.5 | SGD | 10 | 800 | 512 | $2 \times 224^2$ |
| EsViT [41] | Swin-Tiny-7 | 5e-4 | AdamW | 5 | 800 | 1024 | $2 \times 224^2$ |
| MEC [44] | ResNet-50 | 0.5 | SGD | 10 | 800 | 512 | $2 \times 224^2$ |
| Barlow Twins [78] | ResNet-50 | 0.2 | LARS | 10 | 800 | 2048 | $2 \times 224^2$ |
| VICReg [4] | ResNet-50 | 0.2 | LARS | 10 | 1000 | 2048 | $2 \times 224^2$ |
| VICRegL [5] | ResNet-50 | 0.2 | LARS | 10 | 300 | 2048 | $2 \times 224^2$ |
| SwAV [9] | ResNet-50 | 0.6 | SGD | 0 | 400 | 512 | $2 \times 224^2 + 6 \times 96^2$ |
| DINO [10] | ViT-Small-16 | 5e-4 | AdamW | 10 | 800 | 1024 | $2 \times 224^2 + 10 \times 96^2$ |
| iBOT [83] | ViT-Small-16 | 5e-4 | AdamW | 10 | 800 | 1024 | $2 \times 224^2 + 10 \times 96^2$ |
| Mugs [84] | ViT-Small-16 | 8e-4 | AdamW | 10 | 800 | 1024 | $2 \times 224^2 + 10 \times 96^2$ |
| ReSA [72] | ResNet-50 | 0.5 | SGD | 2 | 800 | 1024 | $2 \times 224^2$ |
| MAE [29] | ViT-Small-16 | 1.5e-4 | AdamW | 40 | 800 | 4096 | $224^2$ |
| I-JEPA [3] | ViT-Base-16 | 1e-3 | AdamW | 15 | 800 | 2048 | $224^2$ |

For readers' convenience, we provide a brief introduction to these methods as follows:

- **MoCo v3** [16]: Improves contrastive learning stability and performance by combining MoCo [28] with ViTs.

- **DenseCL** [70]: Enhances self-supervised learning by applying contrastive loss [49] at the dense feature level, enabling better dense-level representation learning.

- **BYOL**[2] [26]: Explores non-contrastive SSL with an asymmetric design including an extra prediction head and a stop-gradient mechanism to avoid collapse.

- **SimSiam** [14]: Proposes a simple siamese network for self-supervised learning without negative samples, relying on stop-gradient and predictor mechanisms to avoid collapsing solutions.

- **EsViT** [41]: Enhances self-supervised ViT training by combining masked image modeling with contrastive learning for improved efficiency and scalability.

- **MEC** [44]: Introduces the idea of maximum entropy encoding that explicitly optimizes on the structure of the representations, leading to generalizable representations.

- **Barlow Twins** [78]: Uses an invariance term and a redundancy-reduction term to optimize representation structure, effectively preventing collapse.

- **VICReg** [4]: Builds on Barlow Twins by adding a variance regularization loss to keep the representation space well spread.

- **VICRegL** [5]: Extends VICReg to local features, learning strong dense representations while maintaining image-level quality; it also shows a trade-off between coarse and fine-grained performance.

- **SwAV** [9]: Combines online clustering with SSL, using swapped prediction between cluster assignments to learn meaningful representations without pairwise comparisons. Assigning representations to clusters naturally enhances the class separability.

- **DINO** [10]: Leverages self-distillation with ViTs and a teacher-student framework to achieve strong instance-level representations.

- **iBOT** [83]: Integrates masked image modeling with self-distillation in features space of ViTs, enabling joint learning of global and local visual representations.

---

[2] For BYOL, we use the pytorch implementation from https://github.com/sthalles/PyTorch-BYOL

- **Mugs** [84]: Proposes a multi-granularity discriminative framework that performs well on both instance-level and dense prediction tasks.
- **ReSA** [72]: Improves clustering-based SSL by learning cluster assignments directly from encoder features, removing the need for prototypes.
- **MAE** [29]: Introduces a simple and scalable masked autoencoder framework for self-supervised learning, reconstructing masked patches from visible ones.
- **I-JEPA** [3]: Proposes a joint-embedding predictive architecture for self-supervised learning, focusing on predicting representations of masked regions in an abstract latent space.

## D.2 Evaluation

**Linear Semantic Segmentation.** To assess dense representation quality, we adopt the standard linear evaluation protocol standard in SSL [50, 86, 83, 84]. We evaluate four benchmarks: COCO-Stuff27 [8], PASCAL VOC [21], ADE20k [82], and Cityscapes [17]. We remove projectors, and train only a lightweight classifier on dense features. For all models, we use the last layer's patch embeddings. Input images are resized to $336 \times 336$ (except $896 \times 896$ for Cityscapes to preserve detail) and the classification head is trained on 100,000 images. The head is optimized using a batch size of 256 (64 for Cityscapes due to GPU memory limitation) and a learning rate of $0.01 \times \sqrt{\text{batch size}/256}$ using an Adam [38] optimizer.

In the linear probing setup, the backbone is kept fixed, and only a small classifier is trained using the dense features. In the linear transfer learning setup, the backbone is fine-tuned with a lower learning rate.

**Semi-supervised Video Object Segmentation.** We evaluate the semi-supervised video object segmentation on the DAVIS 2017 dataset [52] following the [10]. During testing, the labels of the first frame are given, and we propagate predictions to subsequent frames via $k$-NN similarity matching of dense features.

$k$**-NN Image Classification. (for the motivation)** Following the standard $k$-NN setting in DINO [10], we apply $k$-NN classification using class tokens (or averaged patch tokens if no class token exists) mapped to ImageNet-1k labels.

## D.3 Kendall's $\tau$ Coefficient

We measure the correlation between our metric and downstream performance using Kendall's $\tau$ coefficient, which is a standard metric for transferability analysis [76, 25, 75]:

$$\tau = \frac{2}{N(N-1)} \sum_{1 \leq i < j \leq N} \text{sign}(M_i - M_j)\text{sign}(P_i - P_j).$$

Here, $N$ is the number of checkpoints, $M_i, P_i$ denote the value of metrics and downstream performance calculated using $i$-th checkpoint, respectively. When $\tau = 1$, the proposed metric is perfectly aligned with the downstream performance, and $\tau = -1$ represents that the proposed metric is inversely correlated with the downstream performance.

## D.4 Details in DSE Calculation

For metric calculation, we use 2048 randomly sampled images from the pertaining dataset (ImageNet-1k), and no extra data is introduced for performance estimation. As standard testing protocol, the images are first resized to $256 \times 256$, and then center cropped into $224 \times 224$ and normalized. For pseudo-label generation, the $k$-means clustering is applied. To improve robustness, we compute $M_{intra}$ by averaging the results obtained by $B = 1$ and $B = 8$. We set $k = 3$ and $k = 24$ for $B = 1$ and $B = 8$, respectively. Sensitivity analyses of these parameters are discussed in later sections.

## D.5 Computational Resources

All pretraining are conducted on 8 NVIDIA A100 GPUs, the evaluation is done on 8 NVIDIA 4090 GPUs, and the DSE metric is computed on a single NVIDIA 4090 GPU. The results in Tab. 3 are obtained by averaging the time cost of 5 runs on the 4090 GPU.

# E  Additional Experiment Results of the SDD Phenomenon

## E.1  The SDD Phenomenon Exists Across Datasets

To comprehensively analyze the SDD phenomenon, we visualize training curves for the state-of-the-art methods on the PASCAL VOC, COCO-Stuff, ADE20k, and Cityscapes datasets in Fig. 8, Fig. 9, Fig. 10, and Fig. 11, respectively. The SDD phenomenon persists consistently across all datasets, with training curves exhibiting similar degradation patterns. These results confirm that SDD is dataset-agnostic and closely linked to dense representation quality.

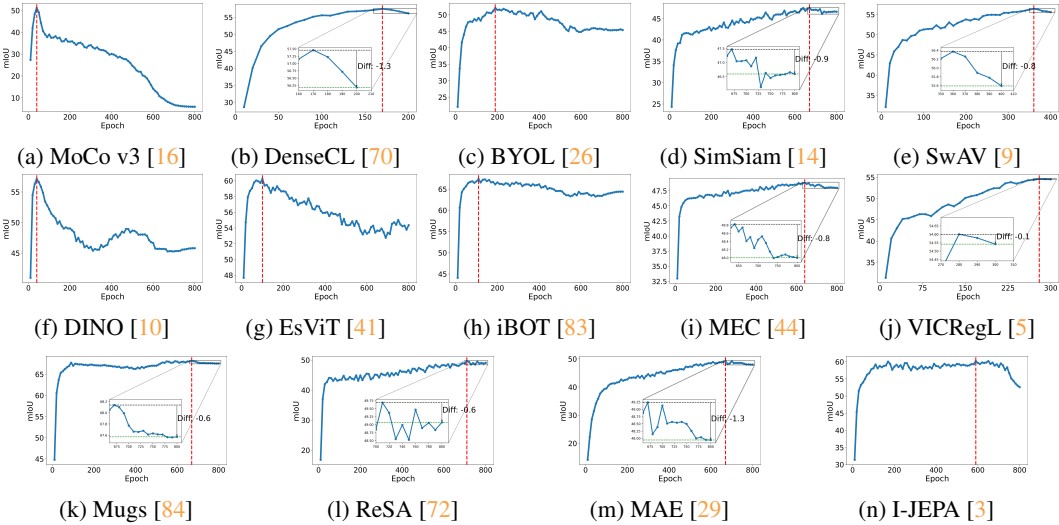

Figure 8: The SDD phenomenon on PASCAL VOC.

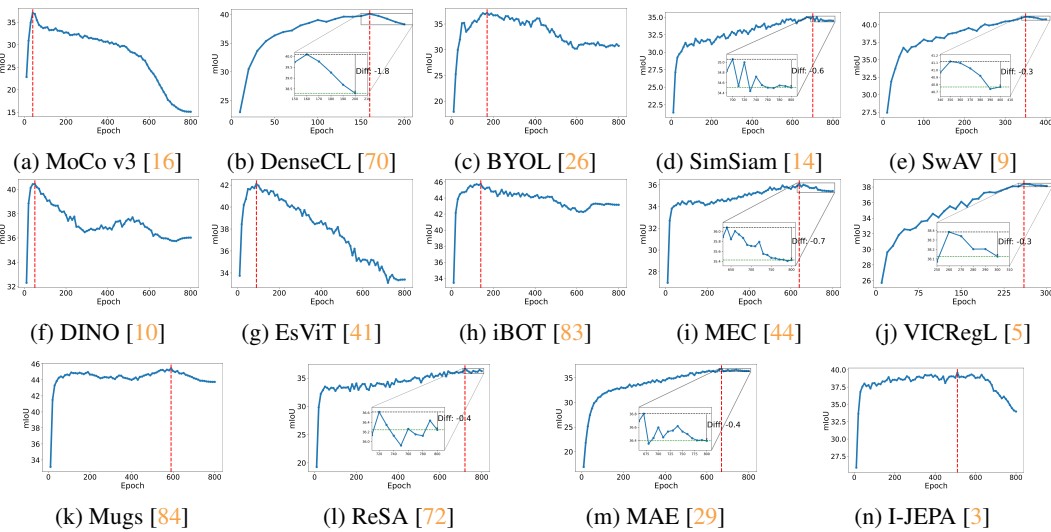

Figure 9: The SDD phenomenon on COCO-Stuff.

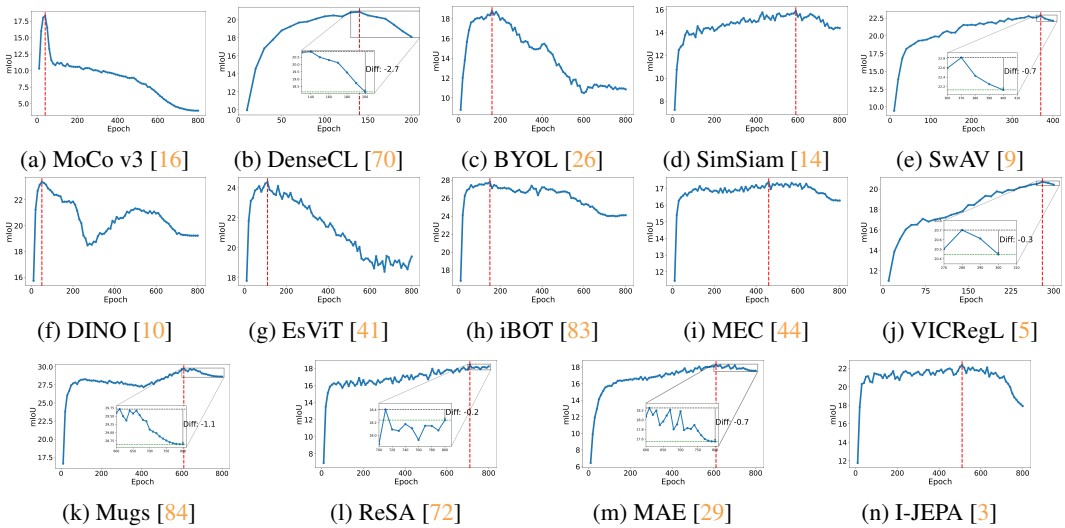

Figure 10: The SDD phenomenon on ADE20k.

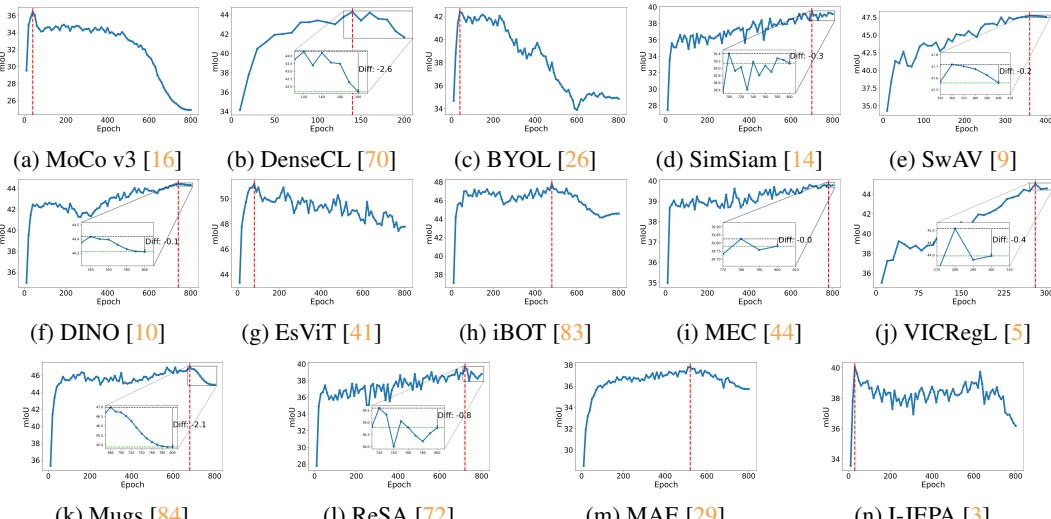

Figure 11: The SDD phenomenon on Cityscapes.

### E.2 The SDD Phenomenon Exists Under Varying Evaluation Protocols

#### E.2.1 The SDD Phenomenon Exists when Backbone is Not Frozen.

To examine how changes to the backbone during fine-tuning influence downstream performance, we evaluated checkpoints where the backbone was unfrozen and present the results in Tab. 5. The findings reveal a pattern similar to that seen in the linear probing setting, with an average decrease of 2.9% in mIoU. While fine-tuning the backbone notably improves downstream performance, the decline becomes less pronounced as the backbone continues to adapt during training. Nevertheless, a similar trend to the fixed-backbone setting remains, further supporting the existence of the SDD phenomenon.

Table 5: A performance gap between the best and the last models is present across all datasets and methods when **backbone is not frozen**.

| Method Type | Method | Architecture | COCO-Stuff Best | Last | Diff | PASCAL VOC Best | Last | Diff | ADE20k Best | Last | Diff |
|---|---|---|---|---|---|---|---|---|---|---|---|
| Contrastive | MoCo v3 [16] | ViT-Small-16 | 39.6 | 35.4 | -4.2 | 57.3 | 18.1 | -39.2 | 21.0 | 6.1 | -14.9 |
| | DenseCL [70] | ResNet-50 | 43.2 | 42.4 | -0.8 | 62.3 | 61.8 | -0.5 | 25.0 | 24.8 | -0.2 |
| Non-Contrastive | MEC [44] | ResNet-50 | 41.4 | 41.2 | -0.2 | 57.8 | 57.6 | -0.2 | 22.0 | 22.0 | 0.0 |
| | SimSiam [14] | ResNet-50 | 42.7 | 42.6 | -0.1 | 60.0 | 59.9 | -0.1 | 22.5 | 22.5 | 0.0 |
| | SwAV [9] | ResNet-50 | 38.8 | 38.8 | 0.0 | 55.3 | 55.1 | -0.2 | 20.1 | 20.0 | -0.1 |
| | DINO [10] | ViT-Small-16 | 42.6 | 41.0 | -1.6 | 62.0 | 60.6 | -1.4 | 25.6 | 25.5 | -0.1 |
| | EsViT [41] | Swin-Tiny-7 | 42.9 | 39.5 | -3.4 | 62.5 | 56.0 | -6.5 | 25.4 | 22.2 | -3.2 |
| | iBOT [83] | ViT-Small-16 | 46.9 | 44.9 | -2.0 | 69.7 | 68.2 | -1.5 | 29.1 | 28.1 | -1.0 |
| Masked Modeling | MAE [29] | ViT-Small-16 | 39.1 | 39.0 | -0.1 | 55.3 | 54.9 | -0.4 | 19.9 | 19.6 | -0.3 |
| | I-JEPA [3] | ViT-Base-16 | 46.1 | 44.8 | -1.3 | 70.0 | 67.5 | -2.5 | 29.5 | 28.1 | -1.4 |

#### E.2.2 The SDD Phenomenon Exists in Varying Evaluation Hyperparameters.

To investigate whether SDD stems from specific evaluation settings, we test different learning rates and fine-tuning durations for downstream tasks. As shown in Fig. 12 and Fig. 13, performance degradation trends remain consistent across hyperparameter configurations. This demonstrates that SDD is not an artifact of specific evaluation hyperparameters but reflects a general correlation with representation quality.

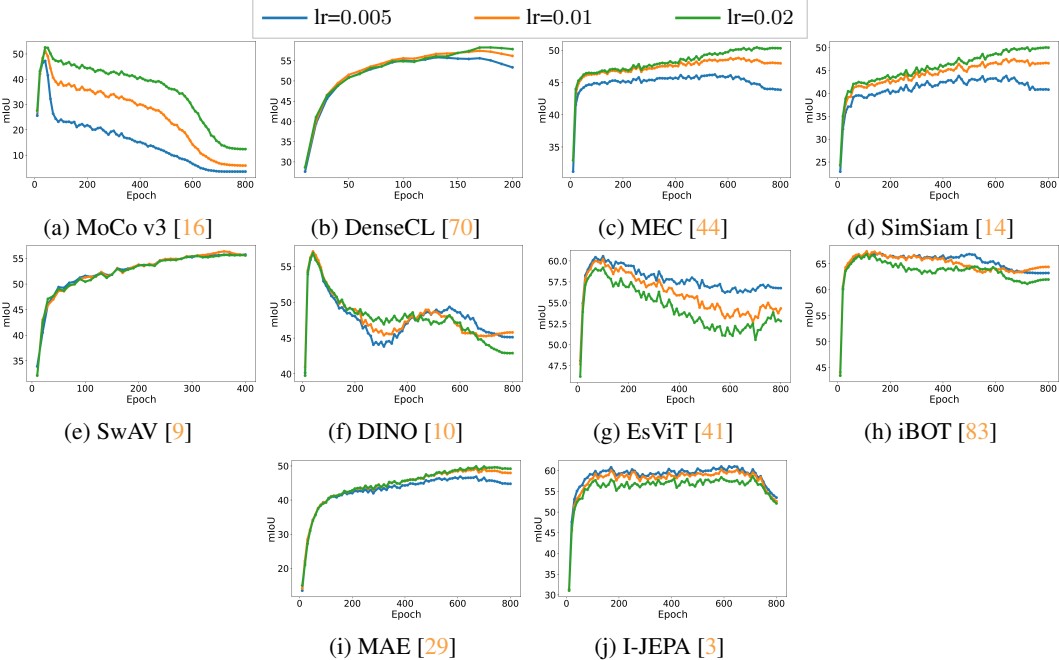

Figure 12: The SDD phenomenon consistently exists across different learning rates.

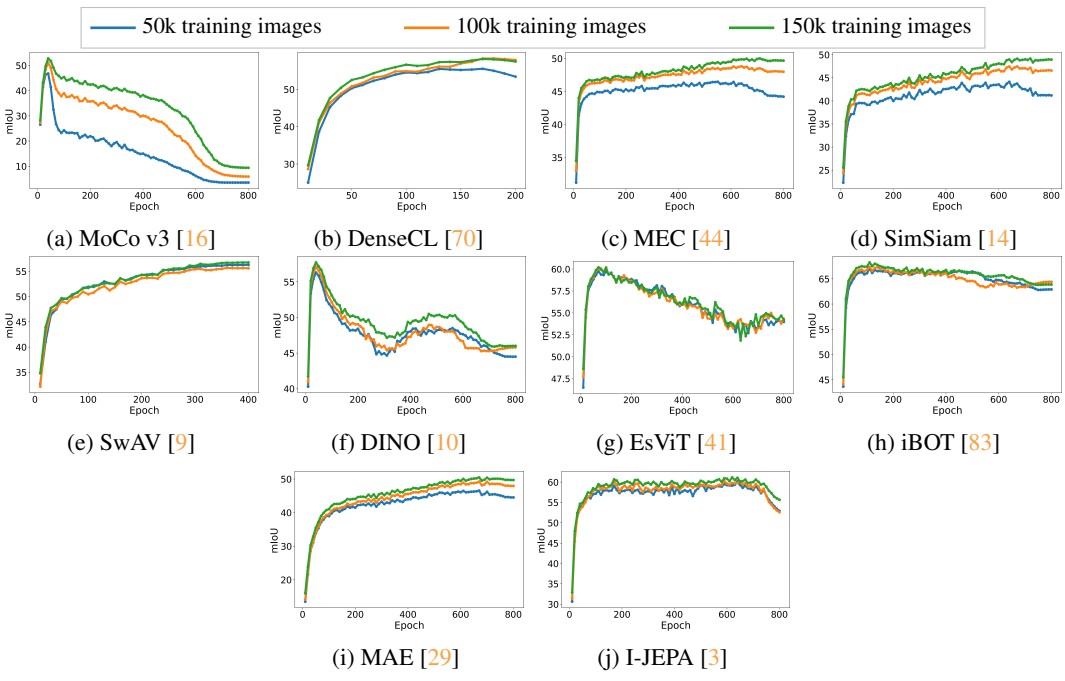

Figure 13: The SDD phenomenon consistently exists across different fine-tuning iterations.

### E.3 The SDD Phenomenon is not Caused by Dataset Overfitting

To examine whether SDD is caused by dataset overfitting, we train DINO [10] and evaluate its performance on the same COCO-Stuff dataset. All settings are kept identical to those used for training on ImageNet, except for the training dataset.

As shown in Tab. 6, the model exhibits a similar performance degradation as when pretrained on ImageNet. This results in a large gap between the best and final checkpoints, suggesting that SDD is not due to dataset overfitting.

Table 6: The SDD phenomenon when training and evaluating on the same COCO dataset.

| Method | Best | Last | Diff |
|---|---|---|---|
| DINO [10] | 36.3 | 32.3 | -4.0 |

### E.4 The SDD Phenomenon Exists in Varying Downstream Tasks

We evaluate semi-supervised video object segmentation on DAVIS-2017 [52] using the protocol from [10]. Due to the limited performance of non-ViT architectures, we focus on ViT models. Results in Tab. 7 reveal persistent performance gaps between optimal and the last checkpoints, confirming that SDD generalizes to diverse dense downstream tasks (More results on depth estimation are provided in Appendix F.1).

Table 7: SDD phenomenon of Video object segmentation task on DAVIS-2017 dataset.

| Method Type | Method | Architecture | $\mathcal{J}\&\mathcal{F}$ Best | Last | Diff | $\mathcal{J}_{\mathrm{mean}}$ Best | Last | Diff | $\mathcal{F}_{\mathrm{mean}}$ Best | Last | Diff |
|---|---|---|---|---|---|---|---|---|---|---|---|
| Contrastive | MoCo v3 [16] | ViT-Small-16 | 61.5 | 60.7 | -0.8 | 59.6 | 59.0 | -0.6 | 63.3 | 62.5 | -0.8 |
| Non-Contrastive | DINO [10] | ViT-Small-16 | 61.9 | 61.8 | -0.1 | 60.2 | 59.8 | -0.4 | 63.8 | 63.7 | -0.1 |
| | iBOT [83] | ViT-Small-16 | 62.4 | 61.6 | -0.8 | 61.1 | 60.3 | -0.8 | 64.3 | 62.9 | -1.4 |
| Masked Modeling | MAE [29] | ViT-Small-16 | 49.2 | 41.2 | -8.0 | 48.7 | 39.7 | -9.0 | 49.7 | 42.8 | -6.9 |
| | I-JEPA [3] | ViT-Base-16 | 59.6 | 54.5 | -5.1 | 58.7 | 53.6 | -5.1 | 60.6 | 55.4 | -5.2 |

# F Additional Experiment Results of the DSE Metric

## F.1 DSE Metric is a Precise Estimator for Depth Estimation Task

We further examine whether SDD is unique to semantic segmentation or affects other dense tasks. First, We conduct an additional experiments on depth estimation. Similar to our linear probing setting, we freeze the model and train a linear layer to predict the depth value on NYU-depth v2 dataset [47]. As shown in Tab. 8, the SDD phenomenon persists in depth estimation task. The results validates that SDD is a general phenomenon that not specific to segmentation task.

Next, to see if the proposed DSE metric works well on depth estimation task, we also compute the Kendall's $\tau$ coefficient between the DSE metric and depth estimation performance. As shown in Tab. 9, metric positively correlates with the depth estimation performance, and DSE-based model selection consistently improves the model performance. While these results demonstrate the effectiveness of DSE across different downstream tasks, we note that the Kendall's $\tau$ coefficient is relatively lower compared with the segmentation task. The reasons are two-fold: 1) Our DSE metric is derived from the class-relevant performance, the class-seperability may not be fully useful for depth estimation task. 2) As a regression task, the RMSE curve quakes, making it hard to predict. We would like to further imporve the DSE metric to strengthen its capability on depth estimation task.

Table 8: The SDD phenomenon in depth estimation task. We report the RMSE metric (lower is better) on NYU-depth v2 dataset.

| Method | Best ↓ | Last ↓ | Difference |
|---|---|---|---|
| MoCo v3 [16] | 0.638 | 1.589 | 0.951 |
| DenseCL [70] | 0.547 | 0.559 | 0.012 |
| SimSiam [14] | 0.597 | 0.608 | 0.011 |
| SwAV [9] | 0.705 | 0.725 | 0.020 |
| DINO [10] | 0.515 | 0.553 | 0.038 |
| MAE [29] | 0.680 | 0.775 | 0.095 |
| I-JEPA [3] | 0.460 | 0.487 | 0.027 |

Table 9: **Left: Kendall's $\tau$ coefficient between DSE and RMSE on depth estimation. Right: DSE-based model selection results.** We report RMSE metric with the improvement compared to the last epoch.

| Method | Kendall's $\tau$ | Method | RMSE ↓ |
|---|---|---|---|
| MoCo v3 [16] | 0.452 | MoCo v3 [16] | 0.638 (-0.951) |
| DenseCL [70] | 0.752 | DenseCL [70] | 0.547 (-0.012) |
| SimSiam [14] | 0.659 | SimSiam [14] | 0.598 (-0.010) |
| SwAV [9] | 0.247 | SwAV [9] | 0.714 (-0.009) |
| DINO [10] | 0.071 | DINO [10] | 0.532 (-0.021) |
| MAE [29] | 0.107 | MAE [29] | 0.709 (-0.066) |
| I-JEPA [3] | 0.190 | I-JEPA [3] | 0.479 (-0.008) |

## F.2 DSE Precisely Indicates Performance Trends Across Unseen Datasets

To comprehensively analyze the relationship between DSE and downstream performance, we present full results across all datasets and methods. These results confirm that DSE consistently identifies performance trends and peaks, establishing it as a reliable metric for model selection.

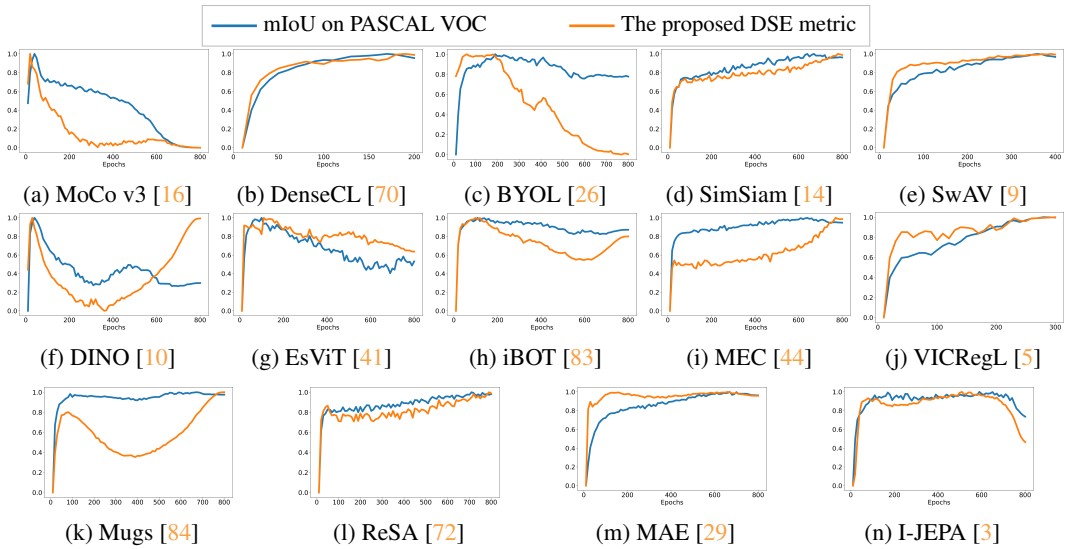

Figure 14: The proposed DSE metric precisely predicts the downstream performance on the PASCAL VOC dataset.

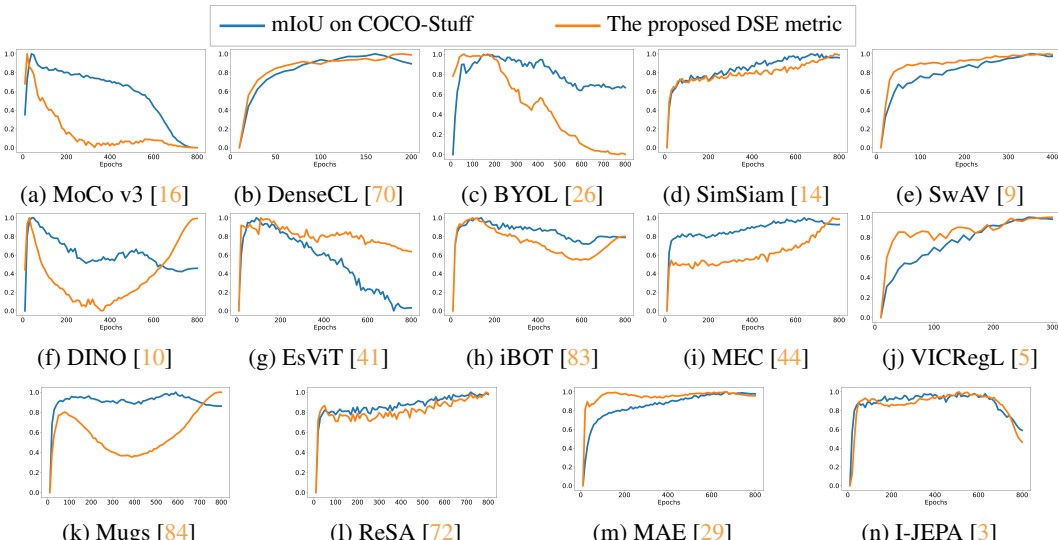

Figure 15: The proposed DSE metric precisely predicts the downstream performance on the COCO-Stuff dataset.

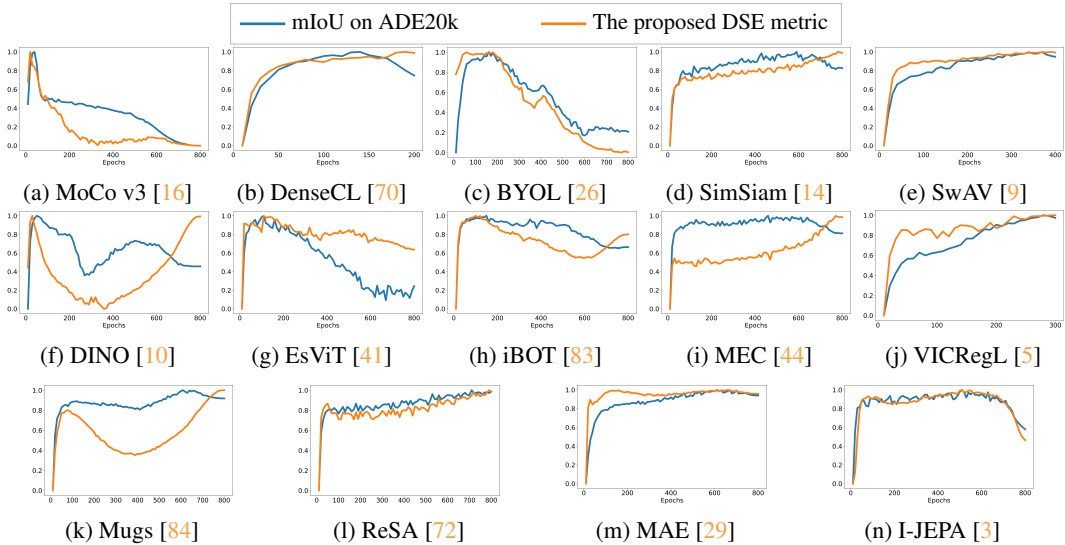

Figure 16: The proposed DSE metric precisely predicts the downstream performance on the ADE20k dataset.

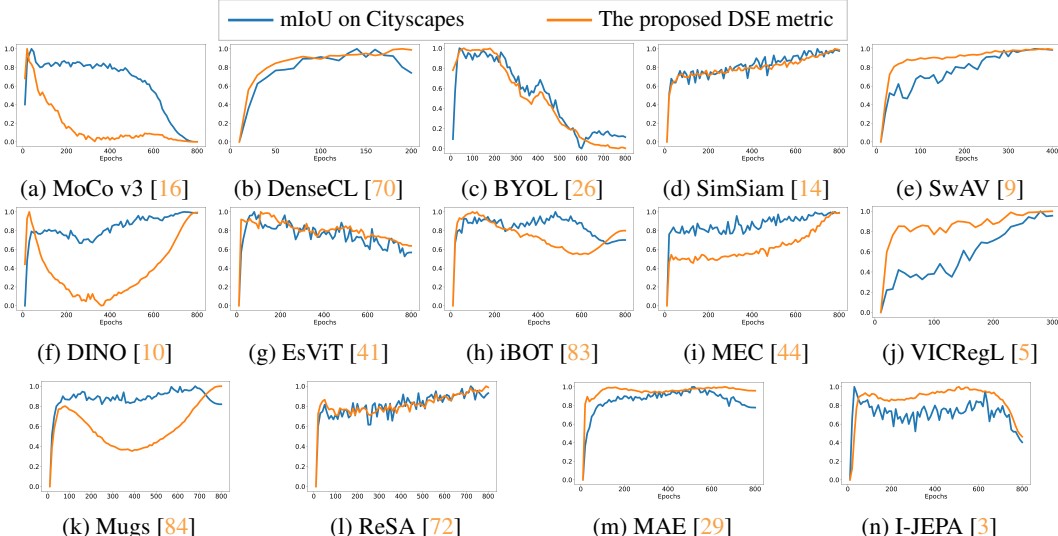

Figure 17: The proposed DSE metric precisely predicts the downstream performance on the Cityscapes dataset.

### F.3 Ablation Study of Different Components.

As shown in Tab. 10, both the class-separability and dimensionality measures exhibit a positive correlation with downstream performance, supporting our theoretical findings. Combining these components further enhances Kendall's $\tau$ coefficient. Notably, when only $M_{dim}$ is used (Line 2 in Tab. 10), it is equivalent to adapting the RankMe metric [24] to dense representations. These results highlight DSE's superior ability to assess dense representation quality.

### F.4 Sensitivity Analysis on Number of Images and Clusters

While DSE itself does not explicitly depend on hyperparameters, pseudo-label generation introduces unavoidable parameters. We analyze their sensitivity to assess whether DSE's accuracy is contingent on specific choices.

Table 10: Ablation study of different components of DSE. We report the average Kendall's $\tau$ coefficient across different methods.

| $M_{inter} - M_{intra}$ | $M_{dim}$ | COCO | VOC | ADE | City | Avg |
|:---:|:---:|:---:|:---:|:---:|:---:|:---:|
| ✓ | | 0.45 | 0.42 | 0.33 | 0.37 | 0.39 |
| | ✓ | 0.25 | 0.26 | 0.22 | 0.23 | 0.24 |
| ✓ | ✓ | **0.58** | **0.60** | **0.56** | **0.49** | **0.57** |

**Effect of the number of images.** By default, we only use a small amount of data (2048 images, $\sim 0.16\%$ of the training dataset). This is the minimum batch size $B'$ to meet the requirement of $B' \gg d$ in order to obtain $B'$ independent representations. In this part, we test whether this limited sample size suffices for robust DSE estimation. Fig. 18 shows no significant change in DSE with larger datasets, confirming that 2,048 images suffice for accurate performance estimation.

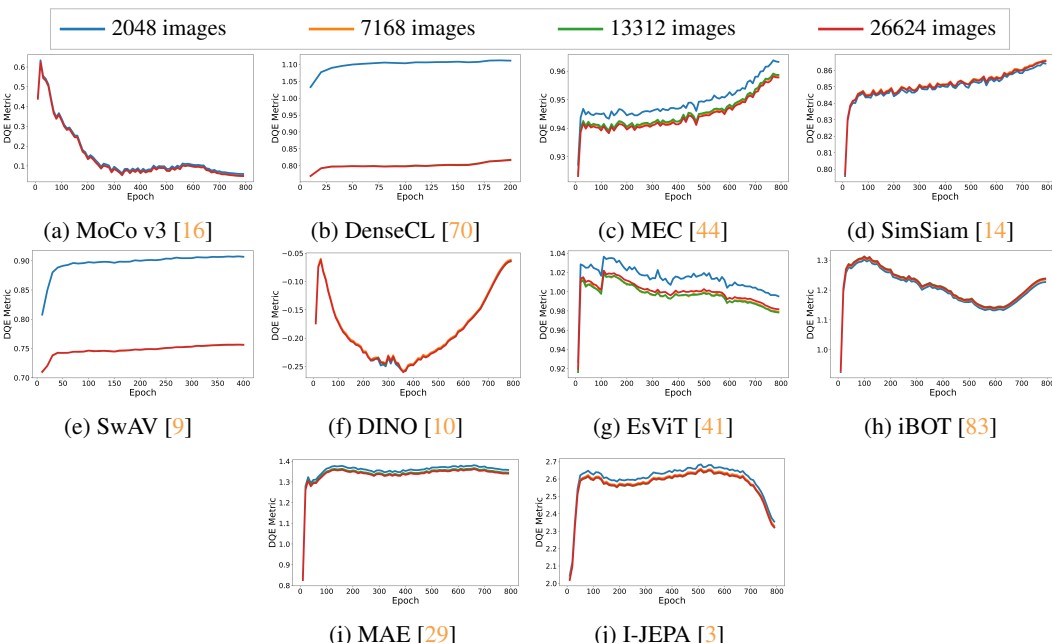

Figure 18: Sensitivity analysis of different numbers of images used for the DSE calculation.

**Effect of the number of clusters.** In this part, we examine the impact of the choice of $k$. The number of clusters is a key parameter for estimating representation quality because the actual number of classes in an image is unknown without labels. As a result, the estimated class-related metrics may be biased if the number of clusters does not match the true number of classes. Based on the earlier theoretical analysis, we recommend choosing $k$ slightly larger than the expected number of classes. Since ImageNet is curated, we assume that the average number of classes per image (including the background as one class) is about 2 or 3. Therefore, we explore the effect of setting $k$ between 3 and 5. Fig. 19 shows that while absolute DSE values vary with $k$, performance trends remain consistent, preserving DSE's utility of predicting the relative performance between models.

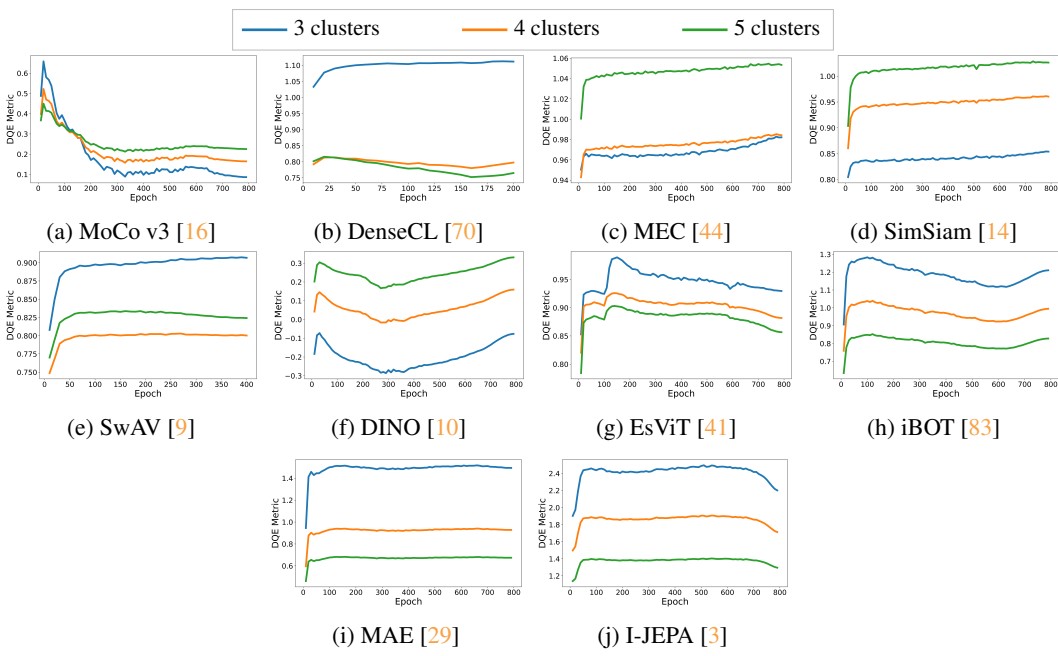

Figure 19: Sensitivity analysis of different numbers of clusters $k$.

## F.5 Analysis of Bias Introduced by Data Distribution Shift and Label Estimation Error

When estimating downstream performance, two types of bias are introduced: 1) Data distribution shift—DSE is computed on the training dataset (i.e., ImageNet), while the target performance is evaluated on testing sets (e.g., COCO, VOC). 2) Label estimation error—DSE relies on pseudo-labels generated by the $k$-means algorithm, which may introduce bias during label assignment.

We conduct two groups of experiments to analyze their impact: 1) DSE calculated on the training set, which serves as the default setting for metric computation, and 2) DSE calculated on the testing set using ground truth labels. As shown in Fig. 21, Fig. 20, and Fig. 22 although systematic errors are present, these biases act as shifting factors rather than altering the overall trend of $M_{inter}$, $M_{intra}$, and $M_{dim}$. Consequently, the estimated DSE metric remains reliable for comparing model performance.

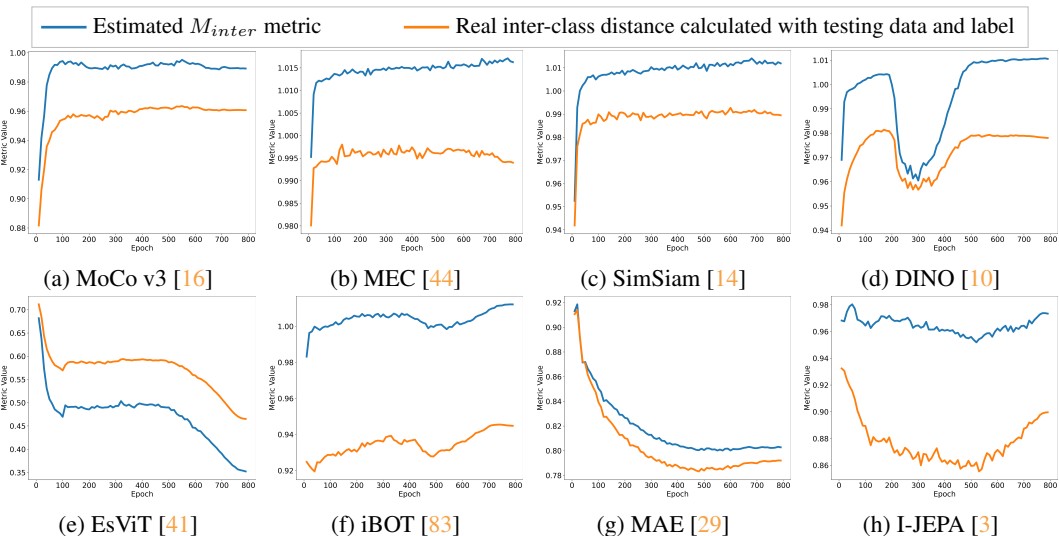

Figure 20: Comparison between the estimated $M_{inter}$ and the real inter-class distance.

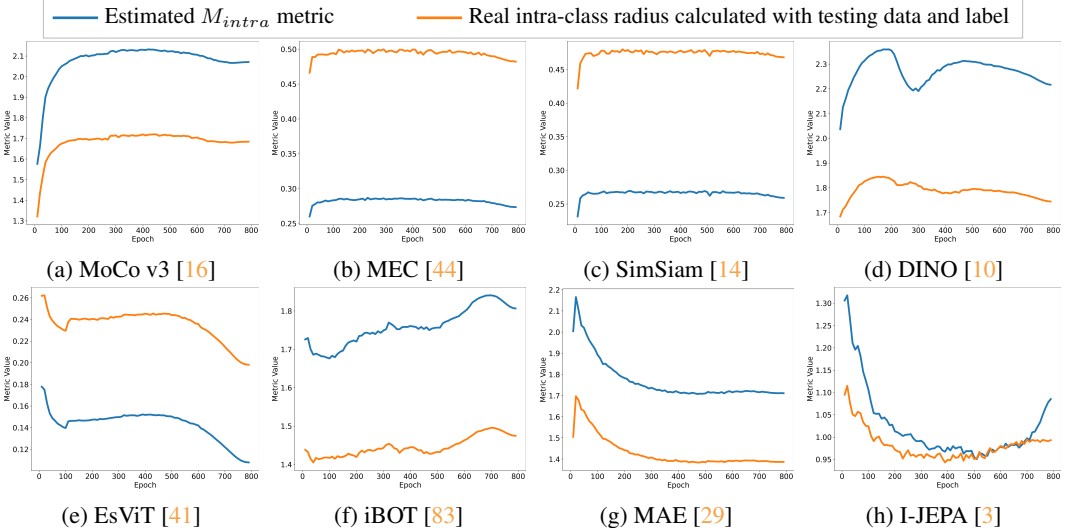

Figure 21: Comparison between the estimated $M_{intra}$ and the real intra-class radius.

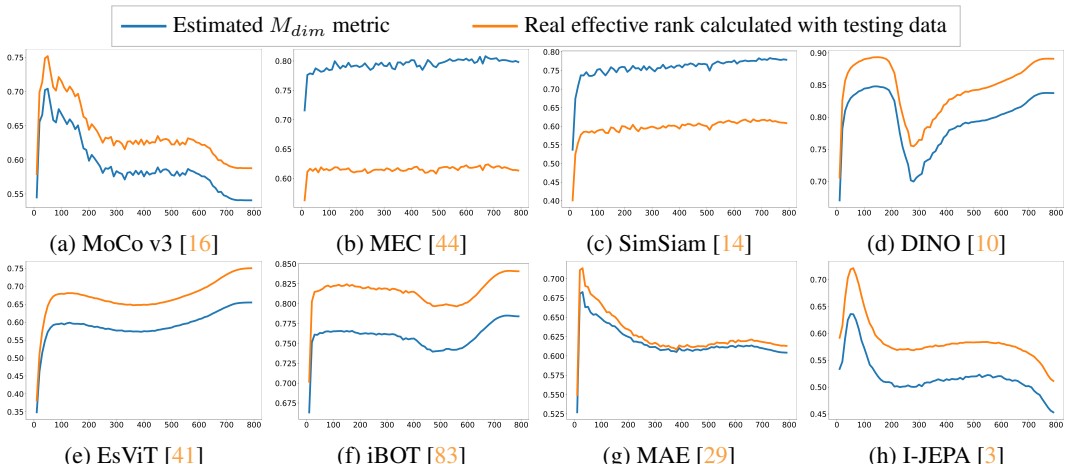

Figure 22: Comparison between the estimated $M_{dim}$ and the real effective rank.

## F.6 Understanding the Reason behind SDD Phenomenon

DSE not only measures downstream performance but also helps explain the causes of SDD. The behavior of different components of DSE offers insights into SDD. For instance, Fig. 23 shows that DINO's performance drop at 300 epochs coincides with a slower reduction in intra-class distance compared to inter-class distance, which decreases class separability. In contrast, MoCo v3's degradation is associated with a collapse in the dimensionality of dense features, making downstream separation more difficult. These findings offer a deeper understanding of the training dynamics in SSL methods and provide useful guidance for improving algorithm design.

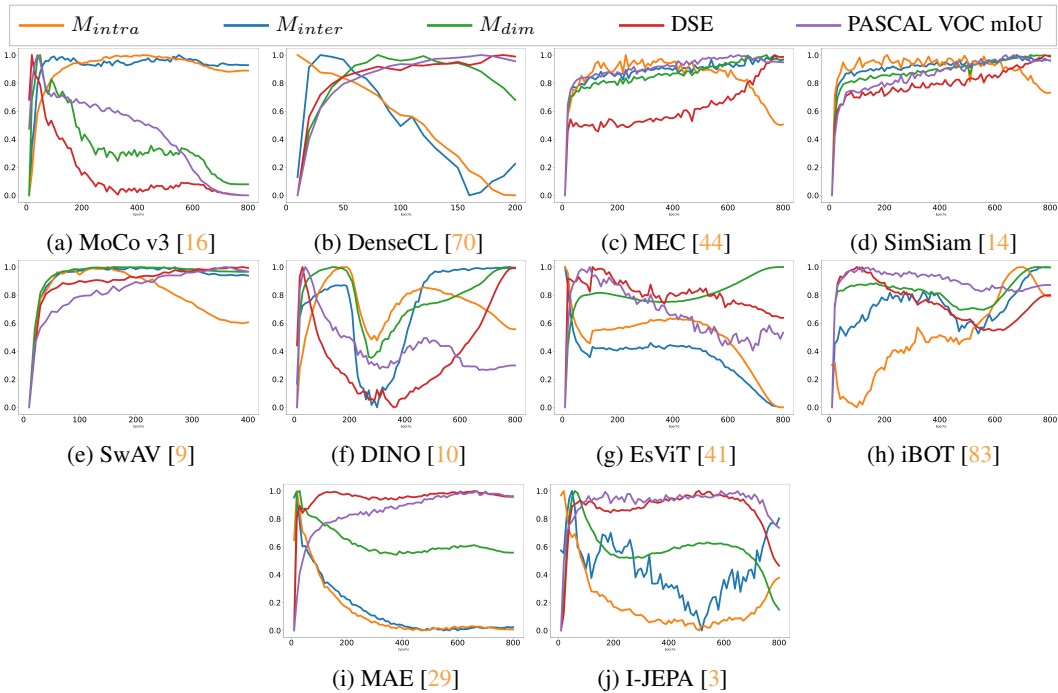

Figure 23: Visualization of different components of DSE metric, which reveals the reason behind SDD phenomenon.

## F.7 Extending the DSE Metric toward Image-level Performance Estimation

Technically, our analysis should hold for both dense and image-level settings, as image-level learning can be viewed as a special case where an image is only split into one patch. To investigate how DSE performs at the image level, we conducted the following experiments. DSE originally accepts input tensors shaped as (Num_images, Num_patch_tokens, Dimensionality). For image-level tasks, we first extract the class token (or averaged dense tokens), obtaining a tensor of shape (Num_images, Dimensionality). Then, we resize this tensor to (Num_images / 200, 200, Dimensionality) and calculate the DSE metric. For evaluation, we computed Kendall's $\tau$ coefficient between the adapted DSE and ImageNet $k$-NN performance, comparing it with the state-of-the-art unsupervised transferability estimation method RankMe. The results are shown in the following table:

Table 11: Kendall's $\tau$ coefficient between metrics and ImageNet $k$-NN performance.

| Method | DINO | MEC | EsViT | iBOT | I-JEPA | Average |
|--------|------|-----|-------|------|--------|---------|
| RankMe | 0.57 | 0.81 | 0.71 | **0.93** | **0.90** | 0.79 |
| DSE(Ours) | **0.87** | **0.90** | **0.92** | 0.86 | 0.73 | **0.86** |

## F.8   Visualizations

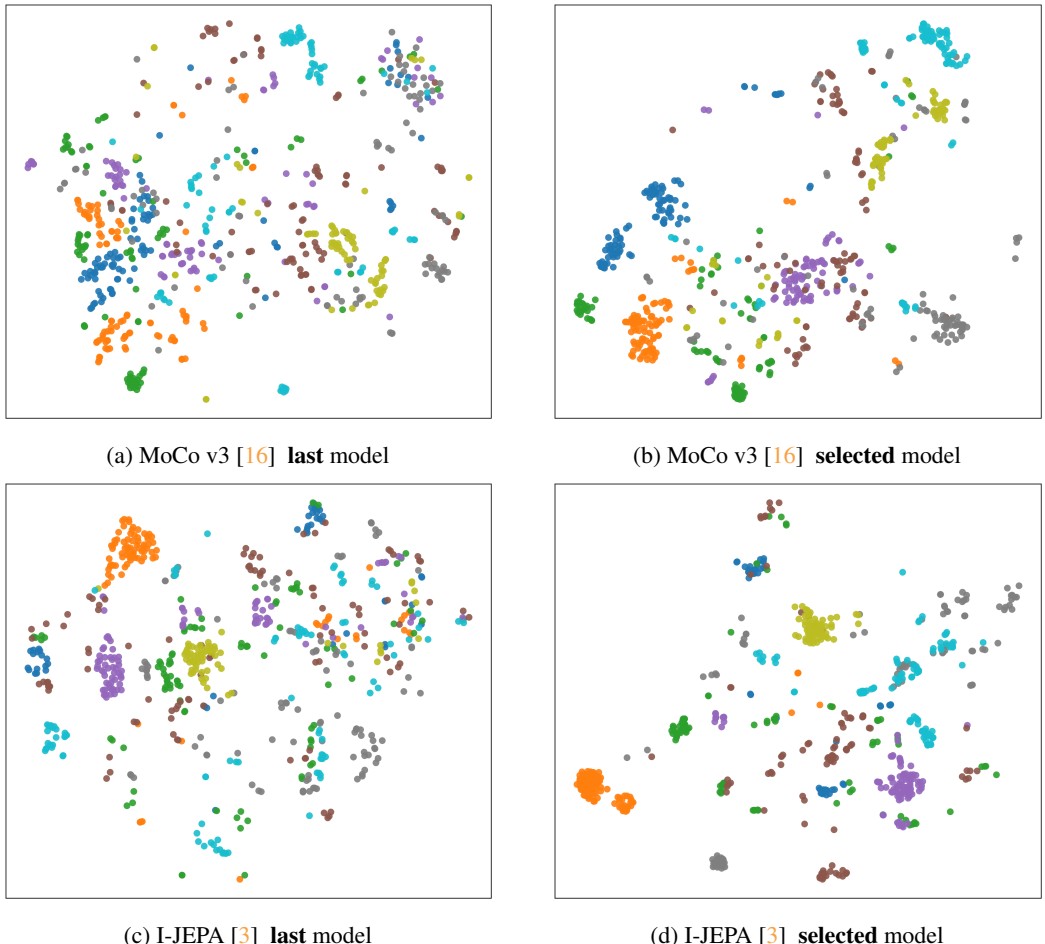

(a) MoCo v3 [16] **last** model

(b) MoCo v3 [16] **selected** model

(c) I-JEPA [3] **last** model

(d) I-JEPA [3] **selected** model

Figure 24: The selected model achieves better intra-class alignment and intra-class separatebility compared with the last model.

Furthermore, we visualize the t-SNE plot and representation similarity matrix of the last and the selected model. The inputs are sorted by class labels thus an ideal similarity matrix should be block diagonal. As spotted in Fig. 24 and Fig. 25, the selected models have better intra-class alignment and larger inter-class distance, which means that it would be easier for the downstream classifier to separate the representations. The results well support our theoretical and empirical conclusions.

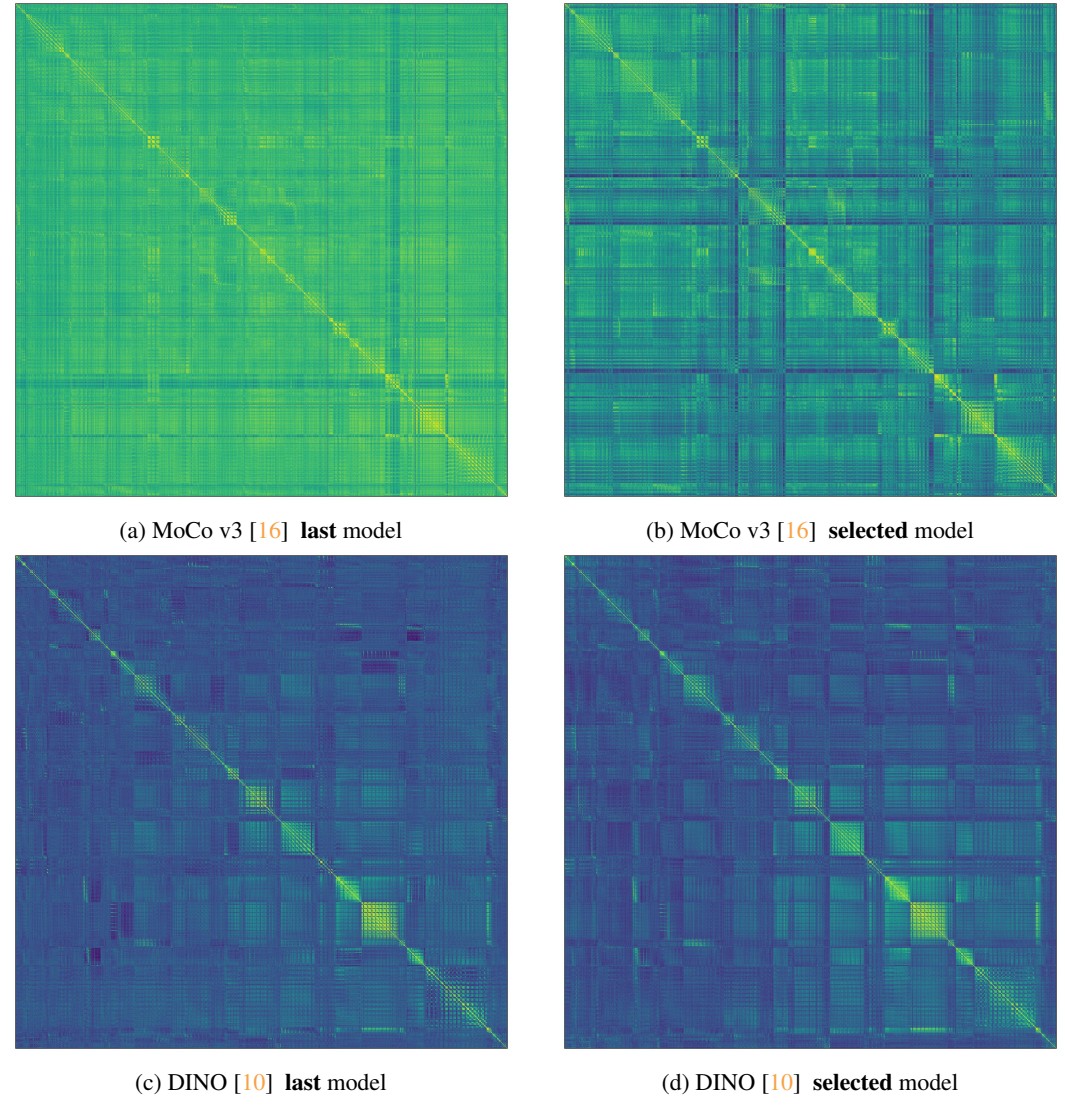

(a) MoCo v3 [16] **last** model

(b) MoCo v3 [16] **selected** model

(c) DINO [10] **last** model

(d) DINO [10] **selected** model

Figure 25: The selected model achieves better intra-class alignment and intra-class separatebility compared with the last model.

### F.9 Ablation Study of DSE Regularization

To evaluate the effectiveness of different components of DSE, we conduct an additional ablation study in Tab. 12. We draw three main conclusions from these results:

- Different components of DSE separately contribute to improving class separability and effective dimensionality. Consequently, this improves downstream performance, which aligns with our theoretical analysis of the DSE metric.

- The performance improvement mainly comes from addressing model-specific degradation. For example, DINO's performance improves primarily due to enhanced class separability, whereas resolving dimensional collapse notably boosts dense-task performance in MoCo v3. These observations align with our model-specific analysis presented in Sec. 5.2.

- When both factors are used together, class separability often dominates optimization. Thus, introducing a hyperparameter to balance these factors may further enhance performance.

Table 12: Ablation studies of different components of DSE regularization. We report $M_{inter} - M_{intra}$, $M_{dim}$, and mIoU on VOC dataset.

| Method | $M_{inter} - M_{intra}$ | $M_{dim}$ | VOC mIoU |
|---|---|---|---|
| DINO [10] | -1.221 | 0.863 | 56.6 |
| DINO [10]+ $M_{dim}$ | -1.210 (+0.011) | 0.884 (+0.021) | 56.9 (+0.3) |
| DINO [10]+ $M_{cls}$ | -1.103 (+0.118) | 0.851 (-0.012) | 57.4 (+0.8) |
| DINO [10]+ $M_{dim}$ + $M_{cls}$ (DSE) | -1.115 (+0.106) | 0.865 (+0.002) | **57.8 (+1.2)** |
| MoCo v3 [16] | -0.958 | 0.744 | 49.1 |
| MoCo v3 [16]+ $M_{dim}$ | -1.021 (-0.063) | 0.892 (+0.148) | 52.3 (+3.2) |
| MoCo v3 [16]+ $M_{cls}$ | | collapsed | |
| MoCo v3 [16]+ $M_{dim}$ + $M_{cls}$ (DSE) | -0.867 (+0.091) | 0.752 (+0.008) | **52.1 (+3.0)** |

# G Additional Discussions

**Limitations.** This paper's theoretical analysis mainly focuses on the linear probing setting and does not account for potential distributional shifts during transfer learning. When the backbone is fine-tuned for more iterations on data that differs significantly from the pre-training distribution, the estimated performance may become inaccurate.

**Future Works** Several prior studies on supervised transferability estimation have addressed related issues [33, 35, 76, 20, 19, 7, 1, 56, 42]. Extending the proposed DSE metric to such scenarios would be an interesting direction for future research.

It would also be valuable to investigate model-specific causes of dense degradation. For example, understanding why dimensional collapse occurs in MoCo v3 or what leads to separability degradation in DINO could provide deeper insights.

**Extensions.** Although the proposed DSE metric is primarily designed to address the SDD phenomenon, and we introduce two methods to reduce its negative impact, its applications are not limited to this context. For example, image-level tasks can be viewed as a special case of dense tasks, where the number of patches is one. In such cases, our theoretical analysis and the DSE metric can be applied directly.

