# OpenReview forum: "Exploring Structural Degradation in Dense Representations for Self-supervised Learning"
_NeurIPS.cc/2025/Conference — NeurIPS 2025 poster_

### Official Review · Reviewer_pWRt · 2025-06-21

**Clarity:** 2
**Significance:** 3
**Originality:** 2
**Rating:** 5
**Confidence:** 4

**Summary:**

This paper identifies Self-supervised Dense Degradation (SDD), a phenomenon where longer self-supervised learning (SSL) pretraining improves image-level tasks (e.g., classification) but degrades dense task performance (e.g., semantic segmentation). The authors demonstrate SDD’s prevalence across 10 diverse SSL methods (contrastive, non-contrastive, masked modeling) and 4 benchmarks. To address SDD, they propose the Dense Representation Quality Estimator (DQE), a theoretically grounded unsupervised metric combining class separability (inter-class distance vs. intra-class radius) and effective dimensionality.

**Questions:**

1. Figure 5 and Table 1 show that there is not a strong correlation between DINO, the proposed DQE metric, and downstream performance. What could be the reason for this?
2. The class-separability measure in Section 3 is interesting. A relevant paper published at ICML 2025 introduces a novel clustering-based SSL method called ReSA[1], which demonstrates the ability to achieve more fine-grained representations and better class-separability compared to previous methods like DINO and MoCoV3. I am curious whether this method might also exhibit the SSD phenomenon.

[1] Clustering Properties of Self-Supervised Learning.

**Ethical Concerns:**

["NO or VERY MINOR ethics concerns only"]

**Final Justification:**

Most of my concerns are adequately addressed.

**Limitations:**

The authors do not mention potential limitations in the paper.

**Paper Formatting Concerns:**

No.

**Quality:**

3

**Strengths And Weaknesses:**

## Strengths
1. The concept of Self-supervised Dense Degradation (SDD) is a valuable contribution, highlighting a gap in SSL research that was previously underexplored.
2. The DQE metric is validated through extensive experiments, showing its high correlation with downstream dense performance.

## Weaknesses
1. Experiments focus solely on semantic segmentation. While the authors briefly mention video object segmentation (Table 6), broader validation (e.g., object detection, depth estimation) would strengthen claims about "dense tasks."
2. Lack of Deep Insight into Causes of Degradation. While the paper presents a broad theoretical understanding of SDD, it could delve deeper into the underlying reasons for the degradation across different SSL methods, providing more clarity on the specific mechanisms at play.
3. The paper lacks an analysis of the evaluation setup, which affects the credibility of the experimental results. The authors state, "For ViT models, we use the last layer’s patch embeddings; for ResNet50 and Swin Transformer, we extract features from the fourth layer." However, they do not explain why this setup was chosen, nor do they analyze whether using intermediate layers in ViT for segmentation embeddings might yield better results than using the last layer. The experimental results presented in the paper show that contrastive learning-based methods, such as DINO and MoCoV3, perform poorly with ViT, while SimSiam and DenseCL perform well with ResNet. This suggests that there might be an issue with the evaluation setup for ViT. I recommend that the authors provide a detailed analysis to explain this phenomenon.

---

> ### Author Rebuttal · Authors · 2025-07-31
>
> Thank you for your time and effort in reviewing our paper! Please find our detailed responses below.
>
> **Q1:** Broader validation on other tasks.
>
> **R1:** We have conducted additional experiments on the depth estimation task. Following our linear probing setting, we freeze the model and train a linear layer to predict depth values on the NYU-depth v2 dataset. The results are presented in the table below:
>
> **Table 1:** The SDD phenomenon on the depth estimation task, reporting RMSE (lower is better) on the NYU-depth v2 dataset.
>
> | Method | Best $\downarrow$ | Last $\downarrow$ | Difference |
> |-|-|-|-|
> | MoCo v3 | 0.638 | 1.589 | 0.951 |
> | DenseCL | 0.547 | 0.559 | 0.012 |
> | SimSiam | 0.597 | 0.608 | 0.011 |
> | SwAV | 0.705 | 0.725 | 0.020 |
> | DINO | 0.515 | 0.553 | 0.038 |
> | MAE | 0.680 | 0.775 | 0.095 |
> | I-JEPA | 0.460 | 0.487 | 0.027 |
>
> **As shown in Table 1, the SDD phenomenon also appears in the depth estimation task.** The performance trend is similar to that observed in the segmentation task. These results confirm that SDD is a general phenomenon and is not limited to segmentation. While we cannot present the full performance curves due to NeurIPS 2025 policy, we will include them in the final version.
>
> To further evaluate the effectiveness of the proposed DQE metric for depth estimation, we also compute Kendall's $\tau$ coefficient between the DQE metric and depth estimation performance. The results are shown below:
>
> **Table 2:** The Kendall's $\tau$ coefficient between DQE and RMSE metric.
> | Method | Kendall's $\tau$|
> |-|-|
> | MoCo v3 | 0.452 |
> | DenseCL | 0.752 |
> | SimSiam | 0.659 |
> | SwAV | 0.247 |
> | DINO | 0.071 |
> | MAE | 0.107 |
> | I-JEPA | 0.190 |
>
>
> **As shown in Table 2, the DQE metric is positively correlated with depth estimation performance.** Although the DQE metric works across different downstream tasks, the Kendall's $\tau$ coefficient is lower compared to the segmentation task. There are two main reasons: 1) The DQE metric is derived from segmentation performance, and class separability may not fully reflect the needs of depth estimation. 2) As a regression task, the RMSE curve is less stable, making prediction more difficult. We plan to further improve the DQE metric to enhance its effectiveness for depth estimation.
>
> **Table 3:** The DQE-based model selection results on the depth estimation task.
> | Method  | RMSE $\downarrow$ |
> |-|-|
> | MoCo v3 | 0.638(-0.951) |
> | DenseCL | 0.547(-0.012) |
> | SimSiam | 0.598(-0.010) |
> | SwAV |    0.714(-0.009) |
> | DINO |    0.532(-0.021) |
> | MAE |     0.709(-0.066) |
> | I-JEPA |  0.479(-0.008)|
>
> **Table 3 shows that DQE-based model selection consistently improves model performance**. This demonstrates that DQE is generalizable to different downstream tasks.
>
> **Q2:** Lack of deep insight into causes of degradation.
>
> **R2:** Intuitively, SSL aims to balance semantic alignment and the effective dimensionality of representations. These two objectives often involve a trade-off. For instance, fully uniform representations have high effective dimensionality but may lack separability; conversely, focusing too much on semantic alignment can lead to dimension collapse. We argue that the degradation seen in different methods is due to the failure to maintain this trade-off during training, which causes a bias toward one objective.
>
> **Q3:** Issue with the evaluation setup.
>
> **R3:** Thank you for your detailed review. In fact, **we use only the last layer's embeddings for evaluation** as is standard in SSL [1,2,3,4]. For ResNet50 and Swin Transformer, the fourth layer is exactly the last layer, as both have only four layers (stages). We apologize for any confusion caused by our previous description and have revised the experimental setup section for clarity.
>
> Beyond this clarification, we also find it interesting and important to explore the SDD phenomenon at different representation layers. We evaluated the dense performance of various layers of DINO on the COCO-Stuff dataset. The results are shown below:
>
> **Table 4:** SDD phenomenon in different layers.
> | Method    | Best | Last | Diff |
> |-|-|-|-|
> | 6th layer  | 30.0 | 29.7| -0.3  |
> | 9th layer  | 39.1 | 37.3| -1.8  |
> | 12th layer | 40.4 | 36.0| -4.4  |
>
> While the SDD phenomenon is weaker in shallower layers, the overall performance trend remains similar. At the 6th layer, the SDD phenomenon is almost negligible, but the performance is much worse, and it is not suitable for downstream tasks.
>
> **Q4:** Analysis of failure case in DINO.
>
> **R4:** Thank you for your insightful comment. The main reason is that DINO uses an EMA update strategy, where the momentum coefficient $\beta$ increases from 0.996 to 1 during training. In the later training stages, the teacher model stops updating, making the student more likely to experience dimension collapse when aligning to a fixed target. As shown in Fig. 5, this leads to a significant increase in the dimensionality metric. While our metric captures this trend, the large value of $\lambda$ gives too much weight to the dimensionality metric, causing a mismatch between the final DQE score and actual performance. Therefore, for DINO, it may be necessary to dynamically adjust $\lambda$ together with the EMA momentum coefficient to better account for different training phases.
>
> **Q5:** More discussion about clustering-based SSL method ReSA [5]
>
> **R5:** Thank you for pointing out this related study. We find the idea of clustering-based optimization in ReSA [5] interesting and novel, and it achieves strong experimental results. To investigate whether clustering-based optimization affects SDD and the DQE metric, we reproduced ReSA and tested its dense performance. The results are shown below:
>
> **Table 5:** The SDD phenomenon in ReSA.
> | Method | COCO Best| COCO Last | COCO Diff | VOC Best | VOC Last | VOC Diff | ADE Best | ADE Last | ADE Diff |
> |-|-|-|-|-|-|-|-|-|-|
> | ReSA | 36.6 | 36.2 | -0.4 | 49.7 | 49.1 | -0.6 | 18.4 | 18.2 | -0.2 |
>
> **Table 6:** The Kendall's $\tau$ coefficient between the DQE metric and dense performance.
> | Method | COCO | VOC | ADE20K |
> |-|-|-|-|
> | ReSA | 0.74 | 0.72 | 0.71 |
>
> **Table 7:** The DQE-based model selection results on ReSA.
> | Method | COCO | VOC | ADE20K |
> |-|-|-|-|
> | ReSA | 36.6(+0.4) | 49.5(+0.4) | 18.3(+0.1) |
>
>
> As shown, the SDD phenomenon also occurs in ReSA. This may be because clustering optimization is not directly designed for dense representations in ReSA. The DQE metric still aligns well with ReSA’s performance. Therefore, **the main contributions of our paper remain valid for ReSA.**
> While we have tried to reduce the SDD phenomenon by directly optimizing the DQE metric, we believe further work exploring clustering-based optimization for dense representations, or studying how SDD behaves on encodings and embeddings, would be valuable future directions. We will add the ReSA results to the final version as an important baseline and provide more discussions in our paper.
>
> [1] Caron, Mathilde, et al. "Emerging properties in self-supervised vision transformers." Proceedings of the IEEE/CVF international conference on computer vision. 2021.
>
> [2] Oquab, Maxime, et al. "Dinov2: Learning robust visual features without supervision." arXiv preprint arXiv:2304.07193 (2023).
>
> [3] Zhou, Jinghao, et al. "ibot: Image bert pre-training with online tokenizer." arXiv preprint arXiv:2111.07832 (2021).
>
> [4] Chen, Xinlei, Saining Xie, and Kaiming He. "An empirical study of training self-supervised vision transformers." Proceedings of the IEEE/CVF international conference on computer vision. 2021.
>
> [5] Weng, Xi, et al. "Clustering Properties of Self-Supervised Learning." arXiv preprint arXiv:2501.18452 (2025).

---

> > ### Comment · Reviewer_pWRt · 2025-08-02
> > **Review update**
> >
> > Thanks for your detailed rebuttal to address my concerns. I believe that incorporating the reviewers’ suggestions will make the paper more solid. I'd like to raise my score to 5. Best wishes!

---

> > > ### Author Response · Authors · 2025-08-02
> > >
> > > We sincerely appreciate your kind words and the improved score. Your constructive comments have greatly helped us enhance the soundness of our paper. We will carefully revise our manuscript to ensure that these improvements are properly incorporated into the final version.

---

### Official Review · Reviewer_riT6 · 2025-06-21

**Clarity:** 4
**Significance:** 3
**Originality:** 4
**Rating:** 5
**Confidence:** 4

**Summary:**

This paper identifies a general and critical issue affecting a wide range of image self-supervised learning algorithms, termed Self-supervised Dense Degradation (SDD). This issue reveals that extended SSL training degrades performance on dense tasks (such as segmentation), even as it improves image-level classification. This finding highlights a fundamental limitation of modern image self-supervised learning approaches. To address this, the authors propose a theoretically grounded metric for evaluating dense representations without the need for labels or downstream data, called the Dense Representation Quality Estimator (DQE). The DQE is based on two key factors: class separability and effective dimensionality. Notably, the DQE outperforms existing unsupervised metrics (e.g., RankMe, α-ReQ), demonstrating a strong correlation to downstream mIoU with a Kendall’s τ value of 0.54. When integrating DQE into the SSL training process, it reverses degradation trends and boosts performance with negligible computational overhead.

**Questions:**

1. Regarding the class separability factor in DQE, is it measured at the image level or the pixel level? If it is at the image level, how does this factor accurately capture the performance trends of downstream dense tasks?
2. As discussed in section 5.3, different methods experience the SDD phenomenon for various reasons. Could you explain why this variation occurs? For instance, MoCo v3’s performance degradation is attributed to dimensional collapse in dense features. Does this imply that increasing the dimensionality of MoCo v3 could lead to better performance?

**Ethical Concerns:**

["NO or VERY MINOR ethics concerns only"]

**Final Justification:**

Most of my concerns are well addressed. I'm satisfied with its current form. Thus I will maintain my score. Thank you!

**Limitations:**

1. The theoretical analysis of DQE is based on sub-Gaussian distributions assumptions, which might not be true for real-world cases.
2. The performance for other dense tasks beyond segmentation is under-explored in this work, for instance, depth estimation.

**Paper Formatting Concerns:**

I don't have further formatting concerns.

**Quality:**

3

**Strengths And Weaknesses:**

Strengths
1. **Novel Phenomenon Discovery**: This paper identifies SDD—a counter-intuitive degradation in dense tasks during prolonged SSL training—which challenges the established belief that "longer pretraining always helps." Additionally, the authors demonstrate SDD’s consistency across 10 SSL methods, 4 benchmarks, and diverse architectures (ViTs, CNNs).
2. **Theoretical Grounding**: The authors provide a robust theoretical foundation by deriving a principled link between downstream error and two factors: class separability and effective dimensionality. This theorem ultimately leads to the development of the DQE.
3. **Practical Solution for SDD**: The evaluations indicate that DQE serves as an effective and practical solution for addressing SDD. As a label-free metric, when applied to model selection, DQE results in a +4.0% mIoU gain with negligible computational cost. Furthermore, when used as a training regularizer, it reverses SDD, thereby improving both DQE scores and task performance.

Weaknesses
1. **Limitations discussion**: It would be beneficial to further discuss the limitations of proposed DQE, for example, the failure mode, the SDD causes (e.g., dimensional collapse vs. separability loss) might offer some architecture-specific fixes etc.

---

> ### Author Rebuttal · Authors · 2025-07-31
>
> Thank you very much for your valuable feedback and suggestions. We appreciate your positive comments on the SDD phenomenon and our proposed DQE metric. Below are our detailed responses:
>
> **Q1:** Limitations discussion
>
> **R1:** Thank you for your valuable comment. As discussed in Appendix G, our DQE metric primarily focuses on off-the-shelf performance estimation. Therefore, when the backbone is fine-tuned for an extended period or with a large prediction head, the estimated performance may become inaccurate. In addition, the DQE is designed as a general performance estimator. Explaining the model-specific causes of the SDD phenomenon, such as the degradation in MoCo v3 or DINO, is an interesting and important future direction. We will move the limitations discussion into the main paper in our revision.
>
> **Q2:** Regarding the class separability factor in DQE, is it measured at the image level or the pixel level?
>
> **R2:** We apologize for the confusion. Both the class separability and dimensionality measures are calculated on **patch/pixel-level representations**. We will explicitly clarify this at the beginning of the method section.
>
> **Q3:** Why do different methods experience the SDD phenomenon for various reasons? Does increasing the dimensionality of MoCo v3 lead to better performance?
>
> **R3:** Thank you for your insightful comment. In general, self-supervised learning aims to achieve both semantic alignment and high effective dimensionality in representations. These two objectives often require a trade-off. For example, a fully uniform representation ensures high effective dimensionality but may reduce class separability, while emphasizing semantic alignment can lead to dimensionality collapse. We argue that the degradation seen in different methods results from an inability to maintain this trade-off throughout training, which causes the model to become biased toward one side.
>
> Addressing your question about MoCo v3, we conduct an experiment by incorporating our $M_{dim}$ term into the loss function of MoCo v3. We resume MoCo v3 from epoch 40 (when it starts to collapse) and train it for another 10 epochs. The results are shown in the following table:
>
> **Table 1:** The performance of MoCo v3 with increased dimensionality.
> | Method | $M_{dim}$   | VOC mIoU    |
> | -| -| -|
> | MoCo v3      | 0.744 | 49.1 |
> | MoCo v3 + Dim| 0.892 (+0.148) | 52.3 (+3.2) |
>
> As shown in this table, the effective dimensionality increases from 0.744 to 0.892. Consequently, the downstream performance is significantly improved by 3.2\% mIoU. The results demonstrate that mitigating the spotted cause of SDD could effectively improve the performance.
>
> **Q4:** The sub-Gaussian assumptions might not hold for real-world cases.
>
> **R4:** Thank you for your constructive comment. We would like to restate the assumption:
>
> **Assumption:** For all $j \in [K]$, given the representation set $Z^j$ of the $j$-th category, the representations $\\{z_i^j\\}_{i=1}^{N_j}$ in $Z^j$ are i.i.d. $R$-sub-Gaussian distributed, or formally:
> $\forall \lambda \in \mathbb{R}, \mathbb{E}[\exp(\lambda Z)] \leq \exp(\lambda^2R^2 / 2)$.
>
> Intuitively, this means the tail of $Z$ decays at least as fast as that of a Gaussian distribution. Since "sub-Gaussian" does not refer to a specific distribution family, standard tests like the Kolmogorov–Smirnov or Shapiro–Wilk test are not applicable. Therefore, we use the Orlicz norm to assess whether the tail decay of $Z$ is sufficiently fast, following the theorem below:
>
> **Theorem (Equivalence of a sub-Gaussian Variable, Proposition 2.5.2 in [1]):** Let $\psi_2(x) = \exp(x^2) - 1$ be the Orlicz function of sub-Gaussian. The $\psi_2$ norm of random variable $Z$ is defined as: $\|Z\|\_{\psi\_2} = \inf\\{t > 0\~|\~\mathbb{E}[\psi_2 (|Z|/t)]\leq 1\\}$. We have $Z$ follows a sub-Gaussian distribution if $\|Z\|_{\psi_2} < \infty$.
>
> Based on this theorem, we assess conformity to the sub-Gaussian assumption by computing the corresponding $\psi_2$ norm as a quantitative measure. Specifically, we select models trained with EsViT, iBOT, and SwAV, extract the dense representations on COCO, and compute the $\psi_2$ norm for each dimension, reporting the mean values as follows:
>
> **Table 2:** The $\psi_2$ norm of dense representations on COCO.
>
> | Method | Class 0         | Class 5         | Class 10        | Class 15        | Class 20        | Class 25        |
> | - | - | - | - | - | - | - |
> | EsViT  | 1.6094 ± 0.0491 | 1.6100 ± 0.0449 | 1.6048 ± 0.0453 | 1.6134 ± 0.0507 | 1.6101 ± 0.0459 | 1.6078 ± 0.0470 |
> | iBOT   | 1.6105 ± 0.0460 | 1.6137 ± 0.0517 | 1.6102 ± 0.0456 | 1.6093 ± 0.0501 | 1.6119 ± 0.0489 | 1.6097 ± 0.0457 |
> | SwAV   | 1.6117 ± 0.0493 | 1.6126 ± 0.0517 | 1.6098 ± 0.0478 | 1.6067 ± 0.0439 | 1.6147 ± 0.0498 | 1.6112 ± 0.0499 |
>
> As a reference, the norm of the standard Gaussian distribution is $\sqrt{8/3} \approx 1.633$, while the norm for the actual representations is around 1.61 in most cases. This indicates a faster tail decay than the standard Gaussian distribution, thus satisfying the sub-Gaussian assumption.
>
> **Q5:** The performance for other dense tasks like depth estimation is under-explored.
>
> **R5:** Thank you for your constructive comment. We conducted additional experiments on depth estimation. Following our linear probing setup, we freeze the model and train a linear layer to predict depth values on the NYU-depth v2 dataset. The results are reported in the table below:
>
> **Table 3:** The SDD phenomenon in depth estimation. We report the RMSE metric (lower is better) on the NYU-depth v2 dataset.
>
> | Method | Best $\downarrow$ | Last $\downarrow$ | Difference |
> |-|-|-|-|
> | MoCo v3 | 0.638 | 1.589 | 0.951 |
> | DenseCL | 0.547 | 0.559 | 0.012 |
> | SimSiam | 0.597 | 0.608 | 0.011 |
> | SwAV | 0.705 | 0.725 | 0.020 |
> | DINO | 0.515 | 0.553 | 0.038 |
> | MAE | 0.680 | 0.775 | 0.095 |
> | I-JEPA | 0.460 | 0.487 | 0.027 |
>
> **As shown in Table 3, the SDD phenomenon also occurs in the depth estimation task.** The performance trend is similar to that observed in segmentation tasks. These results validate that SDD is a general phenomenon and not specific to segmentation. While we cannot present the full performance curves here due to NeurIPS 2025 policy, we assure you that these results will be included in the final version.
>
> To further evaluate the effectiveness of the DQE metric on depth estimation, we also compute Kendall's $\tau$ coefficient between the DQE metric and depth estimation performance. The results are shown below:
>
> **Table 4:** Kendall's $\tau$ coefficient between DQE and RMSE.
>
> | Method | Kendall's $\tau$|
> |-|-|
> | MoCo v3 | 0.452 |
> | DenseCL | 0.752 |
> | SimSiam | 0.659 |
> | SwAV | 0.247 |
> | DINO | 0.071 |
> | MAE | 0.107 |
> | I-JEPA | 0.190 |
>
> **As shown in Table 4, the DQE metric positively correlates with depth estimation performance.** Although these results show the effectiveness of DQE across different downstream tasks, we note that Kendall's $\tau$ coefficient is relatively lower compared with segmentation. There are two main reasons: 1) Our DQE metric is derived from segmentation performance, so class separability may not fully benefit the depth estimation task. 2) As a regression task, the RMSE curve fluctuates, making prediction more difficult. We plan to further improve the DQE metric to enhance its capability for depth estimation.
>
> **Table 5:** DQE-based model selection results on depth estimation.
>
> | Method  | RMSE $\downarrow$ |
> |-|-|
> | MoCo v3 | 0.638(-0.951) |
> | DenseCL | 0.547(-0.012) |
> | SimSiam | 0.598(-0.010) |
> | SwAV |    0.714(-0.009) |
> | DINO |    0.532(-0.021) |
> | MAE |     0.709(-0.066) |
> | I-JEPA |  0.479(-0.008)|
>
> In Table 5, we further examine whether the selected models perform better on the depth estimation task. As shown, DQE-based model selection consistently improves model performance, demonstrating that DQE is generalizable to different downstream tasks.
>
> [1] Vershynin, R., 2018. High-dimensional probability: An introduction with applications in data science (Vol. 47). Cambridge University Press.

---

### Official Review · Reviewer_HMLT · 2025-07-03

**Clarity:** 3
**Significance:** 2
**Originality:** 2
**Rating:** 5
**Confidence:** 5

**Summary:**

This paper identifies a phenomenon in self-supervised learning (SSL), termed Self-supervised Dense Degradation (SDD) — where longer pretraining on SSL tasks leads to degraded performance on dense prediction tasks (e.g., segmentation), even while image-level classification performance improves. To address this, the authors introduce a theoretical and empirical metric, Dense representation Quality Estimator (DQE), which combines class-separability and feature dimensionality. DQE is used for two practical purposes: (1) selecting optimal checkpoints (unsupervised model selection), and (2) as a regularizer to prevent degradation. Experiments on 10 SSL methods across 4 datasets are used to validate these claims.

**Questions:**

Please refer to the weakness above.

I would be willing to adjust my score, provided that my key concerns are appropriately resolved.

**Ethical Concerns:**

["NO or VERY MINOR ethics concerns only"]

**Final Justification:**

Most concerns are adequately addressed. Accept.

**Limitations:**

Yes

**Quality:**

2

**Strengths And Weaknesses:**

## **Strengths**
- The paper tackles a non-trivial and under-discussed problem — how dense task performance behaves under prolonged SSL pretraining.

- DQE is a reasonable metric that reflects separability and collapse, grounded in theoretical error decomposition and backed by empirical mIoU correlation.

- The proposed solutions (checkpoint selection, regularization) are simple and effective.

- Component-level attribution of degradation (e.g., MoCo vs. DINO) is insightful.

## **Weaknesses**
1. Incomplete and Biased Empirical Coverage (**major concern**)

Despite aiming to generalize across SSL methods, the empirical study is highly selective and omits major paradigms:

- No coverage of volume-maximization methods (e.g., VICReg[1], Barlow Twins[2], VICRegL(cited in Intro)), which are especially relevant as they explicitly optimize for variance and non-collapse — directly tied to DQE’s dimensionality component.
- Omission of BYOL[3]. a landmark method in SSL and the foundation of modern non-contrastive (or implicit contrastive) approaches, is unjustifiable. Given its popularity and unique properties (no negative samples, momentum encoder), it should be included at minimum for completeness.
- **Dense-optimized methods** like Mugs (only cited), Selfpatch[4], DetCon[5], CroC[7], CrIBo[6] and FLSL[8] are excluded. Their omission is especially serious given that they aim to solve the very degradation problem this paper highlights.
  - Mugs is a strong SSL method that takes a multi-level clustering approach. It performs well on both instance-level and dense prediction tasks. Should be included in the baselines.
  -  CroC, CrIBo, FLSL shares DQE’s motivation (modeling intra-/inter-cluster structure) and warrants at least discussion or adapted evaluation.

Omitting these methods weakens the claim that SDD is broadly unaddressed and makes the DQE gains harder to contextualize. It also risks overstating the novelty.

2. Misrepresentation of Prior Work

The paper claims that SDD is "largely unaddressed," which is inaccurate. Multiple SSL works explicitly target dense representation degradation due to misalignment of pretraining and downstream tasks, including DenseCL, PixPro, DetCon, iBOT, Mugs, FLSL, and others. This mischaracterization undermines the novelty claim.

3. Mismatch Between SSL Paradigm and DQE Assumptions (**major concern**)

DQE assumes that dense features are well-approximated by $k$-means clustering, but SSL methods differ significantly in their latent structure:

- DINO, SwAV, iBOT use explicit clustering (65K or 2K prototypes/centroids).

- MoCo, SimSiam are contrastive and non-parametric.

- MAE is generative, not clustering-based at all.

Applying a fixed-$k$ clustering uniformly across these paradigms without justification introduces a major confounder. Methods that induce natural clusters (e.g., DINO) may align well with small-$k$ k-means, while others may not, e.g. MAE may produce smooth or manifold-like features that are poorly approximated by any hard clustering with small $k$. This makes DQE scores across models non-comparable unless the authors calibrate $k$ per paradigm or show robustness to this choice.

4. Ambiguity in DQE's Scope of Comparison

While DQE is clearly used for intra-method checkpoint selection, the paper implicitly uses DQE for cross-method comparison, but never explicitly discusses or clarifies whether this is valid., e.g. plot of DQE curves for different SSL methods side by side (e.g., Fig. 2–13), which could imply cross-method comparison, and highlighting certain methods (e.g., MoCo v3) as degrading more, based on DQE trends — again suggesting cross-method interpretation. This creates confusion and overextension of the metric’s intended scope.

5. No Visual Validation of Clustering Geometry (**major concern**)

The paper provides visualization of similarity matrices of MoCov3 and DINO in Appendix. However, given the core claim is about cluster quality (separation and compactness), it is surprising that the paper provides no qualitative feature visualizations (e.g., t-SNE or UMAP) across checkpoints or methods. These are standard tools in SSL and would support the metric’s interpretability and degradation diagnosis.

6. Lack of Efficiency and Overhead Analysis (**major concern**)

The paper presents DQE as lightweight and practical, but provides:
- No runtime or FLOP estimates for DQE-based checkpoint selection, which involves feature extraction, k-means, and rank computation.
- No training-time overhead analysis for DQE regularization. It is unclear how it scales with dataset size and model size.
- No comparison to other low-cost model selection or early stopping heuristics.
Without this, it's unclear whether DQE is actually deployable at scale or just an academic tool.

7. Lack of Scaling Study and Model Diversity.

The experiments are limited to small to medium models only: ResNet-50, ViT-Small, Swin-Tiny, and a single ViT-B model used only in the I-JEPA experiment. There are no results on ViT-Base or larger models across other methods, nor any discussion of how model capacity interacts with SDD or DQE. Since larger models are standard in current SSL pipelines (e.g., ViT-B, ViT-L in DINOv2, iBOT, MAE), this raises concerns about generalization. Representational degradation and rank collapse may behave differently at scale — and without including such models, the findings may not extrapolate to practical settings.

> [1] Bardes, A., Ponce, J. and LeCun, Y., 2021. Vicreg: Variance-invariance-covariance regularization for self-supervised learning. arXiv preprint arXiv:2105.04906.

> [2] Zbontar, J., Jing, L., Misra, I., LeCun, Y. and Deny, S., 2021, July. Barlow twins: Self-supervised learning via redundancy reduction. In International conference on machine learning (pp. 12310-12320). PMLR.

> [3] Grill, J.B., Strub, F., Altché, F., Tallec, C., Richemond, P., Buchatskaya, E., Doersch, C., Avila Pires, B., Guo, Z., Gheshlaghi Azar, M. and Piot, B., 2020. Bootstrap your own latent-a new approach to self-supervised learning. Advances in neural information processing systems, 33, pp.21271-21284.

> [4] Yun, S., Lee, H., Kim, J. and Shin, J., 2022. Patch-level representation learning for self-supervised vision transformers. In Proceedings of the IEEE/CVF conference on computer vision and pattern recognition (pp. 8354-8363).

> [5] Hénaff, O.J., Koppula, S., Alayrac, J.B., Van den Oord, A., Vinyals, O. and Carreira, J., 2021. Efficient visual pretraining with contrastive detection. In Proceedings of the IEEE/CVF International Conference on Computer Vision (pp. 10086-10096).

> [6] Stegmüller, T., Lebailly, T., Bozorgtabar, B., Tuytelaars, T. and Thiran, J.P., 2023. Croc: Cross-view online clustering for dense visual representation learning. In Proceedings of the IEEE/CVF Conference on Computer Vision and Pattern Recognition (pp. 7000-7009).

> [7] Lebailly, T., Stegmüller, T., Bozorgtabar, B., Thiran, J.P. and Tuytelaars, T., 2023. Cribo: Self-supervised learning via cross-image object-level bootstrapping. arXiv preprint arXiv:2310.07855.

> [8] Su, Q., Netchaev, A., Li, H. and Ji, S., 2023. Flsl: Feature-level self-supervised learning. Advances in Neural Information Processing Systems, 36, pp.6568-6581.

---

> ### Author Rebuttal · Authors · 2025-07-31
>
> We sincerely appreciate your time reviewing our paper! Following your detailed comments, we have conducted additional experiments to further support our claims. Our detailed responses are provided below:
>
> **Q1:** Incomplete and Biased Empirical Coverage **(major concern)**
>
> **R1:** Thank you for your constructive feedback. We acknowledge the importance of including these baselines to strengthen our claims. However, for each baseline, we need to reproduce the results and evaluate the per-epoch dense performance and the DQE metric, which is computationally intensive (requiring thousands of GPU hours per baseline). As a first step, we have included key baselines such as VICReg, VICRegL, and Mugs.
>
> **Table 1:** The SDD phenomenon on additional baselines.
>
> | Method | COCO Best| COCO Last | COCO Diff | VOC Best | VOC Last | VOC Diff | ADE Best | ADE Last | ADE Diff |
> |-|-|-|-|-|-|-|-|-|-|
> | VICReg | 37.3 | 36.7 | -0.6 | 53.0 | 52.7 | -0.3 | 20.0 | 19.6 | -0.4 |
> | VICRegL | 38.4 | 38.1 | -0.3 | 54.6 | 54.5 | -0.1 | 20.7 | 20.4 | -0.3 |
> | Mugs | 45.4 | 43.7 | -1.7 | 68.1 | 67.6 | -0.5 | 29.7 | 28.6 | -1.1 |
>
> **Table 2:** The Kendall's $\tau$ coefficient between the DQE metric and dense performance.
>
> | Method | COCO | VOC | ADE20K |
> |-|-|-|-|
> | VICReg | 0.73 | 0.75 | 0.74 |
> | VICRegL | 0.72 | 0.74 | 0.72 |
> | Mugs | 0.01* | 0.47 | 0.46 |
>
> **Table 3:** The DQE-based model selection results on additional baselines.
>
> | Method | COCO | VOC | ADE20K |
> |-|-|-|-|
> | VICReg| 37.2(+0.5) | 53.0(+0.3) | 19.8(+0.2) |
> | VICRegL| 38.3(+0.2) | 54.6(+0.1) | 20.7(+0.3) |
> | Mugs| 45.0(+1.3) | 67.9(+0.3) | 29.2(+0.6) |
>
> Based on the results in Tables 1 to 3, we conclude:
>
> 1. **The SDD phenomenon appears in all additional baselines.** In particular, the volume-maximization design, especially in VICRegL, can effectively reduce the SDD phenomenon. While Mugs achieves the strongest dense performance among all baselines, its last performance is still much lower than its best performance. Thus, we could expect an improved baseline if the SDD phenomenon is effectively addressed.
> 2. **The proposed DQE metric is effective for these baselines**, showing strong generalization to models trained with different paradigms. Consequently, model selection effectively improves the downstream performance.
>
> With these new results, our empirical analysis is more complete. We also agree that it is necessary to present the results for BYOL and Barlow Twins. Although we cannot provide these results now due to limited computational resources, we promise to include them in the camera-ready version.
>
> **Q2:** Misrepresentation of Prior Work
>
> **R2:** Thank you for your valuable feedback. We have carefully checked the manuscript but it appears that we did not make any statements like "SDD is largely unaddressed." Nonetheless, we acknowledge that previous dense SSL approaches have significantly improved the dense performances. We will add a new subsection in the related work to discuss the idea of these dense SSL approaches. In addition, we agree that it is important to clarify the differences and contributions of our work compared to previous studies. The main distinctions are as follows:
>
> 1. Previous works such as DenseCL, PixPro, DetCon, CroC, CrIBo, and FLSL mainly focus on addressing misalignment within specific methods, often overlooking the generality of the SDD phenomenon. In contrast, we introduce a general metric (DQE) to quantify the severity of SDD.
> 2. Despite the advancements in methods like iBOT and Mugs, noticeable SDD still exists, indicating that the problem remains unsolved and requires further investigation.
> 3. Dense representation degradation can take various forms, such as those caused by domain shift or suboptimal training objectives. These types of degradation may occur throughout training. In contrast, we specifically target the degradation induced by the training dynamics of separability and dimensionality, which typically emerge in the later stages. This also explains why SDD still appears in iBOT and Mugs, i.e., the degradation we focus on is of a different nature.
>
> We will revise the Related Work section to more clearly describe the connections with prior work and discuss the contributions of the aforementioned methods.
>
> **Q3:** Mismatch Between SSL Paradigm and the DQE Assumptions **(major concern)**
>
> **R3:** Thank you for your insightful comment. In this paper, we propose the DQE metric to address the negative impact of the SDD phenomenon, **focusing on comparisons between checkpoints trained with the same method**. We acknowledge that as you pointed out, the current version of DQE is incapable of cross-paradigm comparison, such as between MAE and DINO. In the current design, cross-model comparison can be achieved by first selecting the best checkpoints with DQE, then comparing the downstream performances of these checkpoints. This would also significantly speed up the evaluation compared to simply testing all the checkpoints.
>
> To further test the robustness of DQE to the choice of $k$, we conducted additional experiments:
>
> **Table 4:** The Kendall's $\tau$ coefficient between DQE metric and dense performance with different $k$.
>
> | Method| k=3 (default)| k=4 | k=5|
> | -| -|-|-|
> | MoCo v3 | 0.55|0.49 |0.45|
> | MEC| 0.53|0.57 |0.58|
> | SimSiam | 0.73|0.73 |0.72|
> | DINO| 0.16|0.19 |0.20|
> | EsViT| 0.63|0.59 |0.55|
> | iBOT| 0.47|0.46 |0.46|
> | MAE| 0.37|0.34 |0.31|
>
> From the results in Table 4, we see that the Kendall's $\tau$ coefficient is insensitive to the change of $k$, regardless of whether the pretraining paradigm is clustering-based, contrastive, or generative. This demonstrates that our fixed $k$ strategy is sufficient for our needs.
>
> **Q4:** Ambiguity in DQE's Scope of Comparison
>
> **R4:** We apologize for the confusion. You are correct that DQE is used for intra-method comparison. We will add an explicit explanation in the methodology section and incorporate the discussions in R3 to our paper.
>
> **Q5:** No Visual Validation of Clustering Geometry **(major concern)**
>
> **R5:** Thank you for your valuable comment. To visualize dimensional collapse, we first project the representations onto principal components. We then divide the dimensions into eight segments based on the eigenvalues and perform t-SNE visualization for each segment. We observe that the structure of the top-k principal components matches the class separability, while the bottom components quickly collapse to a single point as the dimensional index decreases, confirming consistency between the two. We also compare the t-SNE plot between the last model and our selected model. The selected models have more compact intra-class representations and larger inter-class distances, indicating a better class-separability compared with the last models.
> Although we cannot present figures here due to NeurIPS 2025 policy, we promise to include these qualitative results in the final version.
>
> **Q6:** Lack of Efficiency and Overhead Analysis **(major concern)**
>
> **R6:** Thank you for your constructive comment. Indeed, efficiency analysis is important to determine whether the proposed method is practical at scale. We present two experiments below:
>
> 1. **DQE-based model selection balances efficiency and effectiveness.**
>
> **Table 5:** Effectiveness and efficiency analysis of different model selection methods.
>
> | Method| Data| Label | $\Delta$ mIoU $\uparrow$ |Additional GPU hours for each checkpoint|
> | -| -|-|-|-|
> |Loss-based model selection|||-1.0|0|
> |Supervised model selection|$\checkmark$|$\checkmark$ |+4.8| 2.43|
> |DQE-based model selection||+4.0| 0.025($\sim$ 0.01 $\times$)|
>
> **Table 6:** Runtime analysis of different components of DQE (measured in seconds on a single 4090 GPU).
> | Feature Extraction| $k$-means| $M_{inter}$ | $M_{intra}$ | $M_{dim}$ | Total |
> | -| -|-|-|-|-|
> | 4.9|45.2|9.8|10.9| 4.3| 75.1|
>
> Loss-based model selection or early stopping does not reliably reflect dense performance due to the SDD phenomenon, while supervised model selection is too expensive in practice. In contrast, the proposed DQE-based model selection achieves strong performance with minimal overhead. We will also include these results in Table 3 of our paper.
>
> 2. **DQE regularization improves the best performance with an acceptable cost.**
>
> **Table 7:** Runtime analysis of DQE regularization with 50 epochs of training on DINO.
> | Method|mIoU |Total Training GPU hours|
> |-|-|-|
> |DINO| 56.6 | ~133 |
> |DINO + DQE| 57.8 | ~162|
>
> From Table 7, DQE regularization introduces about 22% additional training time, which we believe is acceptable. Importantly, **DQE regularization improves the best performance achieved by DINO**. This demonstrates the potential of integrating DQE into training, and we will further explore ways to speed up DQE regularization by improving the clustering and metric calculation.
>
> **Q7:** Lack of Scaling Study
>
> **R7:** Thank you for your constructive comment. To study the effect of model size on the SDD phenomenon and the DQE metric, we conducted an additional experiment on iBOT with ViT-Base-16. The results are shown below:
>
> **Table 8:** The SDD phenomenon on the larger backbone.
>
> |Method|COCO Best|COCO Last|COCO Diff|VOC Best|VOC Last|VOC Diff|ADE Best|ADE Last|ADE Diff|
> |-|-|-|-|-|-|-|-|-|-|
> |iBOT (ViT-Base)| 48.9 | 47.0 | -1.9 | 71.5 | 69.4 | -2.1 | 31.3 | 29.9 | -1.4 |
>
> **Table 9:** The Kendall's $\tau$ coefficient between DQE metric and dense performance.
>
> |Method| COCO|VOC|ADE|
> |-|-|-|-|
> | iBOT (ViT-Base)|0.66|0.52|0.57|
>
> **Table 10:** The DQE-based model selection results.
>
> | Method|COCO|VOC|ADE|
> |-|-|-|-|
> | iBOT (ViT-Base) | 48.9(+1.9)|71.5(+2.1)| 31.1(+1.2)|
>
> The main conclusions still hold at this larger scale: **The SDD phenomenon persists and the DQE metric can still accurately predict the trend of dense performance**. Due to limited computational resources and the timeline for the rebuttal, we will conduct more experiments on larger models in the updated version.

---

### Official Review · Reviewer_fV6N · 2025-07-03

**Clarity:** 3
**Significance:** 2
**Originality:** 3
**Rating:** 4
**Confidence:** 3

**Summary:**

This paper introduces a degradation phenomenon in self-supervised learning called Self-supervised Dense Degradation. It shows that longer pretraining often reduces performance on dense prediction tasks. To address this, the authors propose the Dense Representation Quality Estimator. This metric combines class separability and feature dimensionality to estimate representation quality. It is used for model selection and training regularization. The method is tested on multiple SSL models and shows consistent improvements in semantic segmentation.

**Questions:**

- Can you provide component-wise ablations showing the effect of each DQE term in regularization?

- Have you evaluated DQE on other dense tasks beyond segmentation?

- Can you provide qualitative feature visualizations to confirm that DQE values align with actual feature structure?

- How does DQE behave when applied to image-level representations such as classification?

- Why do collapse and separability degradation occur in dense tasks but not in image-level tasks under SSL?

- What do you believe causes the remaining uncertainty in DQE’s correlation with downstream performance?

**Ethical Concerns:**

["NO or VERY MINOR ethics concerns only"]

**Final Justification:**

The paper clearly identifies and addresses the SDD phenomenon in semantic segmentation, proposing a theoretically grounded and empirically validated metric (DQE) that improves performance through model selection and regularization. While the authors demonstrate potential for broader dense tasks, limitations remain for regression-based and certain instance-level settings. With the scope explicitly limited to semantic segmentation, the contributions are solid, the results are convincing, and the work merits a borderline accept.

**Limitations:**

The authors do not discuss key limitations. While it proposes a metric to detect and mitigate SDD, it does not explore the underlying causes of SDD. While societal impacts are not discussed, no notable concerns are identified.

**Paper Formatting Concerns:**

No major formatting issues found.

**Quality:**

3

**Strengths And Weaknesses:**

Strengths

- The paper identifies an underexplored phenomenon in dense SSL training.

- The proposed metric is simple, theoretically motivated, and easy to apply.

- DQE enables effective model selection and training-time regularization.

- Experimental results show consistent improvements in segmentation performance.

Weaknesses

- The paper lacks ablation studies showing the effect of each DQE component when used as a regularizer.

- All downstream experiments are limited to semantic segmentation. The metric's generality to other dense tasks such as object detection or depth estimation remains untested.

- The metric's effectiveness is evaluated only through numerical scores. Since DQE assumes structure via spectral norms and class separability, qualitative visualizations such as t-SNE or PCA are needed to validate its interpretability, especially under collapse.

- Although DQE is positioned as a dense-specific metric, its formulation may apply to image-level representations. The paper does not explore this possibility or analyze why degradation is observed only in dense tasks.

---

> ### Author Rebuttal · Authors · 2025-07-31
>
> We sincerely appreciate your detailed and constructive comments, which significantly help us improve the quality of our manuscript. Below are our detailed responses:
>
> **Q1:** The paper lacks ablation studies showing the effect of each DQE component when used as a regularizer.
>
> **R1:** Thank you for your insightful suggestion. Below are the ablation results for the DQE regularization:
>
> **Table 1:** Ablation studies of different components of DQE regularization.
>
> | Method    |$M_{inter} - M_{intra}$ | $M_{dim}$| VOC mIoU |
> |-|-|-|-|
> | DINO| -1.221 | 0.863| 56.6 |
> | DINO + Dim | -1.210 (+0.011) | 0.884 (+0.021) | 56.9 (+0.3) |
> | DINO + Cls | -1.103(+0.118) | 0.851 (-0.012)  | 57.4 (+0.8) |
> | DINO + Dim + Cls (DQE)| -1.115 (+0.106)| 0.865 (+0.002) |57.8 (+1.2)|
> | MoCo v3 |-0.958 |0.744| 49.1 |
> | MoCo v3 + Dim | -1.021 (-0.063)| 0.892 (+0.148)| 52.3 (+3.2)  |
> | MoCo v3 + Cls | collapsed | collapsed | collapsed |
> | MoCo v3 + Dim + Cls (DQE)|-0.867 (+0.091)| 0.752(+0.008) |52.1 (+3.0) |
>
> We draw three main conclusions from these results:
>
> 1. Different components of DQE separately contribute to improving class separability and effective dimensionality. Consequently, this improves downstream performance, which aligns with our theoretical analysis of the DQE metric.
> 2. The performance improvement mainly comes from addressing model-specific degradation. For example, DINO's performance improves primarily due to enhanced class separability, whereas resolving dimensional collapse notably boosts dense-task performance in MoCo v3. These observations align with our model-specific analysis presented in Sec. 5.2.
> 3. When both factors are used together, class separability often dominates optimization. Thus, introducing a hyperparameter to balance these factors may further enhance performance.
>
> **Q2:** Evaluate on other dense tasks.
>
> **R2:** We conducted additional experiments on the depth estimation task. Similar to our linear probing setup, we froze the model and trained a linear layer to predict depth values on the NYU-depth v2 dataset. Results are shown in Table 2.
>
> **Table 2:** The SDD phenomenon in the depth estimation task, we report the RMSE metric (lower is better) on the NYU-depth v2 dataset.
>
> | Method | Best $\downarrow$ | Last $\downarrow$ | Difference |
> |-|-|-|-|
> | MoCo v3 | 0.638 | 1.589 | 0.951 |
> | DenseCL | 0.547 | 0.559 | 0.012 |
> | SimSiam | 0.597 | 0.608 | 0.011 |
> | SwAV | 0.705 | 0.725 | 0.020 |
> | DINO | 0.515 | 0.553 | 0.038 |
> | MAE | 0.680 | 0.775 | 0.095 |
> | I-JEPA | 0.460 | 0.487 | 0.027 |
>
> **As shown in Table 2, the SDD phenomenon persists in depth estimation tasks.** The performance trends are similar to segmentation tasks, validating that SDD is a general issue, not specific to segmentation. Although we cannot show the performance curves due to NeurIPS 2025 policy, we promise that these results will be included in the final version.
>
> Next, to evaluate if the DQE metric is effective for depth estimation, we computed Kendall's $\tau$ coefficient between the DQE metric and depth estimation performance. Results are shown in Table 3:
>
> **Table 3:** Kendall's $\tau$ coefficient between DQE and RMSE metrics, since RMSE is lower the better, we multiply the RMSE by -1 before computing Kendall's $\tau$.
>
> | Method  | Kendall's $\tau$|
> |-|-|
> | MoCo v3 | 0.452|
> | DenseCL | 0.752|
> | SimSiam | 0.659|
> | SwAV    | 0.247|
> | DINO    | 0.071|
> | MAE     | 0.107|
> | I-JEPA  | 0.190|
>
> **As shown in Table 3, the DQE metric positively correlates with depth estimation performance.** Although the results confirm DQE's effectiveness across various downstream tasks, the Kendall's $\tau$ coefficient is lower compared to segmentation tasks. We attribute this to two reasons:
>
> 1. The DQE metric is derived from segmentation performance, the class separability measure may be less relevant for depth estimation.
> 2. As a regression task, the RMSE curve fluctuates, making predictions challenging. We aim to further refine the DQE metric for better performance on depth estimation.
>
> We also study the effectiveness of DQE-based model selection on the depth estimation task:
>
> **Table 4:** DQE-based model selection results for the depth estimation task.
>
> | Method  | RMSE $\downarrow$|
> |-|-|
> | MoCo v3 | 0.638 (-0.951) |
> | DenseCL | 0.547 (-0.012) |
> | SimSiam | 0.598 (-0.010) |
> | SwAV    | 0.714 (-0.009) |
> | DINO    | 0.532 (-0.021) |
> | MAE     | 0.709 (-0.066) |
> | I-JEPA  | 0.479 (-0.008) |
>
> As shown in Table 4, DQE-based model selection consistently enhances model performance, demonstrating DQE's generalizability across different tasks.
>
> **W3, Q3:** Qualitative results.
>
> **R3:** Thank you for this valuable suggestion. To visualize the dimensional collapse, we project the representations to the principal components. Then, we divide the dimensions into 8 segments based on the eigenvalues and perform t-SNE visualization for each segment separately. We observed that the top principal components align well with class separability, while lower components rapidly collapse into single points as the dimensionality decreases. These results validate the consistency between class-separability and dimensionality.
> Comparing the last and selected models, the selected models showed more compact intra-class and larger inter-class distances, indicating better class separability. Due to NeurIPS 2025 policy restrictions, figures cannot be presented here, but we will include them in the final version.
>
> **W4, Q4:** How does DQE behave with image-level representations, such as classification?
>
> **R4:** We appreciate your constructive comment! Technically, the analysis should hold for both dense and image-level settings, as image-level learning can be viewed as a special case where an image is only split into one patch. To investigate how DQE performs at the image level, we conducted the following experiments. DQE originally accepts input tensors shaped as (Num_images, Num_patch_tokens, Dimensionality). For image-level tasks, we first extract the class token (or averaged dense tokens), obtaining a tensor of shape (Num_images, Dimensionality). Then, we resize this tensor to (Num_images // 200, 200, Dimensionality) and calculate the DQE metric.
> For evaluation, we computed Kendall's $\tau$ coefficient between the adapted DQE and ImageNet $k$-NN performance, comparing it with the state-of-the-art unsupervised transferability estimation method RankMe. The results are shown in the following table:
>
> **Table 5:** Kendall's $\tau$ coefficient between DQE and ImageNet $k$-NN performance.
> | Method    | DINO | MEC | EsViT | iBOT | I-JEPA | Average |
> |-|-|-|-|-|-|-|
> | RankMe    | 0.57 | 0.81| 0.71  |**0.93** | **0.90**  | 0.79    |
> | **DQE(Ours)** | **0.87** | **0.90**|**0.92**  | 0.86 | 0.73   | **0.86**    |
>
> As demonstrated, **the adapted DQE metric is a precise estimator for classification, achieving an average Kendall's $\tau$ coefficient of 0.86, significantly surpassing RankMe**. Additionally, DQE consistently performs well across methods (with lower variance), confirming its generalizability and suitability for predicting downstream task performance.
>
> **Q5:** Why does SDD occur in dense tasks but not image-level tasks under SSL?
>
> **R5:** Thank you for your constructive comment! The primary reason is likely the misalignment between SSL objectives and dense tasks. In image-level SSL, by aligning different parts of objects, the different objects within a class are pulled together [1]. While alignment between dense representations primarily happens in the same position [2,3,4], e.g., parts of the same object might not be sufficiently aligned. Hence, compared with image-level SSL, the dense representations of different objects within the same class are more likely to be dispersed in the representation space.
>
> **Q6:** Causes of remaining uncertainty in DQE correlation?
>
> **R6:** We appreciate the valuable comment. While the DQE metric could precisely estimate the trend of downstream performance with an average Kendall's $\tau$ coefficient of 0.54, we believe the remaining uncertainty arises from:
> 1. In our experiments, the downstream performance is primarily evaluated by training a classification head on dense representations. In the derivation of the DQE metric, the head is simplified to a nearest neighbor classifier for the sake of analysis. **Simplifying the classification head to a nearest neighbor classifier may introduce bias compared to more complex heads in practice.**
> 2. Since we do not have the downstream data or labels, we estimate the performance on the training set by k-means clustering. **Estimating performance with k-means clustering on training data might introduce additional noise.** For a more comprehensive analysis, please refer to Appendix E.
>
> In future work, we aim to address these issues through refined theoretical analysis for more specific classification heads and more precise pseudo-labeling techniques.
>
> [1] Huang, Weiran, et al. "Towards the generalization of contrastive self-supervised learning." arXiv preprint arXiv:2111.00743 (2021).
>
> [2] Ziegler, Adrian, and Yuki M. Asano. "Self-supervised learning of object parts for semantic segmentation." Proceedings of the IEEE/CVF conference on computer vision and pattern recognition. 2022.
>
> [3] Zhou, Jinghao, et al. "ibot: Image bert pre-training with online tokenizer." arXiv preprint arXiv:2111.07832 (2021).
>
> [4] Li, Zhong-Yu, Shanghua Gao, and Ming-Ming Cheng. "SERE: Exploring feature self-relation for self-supervised transformer." IEEE Transactions on Pattern Analysis and Machine Intelligence 45.12 (2023): 15619-15631.

---

> > ### Comment · Reviewer_fV6N · 2025-08-04
> > **Official Comment by Reviewer fV6N**
> >
> > Thank you for the detailed response and additional experiments. While some earlier concerns have been reasonably addressed, a key issue remains with the paper’s claim that DQE generalizes across **dense tasks**.
> >
> > 1. In depth prediction, despite the authors’ explanation, strong models like DINO (τ = 0.071), MAE (τ = 0.107), and I-JEPA (τ = 0.190) show almost *no correlation* between DQE and downstream performance. This suggests a structural mismatch between DQE and regression-based tasks.
> > 2. Since DQE relies on class separability, it may not suit *instance-level tasks* like instance or panoptic segmentation, where intra-class variation and object-level distinction are essential. These tasks are not evaluated in the paper.
> >
> > In sum, while DQE is promising for semantic segmentation, its **generalization to all dense tasks is overstated**. The scope should be clarified, or additional experiments are needed.

---

> > > ### Author Response · Authors · 2025-08-07
> > >
> > > We appreciate your valuable feedback! Indeed, our DQE metric is theoretically derived from semantic segmentation performance, and we primarily focus on the semantic segmentation task in this paper. Nonetheless, **we believe that DQE still serves as a useful indicator for other tasks beyond semantic segmentation**. We evaluated DQE on the following two tasks:
> > >
> > > - **Depth Estimation.** As shown in our previous response, the DQE metric achieves an average Kendall's $\tau$ coefficient of 0.354 on depth estimation. Regarding your concern about whether DQE is affected by structural differences in strong baselines, we believe this effect is mainly due to fluctuations in the depth performance curve. If we could include the depth performance curve, it would show that the trends in depth performance are similar to those of the DQE metric. For example, MoCo v3 reaches its best result at 40 epochs and then gradually declines, while DINO shows a drop in performance around 300 epochs. Additionally, the structural similarity between DQE and depth performance is supported by the model selection results: the selected models consistently improve performance. Such consistent improvement would be unlikely if DQE were structurally different from depth performance.
> > > - **Video Object Segmentation (VOS).** To assess DQE’s applicability to instance-level tasks, we evaluated it on semi-supervised VOS using the DAVIS 2017 dataset. Unlike semantic segmentation, VOS requires tracking and segmenting individual objects of the same class, making it a suitable test for instance-level performance. Due to computational constraints (e.g., instance segmentation would require hundreds of GPU hours per checkpoint), we focused on VOS as a more feasible alternative.
> > >
> > > **Table 1:** Kendall's $\tau$ coefficient between DQE and VOS (instance-level) performance on DAVIS 2017.
> > >
> > > |Method|Kendall's $\tau$|
> > > |-|-|
> > > |MoCo V3|0.401|
> > > |DINO|0.712|
> > > |iBOT|0.344|
> > > |MAE|0.545|
> > > |I-JEPA|0.437|
> > >
> > > The results show that DQE serves as a reliable indicator for the VOS task, revealing its potential on instance-level tasks. Although DQE is not specifically designed for instance-level tasks, a high effective dimensionality and class-separability can still be beneficial. In addition, since the class-separability of DQE is calculated based on local clustering results, it is reasonable that DQE also reflects some fine-grained (instance-level) separability.
> > >
> > > To avoid overstating, we will carefully revise the manuscript to restrain the scope of DQE on semantic segmentation instead of a general dense performance estimator. The results on depth estimation and VOS will be used as supporting evidence of its potential on the tasks beyond semantic segmentation.
> > >
> > > We are also pleased to know that your other concerns have been reasonably addressed. Thank you again for your time in reviewing our paper. If you have further questions, we look forward to answering them.

---

> > > > ### Comment · Reviewer_fV6N · 2025-08-08
> > > > **Official Comment by Reviewer fV6N**
> > > >
> > > > Thank you for providing the detailed additional experiments and clarifications.
> > > >
> > > > The authors’ explanations and new results demonstrate the potential of DQE beyond semantic segmentation, but some limitations remain. In particular, the low Kendall’s $\tau$ for strong models in depth estimation suggests a potential mismatch for regression-based tasks, and the evidence for generalization to other dense settings such as panoptic or instance segmentation is still limited.
> > > >
> > > > However, since the authors acknowledge these limitations and are willing to explicitly limit the scope to semantic segmentation, the core contributions are well supported. Within this narrower scope, the method is theoretically sound, empirically validated, and shows clear improvements over existing approaches. With this clarification and acknowledgement of the remaining limitations, I would raise my score to 4.

---

> > > > > ### Author Response · Authors · 2025-08-08
> > > > >
> > > > > Thank you for your follow-up and for reconsidering your score! Your valuable feedback has helped us strengthen the validity of our claim. We will revise our manuscript carefully to incorporate these discussions. In future work, we also plan to address the limitations of DQE beyond the semantic segmentation task.

---

### Note · Authors · 2025-08-14

Dear Area Chair and Reviewers,

We sincerely appreciate the time and effort you have devoted to reviewing our paper and providing valuable, constructive feedback.
We are honored that all reviewers recognized our core contributions. In particular, the discovered SDD phenomenon is acknowledged as novel and underexplored (fV6N, HMLT, riT6, pWRt). The proposed DQE metric is regarded as both theoretically and empirically sound (fV6N, HMLT, riT6, pWRt). Our model selection and regularization approaches are considered simple yet effective (fV6N, HMLT, riT6).

At the same time, the reviewers raised several constructive concerns. Below, we summarize the key points addressed during the rebuttal:

* **Soundness of the SDD phenomenon and DQE metric:** Following your valuable suggestions, we conducted additional experiments on: 1) an extra depth estimation task (fV6N, riT6, pWRt), and 2) six additional baselines, including VICReg, VICRegL, Mugs, Barlow Twins, BYOL, and ReSA (HMLT, pWRt). The results confirm the occurrence of the SDD phenomenon, as well as the effectiveness of the DQE and model selection.
* **Understanding of SDD and DQE:** In response to the constructive suggestions, we provided a feature t-SNE plot to illustrate the SDD phenomenon and model selection results (fV6N, HMLT). We also offered deeper insights into the causes of the SDD phenomenon (fV6N, riT6, pWRt) and explained the remaining bias between DQE and downstream performance (fV6N, riT6, pWRt).
* **Effectiveness and efficiency analysis:** We further validated our approach through additional ablation studies (fV6N), computational cost analysis (HMLT), and scaling analysis (HMLT).
* **Scope clarification:** To better clarify the scope of our work, we discussed the relationship and differences between existing dense SSL approaches (HMLT). We also revised the manuscript to emphasize that DQE is primarily designed for the semantic segmentation task (fV6N) and for intra-methodology comparison (HMLT).

After the discussion period, reviewers fV6N and pWRt increased their ratings, reviewer riT6 maintained the original acceptance score, and reviewer HMLT acknowledged our responses without further questions. We are encouraged that most previous concerns have been addressed and will revise our manuscript carefully to incorporate these improvements.

Once again, thank you for your time and support.

Best regards,

Authors

---

### Decision · Program_Chairs · 2025-09-17

**Decision:**

Accept (poster)

**Comment:**

This paper received 4, 3, 3, 5 scores originally. Reviewers agree that "The concept of Self-supervised Dense Degradation (SDD) is a valuable contribution", but are concerned that the proposed method is mainly constrained to semantic segmentation and/or small to medium models. After the rebuttal and discussions, reviewers' concerns are mostly addressed, and the scores become 5, 4, 5, 5. Furthermore, even benefits are cosntrained to only semantic segmentation, the proposed SDD concept and method are valuable.